# Does non-stationarity induced by multiyear drought invalidate the paired-catchment method?

Yunfan Zhang[1, 2], Lei Cheng[1, 2, *], Lu Zhang[1, 3], Shujing Qin[1, 2], Liu Liu[4], Pan Liu[1, 2], and Yanghe Liu[1, 2]

[1] State Key Laboratory of Water Resources and Hydropower Engineering Science, Wuhan University, Wuhan 430072, China.

[2] Hubei Provincial Collaborative Innovation Center for Water Resources Security, Wuhan 430072, China.

[3] CSIRO Land and Water, Black Mountain, Canberra, ACT 2601, Australia.

[4] College of Water Resources and Civil Engineering, China Agricultural University, Beijing 100083, China.

*Correspondence to*: Lei Cheng (lei.cheng@whu.edu.cn)

**Abstract.** Multiyear drought has been demonstrated to cause non-stationary rainfall-runoff relationship. But whether this change can occur in catchments that have also experienced vegetation change and whether it invalidates the most widely used methods for estimating impacts of vegetation change (i.e., the paired-catchment method (PCM), the time-trend method (TTM), and the sensitivity-based method (SBM)) on runoff is still unknown and rarely discussed. Estimated inconsistent afforestation impacts were 32.8%, 93.5%, and 76.1% of total runoff changes in the Red Hill paired experimental catchments in Australia during the period of 1990–2015 by the PCM, TTM and SBM, respectively. In addition to afforestation, the Red Hill paired experimental catchments have experienced a 10-year drought (2000–2009) and have been demonstrated to lead to non-stationary rainfall-runoff relationships of paired catchments. Estimated impacts of vegetation change by the PCM (32.8%) is still reliable and is not invalided by multiyear drought induced non-stationarity because the PCM can eliminate all impacts by different factors on paired catchments (multiyear drought and climate variability) except the purposed treatment (afforestation). For the TTM and SBM, traditional application did not further differentiate different drivers of non-stationary rainfall-runoff relationship (i.e., multiyear drought and vegetation change), which led to significant overestimation of afforestation effects. A new framework was further proposed to separate the effects of three factors on runoff changes including vegetation change, climate variability and hydroclimatic non-stationarity (i.e., multiyear drought). Based on the new framework, impacts of multiyear drought and climate variability on runoff of the control catchment (Kileys Run) were 87.2% and 12.8%, respectively. Impacts of afforestation, multiyear drought and climate variability on runoff of the treated catchment (Red Hill) were 32.8%, 54.7% and 23.9%, respectively. Impacts of afforestation on runoff were 38.8% by the TTM and 21.4% by the SBM, agreeing well with that by the PCM (32.8%). This study not only demonstrated that multiyear drought can induce non-stationary rainfall-runoff relationship using field observations, but also proposed a new framework to better separate the impact of vegetation change on runoff under climate-induced non-stationary condition. More importantly, it is shown that non-stationarity induced by multiyear drought does not invalidate the PCM, and PCM is still the most reliable method even the control catchment experienced climate-induced shift in the rainfall-runoff relationship.

## 1 Introduction

Vegetation change can exert significant impacts on catchment runoff (Farley et al., 2005; Filoso et al., 2017; Hallema et al., 2018). In addition to vegetation change, climate variability can also cause changes in catchment flow regimes and water yield (Kim et al., 2011; Ryberg et al., 2012). Understanding of the response of runoff to vegetation change was mainly gained through the use of paired catchment experiments over the past century (Wei et al., 2018). The paired catchment method (PCM), which is the standard

approach for quantifying the effects of forest management on runoff, is based on paired catchment experiments and is still used today (Van Loon et al., 2019). However, separating the effects of vegetation change and climate variability on runoff remains a great challenge due to the complex interactions between climate variability and vegetation change (Cavalcante et al., 2019; Jones et al., 2006; Zhang et al., 2021). Moreover, persistent hydroclimatic non-stationary changes observed during the past few decades have increased both temperatures and occurrences of extreme weather events (such as multiyear drought). These changes have led to non-stationary rainfall-runoff relationships in many catchments around the world (Li et al., 2018; Wang et al., 2013; Zhang et al., 2016). Therefore, the combined effect of these influencing factors will lead to greater uncertainty in estimating the impact of vegetation change on runoff using different methods. In particular, is the paired catchment method still valid under non-stationary rainfall-runoff relationships?

Hydroclimatic non-stationary changes such as multiyear drought-induced non-stationarity in rainfall-runoff relationships has been reported in some catchments around the world, such as prolonged drought in the United States (Griffin and Anchukaitis, 2014) Amazonia (Lewis et al., 2011), and China (Tian et al., 2018). It is widely known that Australia experienced multiyear drought (known as the Millennium Drought) between 1997 and 2009 (King et al., 2020; Peterson et al., 2021). Some studies have also reported that stationary rainfall-runoff relationships in southeast regions of Australia were broken by multiyear drought (Chiew et al., 2014; Petrone et al., 2010; Saft et al., 2016). Multiyear drought can also lead to shift in catchment rainfall-runoff relationship (or non-stationarity) as vegetation change and thus pose great challenges to the basic idea of methods for quantifying runoff changes caused by vegetation change, and how to separate the effects of vegetation change under multiyear drought conditions.

Three commonly used methods for separating the impacts of vegetation change on catchment water yield are the paired-catchment method (PCM), the time-trend method (TTM), and the sensitivity-based method (SBM). The PCM requires a control and treated catchment located in close proximity and the primary role of the control catchment is to eliminate the impact of climate change on runoff. Essentially, observations from the control catchment can remove the effect of all factors that lead to change in the rainfall-runoff relationship except vegetation change between two paired catchments (Lee, 1980). This method has been applied in many paired catchments around the world to provide fundamental understanding and knowledge for water resource management under vegetation change (Brown et al., 2005; Stoof et al., 2012; Van Loon et al., 2019). The time-trend method (TTM) assumes that the rainfall-runoff relationship driven by climate variability during pre- and post-change periods is stationary. Thus the impact of vegetation change on runoff is obtained as the difference between observed runoff during post-change period and estimated runoff based on the rainfall-runoff relationship obtained during the pre-change period (Lee, 1980; Zhang et al., 2019; Zhao et al., 2010). The sensitivity-based method (SBM) is a combination of the Budyko framework (Budyko, 1974) and the elastic response of runoff to rainfall and potential evapotranspiration developed by Zhang et al. (2001). The direct result from the SBM is runoff change caused by climate variability, and the effect of vegetation change on runoff is derived by subtracting the effects of climate variability on runoff from total runoff changes. Generally, the PCM, TTM and SBM should provide consistent results for a specific catchment where non-stationary change in the rainfall-runoff relationship is only affected by vegetation change. Zhang et al. (2011) applied the TTM and SBM to 15 catchments in Australia and demonstrated that this two methods yielded similar estimates with differences smaller than 25%.

However, the Red Hill catchment (treated catchment for afforestation), which is located in the southeast Australia, has experienced multiyear drought. Based on the data from 1990-2009 including the Millennium Drought period, Zhao et al. (2010) showed that estimated contributions of afforestation to the decrease in runoff between pre- and post-change periods using the PCM is only 27%, which was even smaller than ½ of estimated contributions derived using the other two widely used methods (71% for the TTM

and 57% for the SBM). In addition to vegetation change, multiyear drought may also cause non-stationary change in the rainfall-runoff relationship, which may undermine prior assumptions of three widely used methods resulting inconsistency in their results. However, this question has not been explored and verified, and clarifying whether multiyear drought will have an important impact on the application ability of the three widely used methods will provide a meaningful reference for ecological engineering under changing climate with frequent extremes in future. If this hypothesis mentioned above is demonstrated to be correct, it will require us to propose a new method to solve this problem. Red Hill paired experiments provide a very good case study to investigate this issue. The primary objectives of this study are: (1) to detect whether multiyear drought has induced non-stationarity in the rainfall-runoff relationships of the Red Hill paired experimental catchments; (2) to test whether multiyear drought undermine prior assumptions of three widely used methods and is the reason for inconsistency amongst three widely used methods; and (3) to develop a new framework for separating the effects of vegetation change and other influencing factors on runoff under non-stationary conditions.

## 2 Paired Catchments and Data

The Red Hill catchment (1.95 km$^2$) and the Kileys Run catchment (1.35 km$^2$) were paired catchments located 23 km northeast of Tumut and 100 km west of Canberra in New South Wales, Australia (35.322ºS, 149.137ºE) (Fig. 1). The catchments are adjacent, and the soil texture, topographic characteristics, and climatic conditions are similar. The altitude of the two catchments ranges from 590 m to 835 m above sea level. The slope in the lower part of catchments is mostly gentle, and gradually increases towards the ridge in a convex form. Geology of the Red Hill catchment is predominately Young granodiorite, while the Kileys Run catchment is dominated by Alkali diorite. Valley floor, midslope yellow duplex, shallow red soils and upslope red duplex are four main soil types in these two catchments. Upslope red duplex soils has the highest saturated hydraulic conductivities and valley floor soils has the lowest saturated hydraulic conductivities (Major et al., 1998). The climate of these two catchments is temperate with highly variable and winter-dominated rainfall. In 1988, *P. radiata* was planted in the Red Hill catchment (0.5 km$^2$), and the remainder (1.45 km$^2$) was planted in April 1989. By 1997, pine occupied 78% of the Red Hill catchment. During multiyear drought period, no trees died in the treated catchment (Bren et al., 2006). The neighboring catchment (Kileys Run) was the control catchment, which has been maintained as a grazed pasture control over the entire observation period (Webb and Kathuria, 2012).

Daily rainfall and runoff from these two catchments were collected during the period of 1990–2015. The daily rainfall was measured by tipping bucket rain gauges had been located at catchment outlet and the daily runoff was measured by a flat-v style crump weir at a gauging station at the outlet of each catchment (Major et al., 1998). Mean annual rainfall and mean annual runoff of the Red Hill catchment were 817 mm and 75 mm, respectively, during the study period. Mean annual rainfall and runoff were 817 mm and 161 mm, respectively, in the Kileys Run catchment over the same period. Monthly potential evapotranspiration (PET) records were obtained from the SILO Data ([www.longpaddock.qld.gov.au/silo/point-data/](www.longpaddock.qld.gov.au/silo/point-data/)). The daily data were only used for the analysis of flow duration curves (FDCs). The monthly data were used for the paired-catchment method (PCM), time-trend method (TTM), the new framework and the analysis of double mass curves (DMCs). The annual data are used in the sensitivity-based method (SBM). Figure 2 shows the change of rainfall anomaly (%) in the Kileys Run and Red Hill catchments. Rainfall anomaly (%) is defined as the percentage deviation of annual rainfall to mean manual rainfall. It can be seen that three-year moving average of the rainfall anomaly (the black line) is lower than zero from 2000 to 2009. According to the method of determining multiyear drought period proposed by Saft et al. (2015), two catchments experienced prolonged drought lasted 10 years from 2000 to 2009 and this coincided with the period of the Millennium Drought of Australia (Peterson et al., 2021; van Dijk et al., 2013). The minimum measured annual rainfall from 1990 to 2015 were 388.6 mm.

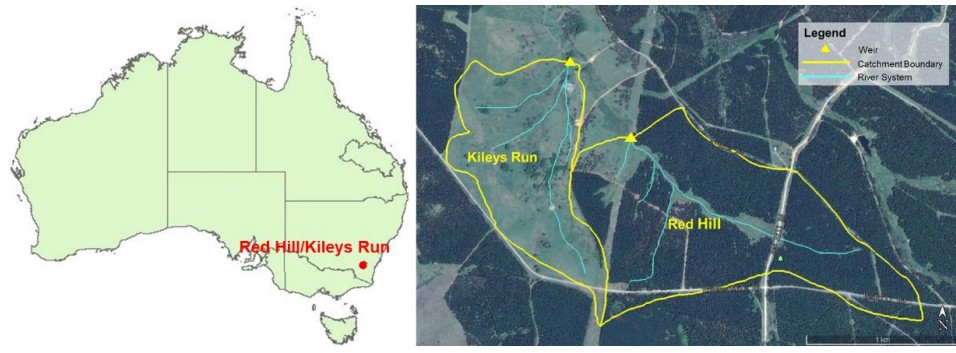

**Figure 1: Location and satellite remote sensing image map of the Red Hill/Kileys Run catchments in New South Wales, Australia (© Google Earth).**

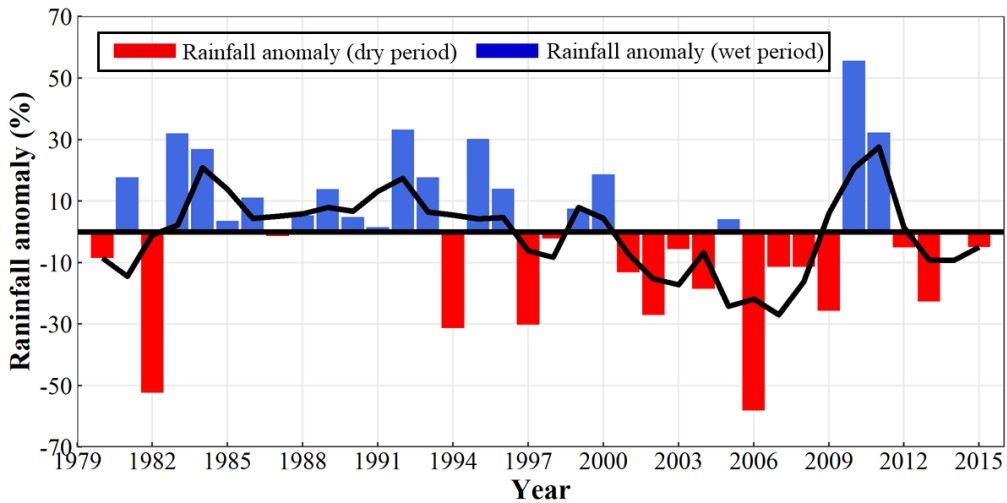

**Figure 2: Rainfall anomaly (%) as a percentage of the mean annual rainfall of the Kileys Run and Red Hill catchments, New South Wales, Australia. Red bars represent dry years and blue bars represent wet years. The black line represents the three-year moving average of the rainfall anomaly.**

## 3 Methods

### 3.1 Detecting non-stationarity in the rainfall-runoff relationship

The Mann-Kendall test (Kendall, 1975; Mann, 1945) was used to detect the long-term trend of annual rainfall, runoff, and potential evapotranspiration and the Sen's slope estimator (Sen, 1968) was used to obtain the degrees of above changes. If the $Z$ value estimated by the Mann-Kendall test method is less than zero, it indicates a downward trend; on the contrary, if the $Z$ value is greater than zero, it indicates an upward trend. $\beta$ estimated by the Sen's method represents the slope of the change trend. The Pettitt method (Pettitt, 1979) is a rank-based nonparametric statistical test method and is used to detect abrupt change points of annual rainfall, runoff, potential evapotranspiration records and the rainfall and runoff cumulative curves. The abrupt change point of annual runoff is used to divide the calibration period and the prediction period. The Mann-Kendall and Pettitt methods are the most frequently used statistical methods for identification of changes in hydro-meteorological data (Peng et al., 2020).

Double mass curves (DMCs), flow duration curves (FDCs), and rainfall-runoff linear regression curves were employed to detect changes in the rainfall-runoff relationship caused by vegetation change and multiyear drought. The DMCs plot the accumulated values of one variable against the accumulated values of another related variable for a concurrent period (Searcy and Hardison,

1960; Wang et al., 2013). It can still appear as a straight line when both hydrometeorological variables (rainfall and runoff) equilibrate quickly, or at the same rate under the condition of stationary changes. A break in the slope of the DMCs detected by the Pettitt method means that a change in the constant of proportionality between rainfall and runoff has occurred. The difference in the slope of the lines indicates the shift in the rainfall-runoff relationship and the degree of change in the relation. The FDCs

represent the relationship between magnitude and frequency of runoff, providing thus an important synthesis of the relevant hydrological processes occurring at the catchment scale (Pumo et al., 2013) and apparent change in the shape of the FDCs indicates the change in the rainfall-runoff relationship. Moreover, the upward or downward changes in rainfall-runoff linear regression curves also can detect non-stationary rainfall-runoff relationship (Liu et al., 2021).

### 3.2 Traditional methods for quantifying the effects of vegetation change on runoff

For a given catchment, the change in mean annual runoff between two periods can be estimated as:

$$\overline{\Delta Q_t^{total}} = \overline{Q_{t2}^{obs}} - \overline{Q_{t1}^{obs}} \tag{1}$$

where $\overline{\Delta Q_t^{total}}$ represents the total change in mean annual runoff, $\overline{Q_{t1}^{obs}}$ is the average annual runoff during the first period, and $\overline{Q_{t2}^{obs}}$ is the average annual runoff during the second period. In paired-catchment studies, the first period and the second period are usually defined as the calibration period (or pre-treatment period) and the prediction period (or post-treatment period), respectively.

The total runoff change can be considered to result from vegetation change ($\overline{\Delta Q_t^{veg}}$), climate variability ($\overline{\Delta Q_t^{clim}}$), and hydroclimatic

non-stationarity ($\overline{\Delta Q_t^{n}}$). Hydroclimatic non-stationarity can be caused by multiyear drought or other factors. Hence one can write:

$$\overline{\Delta Q_t^{total}} = \overline{\Delta Q_t^{veg}} + \overline{\Delta Q_t^{clim}} + \overline{\Delta Q_t^{n}} \tag{2}$$

Equation (2) has three unknowns and additional relationships are required to attribute the total runoff change to the respective three causes.

### 3.2.1 Paired-catchment method (PCM)

The PCM assumes that the correlation between runoff in two paired catchments will remain the same if the vegetation cover

remains the same or changes in a similar fashion. This correlation is established by regression analysis during the calibration period, and then is used to predict the runoff for the treated catchment during the prediction period. The difference between the measured and predicted runoff of the treated catchment during the prediction period represents the impact of the vegetation treatment (e.g., afforestation, deforestation) on runoff (Bosch and Hewlett, 1982; Lee, 1980; Stoneman, 1993; Williamson et al., 1987):

During the calibration period:

$$Q_{t1}^{obs} = a_1 Q_{c1}^{obs} + b_1 \tag{3}$$

During the prediction period:

$$Q_{t2}^{sim} = a_1 Q_{c2}^{obs} + b_1 \tag{4}$$

$$\overline{\Delta Q_t^{veg1}} = \overline{Q_{t2}^{obs}} - \overline{Q_{t2}^{sim}} \tag{5}$$

where $Q_t^{obs}$ and $Q_c^{obs}$ represent measured runoff from the treated and control catchments, respectively; $Q_{t2}^{sim}$ is the predicted runoff for the treated catchment; subscripts 1 and 2 represent the calibration period and the prediction period; and $a_1$ and $b_1$ are the fitted regression coefficients; $\overline{\Delta Q_t^{veg1}}$ is the change in mean annual runoff caused by vegetation change estimated by the PCM. The difference between total change ($\overline{\Delta Q_t^{total}}$) and $\overline{\Delta Q_t^{veg1}}$ of the treated catchment represents the combined effect of climate variability and hydroclimatic non-stationarity (i.e. $\overline{\Delta Q_t^{clim}} + \overline{\Delta Q_t^{n}}$).

### 3.2.2 Time-trend method (TTM)

The TTM can be applied to a single catchment that experienced vegetation change during two different periods. Runoff without vegetation change can be simulated by using the rainfall-runoff relationship that was developed over the calibration period (Lee, 1980):

During the calibration period:

$$Q_{t1}^{obs} = a_2 P_{t1}^{obs} + b_2 \tag{6}$$

During the prediction period:

$$Q_{t2}^{sim} = a_2 P_{t2}^{obs} + b_2 \tag{7}$$

where $P$ is precipitation; $a_2$ and $b_2$ are the fitted regression coefficients.

When the rainfall-runoff relationship of the treated catchment is not subject to hydroclimatic non-stationarity, the third term of Eq. (2) (i.e., $\overline{\Delta Q_t^{n}}$) can be ignored and hence the effect of vegetation change on runoff can be estimated as:

$$\overline{\Delta Q_t^{veg2}} = \overline{Q_{t2}^{obs}} - \overline{Q_{t2}^{sim}} \tag{8}$$

where $\overline{\Delta Q_t^{veg2}}$ is the change in mean annual runoff caused by vegetation change estimated by the TTM, $Q_{t1}^{obs}$ and $Q_{t2}^{sim}$ are the same as defined above.

### 3.2.3 Sensitivity-based method (SBM)

The SBM is widely used to directly estimate runoff change caused by climate variability. Runoff change caused by vegetation change can be estimated by subtracting the runoff change caused by climate variability from the total runoff changes. Runoff change caused by climate variability can be determined by changes in precipitation and potential evapotranspiration (Koster and Suarez, 1999; Milly and Dunne, 2002), expressed as:

$$\overline{\Delta Q_t^{clim}} = \beta \overline{\Delta P_t^{obs}} + \gamma \overline{\Delta PET_t^{obs}} \tag{9}$$

where $\overline{\Delta Q_t^{clim}}$ is change in mean annual runoff caused by climate variability; $\Delta P$, and $\Delta PET$ are changes in precipitation ($P$) and potential evapotranspiration (PET), respectively; $\beta$ and $\gamma$ are the sensitivity coefficients of runoff to precipitation and potential evapotranspiration, respectively, as estimated in Li et al. (2007) as:

$$\beta = \frac{1 + 2x + 3wx^2}{(1 + x + wx^2)^2} \tag{10}$$

$$\gamma = -\frac{1 + 2wx}{(1 + x + wx^2)^2} \tag{11}$$

where $x$ is the mean annual dryness index (estimated as $PET/P$) and $w$ is a fitted model parameter related to catchment conditions such as vegetation type, soil, and PET. $w$ was set as 1.66 for the Red Hill catchment in this study according to Zhao et al. (2010).

When the rainfall-runoff relationship of the treated catchment is not subject to hydroclimatic non-stationarity, the third term of Eq. (2) can be ignored. Runoff change caused by vegetation change can be estimated by subtracting the runoff change caused by climate variability from the total runoff changes.

$$\overline{\Delta Q_t^{veg3}} = \overline{Q_t^{total}} - \overline{Q_t^{clim}} \tag{12}$$

where $\overline{\Delta Q_t^{veg3}}$ is the change in mean annual runoff caused by vegetation change estimated by the SBM, $\overline{Q_t^{total}}$ and $\overline{Q_t^{clim}}$ are the same as defined above.

The calibration and prediction periods for paired-catchment studies are usually defined by the vegetation change history. However, calibration period data were absent for the Red Hill and Kileys Run catchments because runoff observations started only about one year before the treatment. Therefore, the calibration period and the prediction period were taken as the pre-change period and post-change periods of runoff, respectively, as determined by the step change-point in the runoff of the treated catchment as previous studies on this site. This approximation will have little effect on the results as previous studies have shown that the establishment of the young pine tree plantation at the Red Hill catchment had very limited impacts on runoff in the first several years of establishment (Zhao et al., 2010).

### 3.3 Proposed new framework for quantifying the effects of vegetation change on runoff under non-stationary conditions

The three methods have been successfully applied to paired-catchment studies to estimate the effect of vegetation change on runoff and there is little difference amongst $\overline{\Delta Q_t^{veg1}}$, $\overline{\Delta Q_t^{veg2}}$ and $\overline{\Delta Q_t^{veg3}}$ in catchments that did not experienced hydroclimatic non-stationarity (Zhao et al., 2010). However, when both catchments (i.e., the control and treated catchments) experienced hydroclimatic non-stationarity, the use of the time-trend method (TTM) becomes problematic as the rainfall-runoff relationship represented by Eq. (7) does not capture this non-stationarity effect because it is based on the assumption that the rainfall-runoff relationships are stationary with respect to hydroclimatic conditions. The sensitivity-based method (SBM) also has similar problems in quantifying the effect of vegetation change. In the case of non-stationarity induced by multiyear drought, the TTM and SBM will overestimate the effect of vegetation change on runoff and the results of this two methods are actually the combined effect of vegetation change and hydroclimatic non-stationarity (i.e. $\overline{\Delta Q_t^{veg}} + \overline{\Delta Q_t^n}$). The basic concept of the paired-catchment method (PCM) is to compare the streamflow of two nearby catchments with similar physical characteristics, one being a control and the other being a treated catchment. The PCM assumes that the control and treated catchments would experience the same conditions or changes except the treatment implemented. To a first approximation, the PCM should provide accurate estimates of the effect of vegetation change even under hydroclimatic non-stationarity because the PCM assumes that the control and treated catchments would experience the same conditions or changes except the treatment of interest.

In previous studies on the Red Hill paired catchment site, the third term ($\overline{\Delta Q_t^n}$) in Eq. (2) was ignored, or the second term ($\overline{\Delta Q_t^{clim}}$) and the third term ($\overline{\Delta Q_t^n}$) in Eq. (2) were taken as a whole without being separated when the hydroclimatic non-stationarity

happened. We proposed a new framework for quantifying the effects of vegetation change on runoff under non-stationary hydroclimatic conditions. The new framework considers three factors that affect runoff: vegetation change, climate variability and hydroclimatic non-stationarity, respectively. For a treated catchment, one can assume that the runoff reduction ($\overline{\Delta Q_t^{total}}$) is caused by vegetation change ($\overline{\Delta Q_t^{veg}}$), climate variability ($\overline{\Delta Q_t^{clim}}$); and hydroclimatic non-stationarity ($\overline{\Delta Q_t^n}$). (It is assumed that climate variability does not change the rainfall-runoff relationship. That is to say, climate variability does not alter runoff ratio (or slope between accumulated annual rainfall and accumulated annual runoff) and runoff sensitivity to $P$ and PET. For the control catchment, the runoff reduction ($\overline{\Delta Q_c^{total}}$) is mainly caused by climate variability ($\overline{\Delta Q_c^{clim}}$) and multiyear drought ($\overline{\Delta Q_c^n}$). The principle of the new framework is shown in Fig. 3.

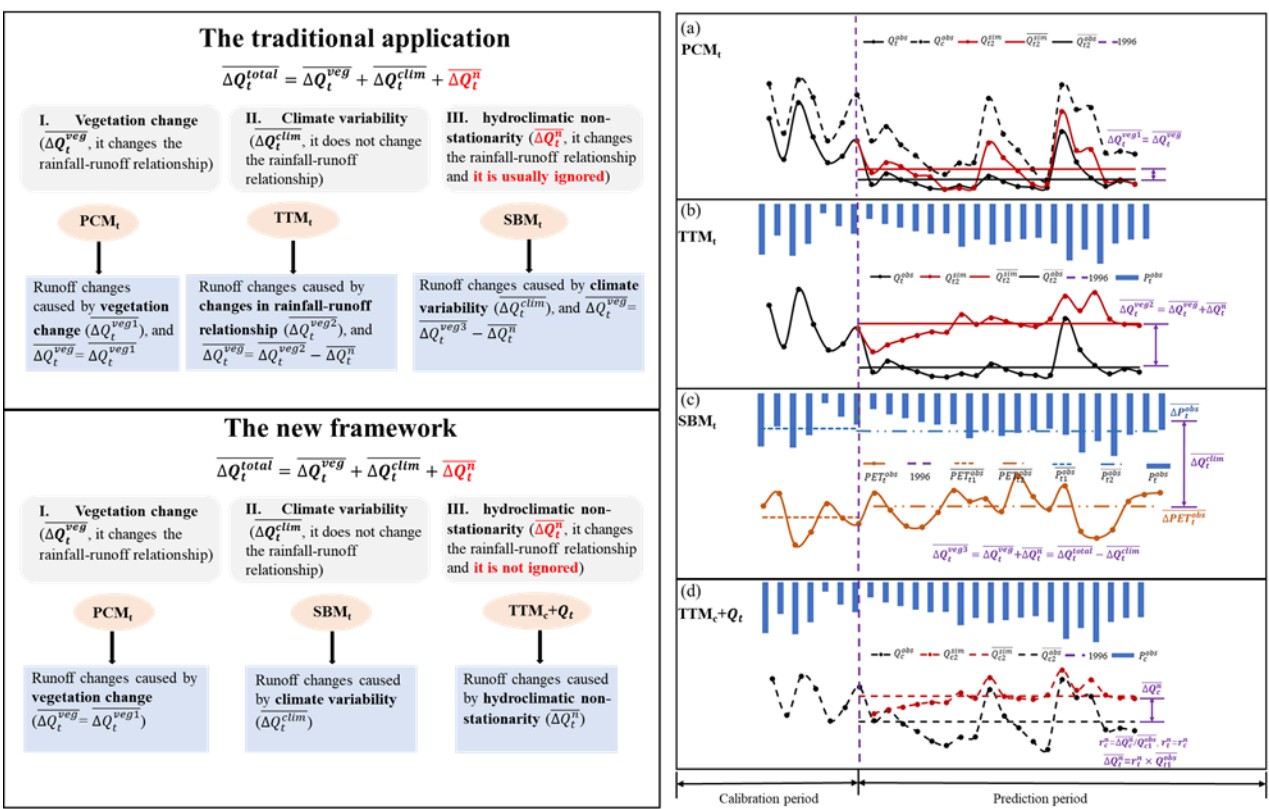

**Figure 3: Schematic diagram showing principles of the traditional application and the new framework. PCM means paired-catchment method, TTM means time-trend method, SBM means sensitivity-based method, and subscripts *c* and *t* represent the control catchment and the treated catchment. (a) the PCM for estimating runoff change caused by vegetation change, (b) the TTM for estimating runoff change caused by vegetation change and hydroclimatic non-stationarity, (c) the SBM for estimating runoff change caused by climate variability and (d) the TTM used in the control catchment and runoff of the treated catchment for estimating runoff change caused by hydroclimatic non-stationarity.**

### 3.3.1 Separating the effects of hydroclimatic non-stationarity on runoff

The control catchment is only affected by climate variability and hydroclimatic non-stationarity and the impact of hydroclimatic non-stationarity on runoff can be estimated by the TTM and the runoff and rainfall data. In view of the similarity of the attributes of the control and treated catchments, the impact of hydroclimatic non-stationarity on runoff of the treated catchment can be indirectly abtained by the control catchment and the runoff data.

$$Q_{c1}^{obs} = a_3 P_{c1}^{obs} + b_3 \tag{13}$$

where $Q_{c1}^{obs}$ and $P_{c1}^{obs}$ represent measured runoff and rainfall from the control catchment during the calibration period, respectively; $a_3$ and $b_3$ are the fitted regression coefficients.

The simulated runoff not affected by hydroclimatic non-stationarity during the prediction period can be obtained by Eq. (14), while the runoff change caused by hydroclimatic non-stationarity ($\overline{\Delta Q_c^n}$) can be obtained by Eq. (15).

$$Q_{c2}^{sim} = a_3 P_{c2}^{obs} + b_3 \tag{14}$$

$$\overline{\Delta Q_c^t} = \overline{Q_{c2}^{obs}} - \overline{Q_{c2}^{sim}} \tag{15}$$

where $Q_{c2}^{obs}$ and $P_{c2}^{obs}$ represent measured runoff and rainfall from the control catchment during the prediction period, respectively; $Q_{c2}^{sim}$ is the predicted runoff for the control catchment.

The percentage runoff reduction ($r_c^n$) caused by multiyear drought in the control catchment can be estimated as:

$$r_c^n = \overline{\Delta Q_c^n} / \overline{Q_{c1}^{obs}} \tag{16}$$

Assumed that the percentage of runoff reduction caused by hydroclimatic non-stationarity is the same for both control and treated catchments (i.e., $r_t^n = r_c^n$) because they have similar physical characteristics. Runoff reduction caused by hydroclimatic non-stationarity in the treated catchment ($\overline{\Delta Q_t^n}$):

$$\overline{\Delta Q_t^n} = r_t^n \times \overline{Q_{t1}^{obs}} \tag{17}$$

### 3.3.2 Separating the effects of vegetation change on runoff

For the PCM, the actual effects of vegetation change on runoff ($\overline{\Delta Q_t^{veg}}$) is equal to $\overline{\Delta Q_t^{veg1}}$. For the TTM, the actual effects of vegetation change on runoff ($\overline{\Delta Q_t^{veg}}$) is equal to $\overline{\Delta Q_t^{veg2}} - \overline{\Delta Q_t^n}$. For the SBM, the actual effects of vegetation change on runoff ($\overline{\Delta Q_t^{veg}}$) is equal to $\overline{\Delta Q_t^{veg3}} - \overline{\Delta Q_t^n}$. The difference amongst $\overline{\Delta Q_t^{veg1}}$, $\overline{\Delta Q_t^{veg2}} - \overline{\Delta Q_t^n}$ and $\overline{\Delta Q_t^{veg3}} - \overline{\Delta Q_t^n}$ should be small.

### 3.3.3 Separating the effects of climatic variability on runoff

For the PCM, the actual effects of climate variability on runoff ($\overline{\Delta Q_t^{clim}}$) is equal to $\overline{\Delta Q_t^{total}} - \overline{\Delta Q_t^{veg1}} - \overline{\Delta Q_t^n}$. For the TTM, the actual effects of climate variability on runoff ($\overline{\Delta Q_t^{clim}}$) is equal to $\overline{\Delta Q_t^{total}} - \overline{\Delta Q_t^{veg2}}$. For the SBM, the actual effects of climate variability on runoff ($\overline{\Delta Q_t^{clim}}$) is equal to the result of Eq. (9). The difference amongst $\overline{\Delta Q_t^{total}} - \overline{\Delta Q_t^{veg1}} - \overline{\Delta Q_t^n}$, $\overline{\Delta Q_t^{total}} - \overline{\Delta Q_t^{veg2}}$ and the result of Eq. (9) should be small.

### 3.3.4 The contribution of climate variability, vegetation change and hydroclimatic non-stationarity to runoff reduction

The percentage contribution of vegetation change, climate variability and hydroclimatic non-stationarity to total runoff reduction can be estimated as:

$$p_t^{veg} = \overline{\Delta Q_t^{veg}} / \overline{\Delta Q_t^{total}} \tag{18}$$

$$p_t^{clim} = \overline{\Delta Q_t^{clim}} / \overline{\Delta Q_t^{total}} \tag{19}$$

$$p_t^n = \overline{\Delta Q_t^n} / \overline{\Delta Q_t^{total}} \qquad\qquad (20)$$

## 4 Results

### 4.1 Detecting non-stationarity in the rainfall-runoff relationships of control and treated catchments

The double mass curves (DMCs) of monthly rainfall and runoff of the two paired catchments are shown in Fig. 4 (a) and Fig. 4 (b). The cumulative rainfall-runoff relationship of the two catchments changed significantly twice as seen in the slope changes of the regressions applied to the DMCs data. Two change points estimated by the Pettitt method occurred in December 1996 and January 2010 in the Red Hill catchment, and in October 2001 and May 2010 in the Kileys Run catchment. Thus, the entire study period can be divided into three periods in the two catchments, i.e. the first period (January 1990 to December 1996 in the Red

Hill catchment and January 1990 to October 2001 in the Kileys Run catchment), the second period (January 1997 to December 2009 in the Red Hill catchment and November 2001 to May 2010 in the Kileys Run catchment), and the third period (January 2010 to December 2015 in the Red Hill catchment and June 2010 to December 2015 in the Kileys Run catchment).

Figure 4 (a) and Figure 4 (b) shows that the slopes and intercepts of the DMCs regressions of the two catchments in the different periods were quite different. The slopes of the linear regression lines in the first, second, and third periods were 0.27, 0.11, and

0.19 in the Kileys Run catchment, respectively. And the slopes were 0.21, 0.02, and 0.06 in the Red Hill catchment, respectively. Runoff of the two catchments both experienced a large reduction during the second period (i.e., the period of multiyear drought) and then slightly increased during the third period (i.e., the post-drought period), but still well below the runoff of the first period. And the decrease of runoff or the change of the rainfall-runoff relationship in the second period of the Red Hill catchment was much higher than that of the Kileys Run catchment, suggesting that the Red Hill catchment was affected by both vegetation change

and multiyear drought, while the Kileys Run catchment was only affected by multiyear drought. It showed that the rainfall-runoff relationships of the two catchments became non-stationary during and after multiyear drought.

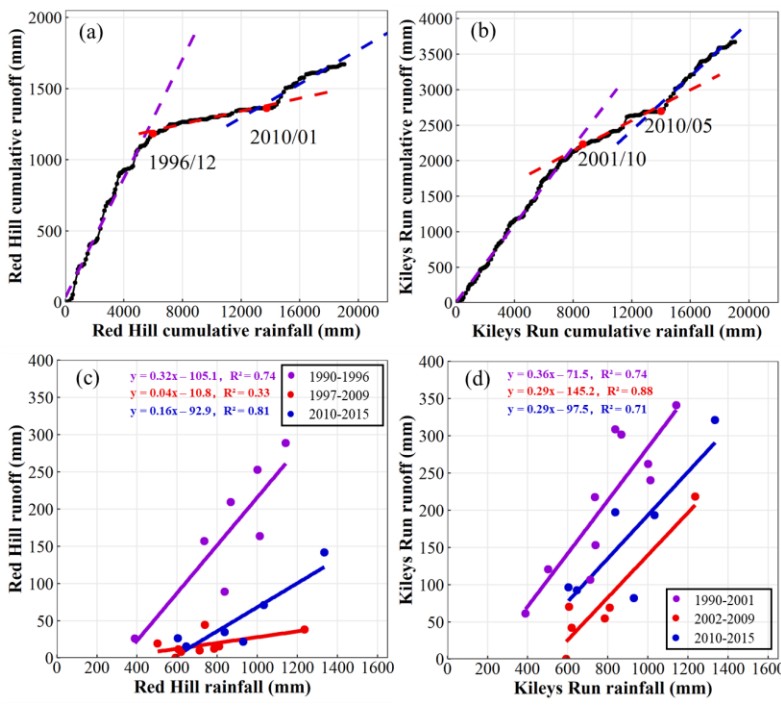

**Figure 4: (a) Double mass curve of monthly rainfall and runoff of the Red Hill catchment (treated catchment), (b) Double mass curve of monthly rainfall and runoff of the Kileys Run catchment (control catchment), (c) Relationships between annual rainfall and runoff of the Red Hill catchment (treated catchment), (d) Relationships between annual rainfall and runoff of the Kileys Run catchment (control catchment). The dashed lines in (a) and (b) represent the linear regression lines between cumulative rainfall and cumulative runoff during three different periods (January 1990 to December 1996 (purple), January 1997 to January 2010 (red), and February 2010 to December 2015 (blue) in Red Hill; January 1990 to October 2001 (purple), November 2001 to May 2010 (red), and June 2010 to December 2015 (blue) in Kileys Run). The purple, red, and blue lines in (c) and (d) represent the linear regression lines for three different periods (1990–1996, 1997–2009, and 2010–2015 in Red Hill; 1990–2001, 2002–2009, and 2010–2015 in Kileys Run).**

The linear regression lines defining the relationship between annual rainfall and runoff for the periods of 1990–1996, 1997–2009, and 2010–2015 in the Red Hill catchment and the periods of 1990–2001, 2002–2010, and 2011–2015 in the Kileys Run catchment are shown in Fig.4 (c) and Fig. 4 (d). The differences in the slope and intercept of the Red Hill catchment were −0.28 and 94.3 mm, respectively, between the second and first period, indicating a significant reduction in runoff and a great change in the rainfall-runoff relationship because of afforestation and multiyear drought during the second period. The runoff coefficient decreased by 87.8% and 63.3% during the drought period in the Red Hill and Kileys Run catchments, respectively. Runoff of the Red Hill catchment partially recovered during the third period, as shown in Fig. 4 (c). The intercept and slope of the Kileys Run catchment had similar changes, as shown in Fig. 4 (d). These results suggested that the rainfall-runoff relationship of the two catchments experienced considerable change during and after multiyear drought in the second period.

The daily flow duration curves (FDCs) of the two catchments in three different periods (same as periods for DMCs analysis) are shown in Fig. 5. Zero flows were not observed during the first period (before the drought period), but they were observed in 14% and 8% of the times during the second and third periods (i.e., the multiyear drought period and the post-drought period), respectively in the Kileys Run catchment. But in the Red Hill catchment, zero flows were observed in 3%, 70% and 59% of the times during the three periods, respectively. The FDCs during the first period (purple line) were flatter and smoother than the lines for the other two periods, indicating that runoff changes before the multiyear drought period or runoff reached a new equilibrium state were relatively stable and had a stationary relationship with rainfall. However, for most percentages of the FDCs during the second period (red line), runoff decreased by more than 50%. Especially low flow decreased most rapidly, and there were 14% and 70% no-flow days. Runoff during the third period (blue line) increased compared with the second period. Especially in the high flow region, daily flow recovered to more than 50% of the runoff that occurred before the multiyear drought, but the low flow increased relatively less, and there were also 8% and 59% no-flow days. In summary, the shape and percentage of the zero flows of FDCs in Fig. 5 further demonstrated that the relationship between rainfall and runoff of the two catchments changed significantly over the three periods, especially for the Red Hill catchment suffered from both multiyear drought and afforestation.

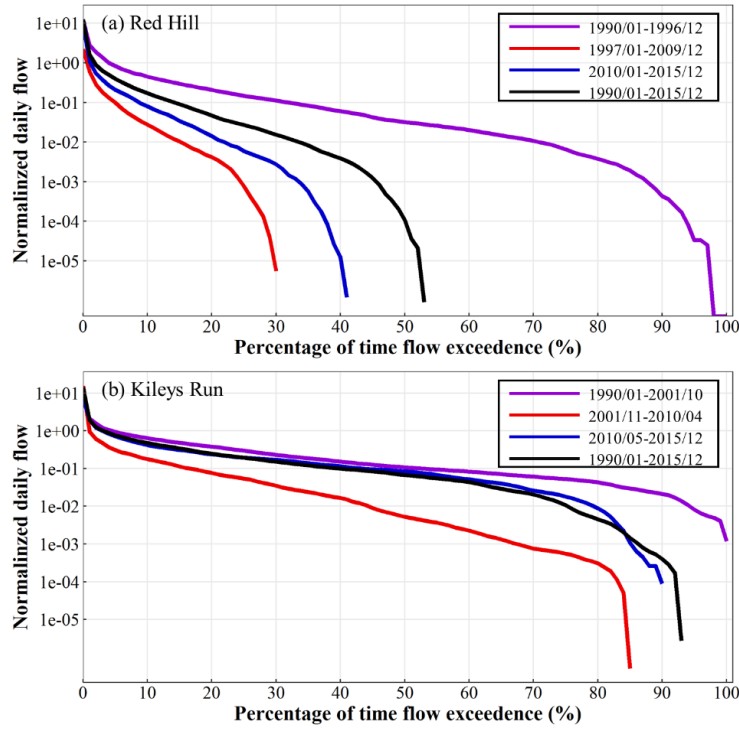

**Figure 5: Daily flow duration curves of (a) the Red Hill catchment (treated catchment) and (b) the Kileys Run catchment (control catchment), New South Wales, Australia, over three different periods.**

### 4.2 Separated effects of vegetation change using three traditional methods

The statistical information of the trends and change points in annual runoff, rainfall, and PET of both catchments based on observed

data from 1990 to 2015 are shown in Table 1. The change point in annual runoff in the Red Hill catchment occurred in 1996 and annual runoff decreased significantly after 1996 ($\beta=-5.3$, $p<0.05$). Annual runoff in the Kileys Run catchment also decreased, but the reduction was not significant ($\beta=-8.1$, $0.05<p\leq0.1$). Annual rainfall and PET of two catchments decreased and increased respectively ($\beta=-3.4$, $\beta=3.5$, $p>0.1$). Thus, the calibration period was set as 1990–1996 and the prediction period was set as 1997–2015.

**Table 1: Estimated trends and abrupt change points in annual runoff ($Q$), precipitation ($P$), and potential evapotranspiration (PET) of the Red Hill and Kileys Run catchments, New South Wales, Australia, during the period of 1990–2015.**

| Catchment | $Q$ | | | $P$ | | | PET | | |
|---|---|---|---|---|---|---|---|---|---|
| | $Z$ | $\beta$ (mm yr$^{-1}$) | Year[a] | $Z$ | $\beta$ (mm yr$^{-1}$) | Year[a] | $Z$ | $\beta$ (mm yr$^{-1}$) | Year[a] |
| Kileys Run | −1.9 | −8.1* | 1996 | −0.3 | −3.4 | 1993 | 1.1 | 3.5 | 2001 |
| Red Hill | −2.4 | −5.3** | 1996* | −0.3 | −3.4 | 1993 | 1.1 | 3.5 | 2001 |

*Note.* *** represents *p-value*≤0.01, ** represents 0.01<*p-value*≤0.05, * represents 0.05<*p-value*≤0.1. [a]the change point year estimated by the Pettitt method.

The $R^2$ values of the monthly runoff-runoff relationship and the monthly rainfall-runoff relationship were 0.82 and 0.52, respectively. The linear relationships were $Q_{RH}=0.87\times Q_{KR}-3.9$ (where $Q_{RH}$ is monthly runoff of the Red Hill catchment, $Q_{KR}$ is monthly runoff of the Kileys Run catchment), and $Q_{RH}=0.28\times P_{RH}-6.0$ (where $P_{RH}$ is monthly rainfall of the Red Hill catchment).

These results indicate a good relationship between monthly runoff at these two catchments during the calibration period. Therefore, the relationships can be used to predict runoff of the Red Hill catchment during the prediction period and to estimate runoff change caused by vegetation change.

**Table 2: Total runoff reduction ($\overline{\Delta Q_t^{total}}$) and runoff reduction caused by vegetation change, climate variability and hydroclimatic non-stationarity ($\overline{\Delta Q_t^{veg}}, \overline{\Delta Q_t^{clim}}, \overline{\Delta Q_t^n}$) of the Red Hill catchment, New South Wales, Australia, estimated using the traditional methods and the new framework. The bold black numbers represent results that can be calculated directly from the observation data. The bold italic black numbers are final results that further calculated by the bold black numbers.**

| (mm) | | Paired-catchment method | Time-trend method | Sensitivity-based method |
|---|---|---|---|---|
| $\overline{\Delta Q_t^{total}}$ | | 138.1 | 138.1 | 138.1 |
| A. Traditional methods | $\overline{\Delta Q_t^{veg}}$ | **45.3** | **129.1** | 138.1-33.0=***105.1*** |
| | $\overline{\Delta Q_t^{clim}}$ | 138.1-45.3=***92.8*** | 138.1-129.1=***9*** | **33.0** |
| B. New framework | $\overline{\Delta Q_t^{veg}}$ | **45.3** | 129.1-75.5=***53.6*** | 138.1-33.0-75.5=***29.6*** |
| | $\overline{\Delta Q_t^{clim}}$ | 138.1-45.3-75.5=***17.3*** | 138.1-129.1=***9.0*** | **33.0** |
| | $\overline{\Delta Q_t^n}$ | **75.5** | **75.5** | **75.5** |
| | $\overline{\Delta Q_t^{veg}} + \overline{\Delta Q_t^n}$ | 45.3+75.5=***120.8*** | **129.1** | 138.1-33.0=***105.1*** |
| | $\overline{\Delta Q_t^{veg}} + \overline{\Delta Q_t^n} + \overline{\Delta Q_t^{clim}}$ | 45.3+75.5+33.0=***153.8*** | ***153.8*** | ***153.8*** |

Estimated runoff change caused by vegetation change in the Red Hill catchment using the traditional three methods with 26 years of data are shown in Table 2. The total runoff change was −138.1 mm between the prediction and calibration period. By using the paired-catchment method (PCM), time-trend method (TTM), and sensitivity-based method (SBM), the estimated runoff changes due to vegetation change were −45.3 mm, −129.1 mm, and −105.1 mm, respectively, percentage change of 32.8%, 93.5%, and 76.1%. Clearly, the contribution of vegetation change to the changes in total runoff estimated by the three methods were still quite different. The decrease in runoff caused by vegetation change estimated by the PCM was much lower than that calculated by the other two methods. This inconstancy amongst the three methods was the same as described by Zhao et al. (2010) although a much longer observation period was used in this study.

### 4.3 Separated effects of vegetation change using the new framework

The results presented in section 4.2 demonstrated that the rainfall-runoff relationship of the control catchment (Kileys Run) was altered by multiyear drought and the rainfall-runoff relationship of the treated catchment (Red Hill) was altered by both multiyear drought and afforestation. Based on the new framework, impacts of vegetation change on runoff of the Red Hill catchment were re-estimated using the three methods again, and the results are listed in Table 2. The percentage runoff reduction induced by multiyear drought ($r_t^n$) is −45%. For the Red Hill catchment, runoff changes in the rainfall-runoff relationship induced by vegetation change ($\overline{\Delta Q_t^{veg}}$) and multiyear drought ($\overline{\Delta Q_t^n}$) are −45.3 mm and −75.5 mm, respectively, and runoff change caused by climate variability ($\overline{\Delta Q_t^{clim}}$) calculated by the SBM is −33.0 mm. For the Kileys Run catchment, runoff change induced by multiyear drought ($\overline{\Delta Q_c^n}$) is −110.2 mm, and runoff change caused by climate variability ($\overline{\Delta Q_c^{clim}}$) by subtracting runoff change caused by multiyear drought from the total runoff changes ($\overline{\Delta Q_c^{total}}$=−126.4 mm) is −16.2 mm. Impacts of afforestation, multiyear drought and climate variability on runoff of the Red Hill catchment are 32.8%, 54.7% and 23.9%, respectively, and impacts of

350 multiyear drought and climate variability on runoff of the Kileys Run catchment are 87.2% and 12.8%, respectively. The sum of

the three terms $\overline{\Delta Q_t^{veg}}$, $\overline{\Delta Q_t^{n}}$ and $\overline{\Delta Q_t^{clim}}$ is $-153.8$ mm, which is close to the total runoff changes ($\overline{\Delta Q_t^{total}}=-138.1$ mm). Figure 6

shows the contribution of vegetation change to the total runoff changes estimated using the traditional methods and the new

framework. By considering the effects of multiyear drought on runoff of the treated catchment, apparent large differences amongst

the three methods no longer existed by using the new framework. Estimated impacts of afforestation on runoff decreased greatly

from 93.5% to 38.8% (=93.5%−54.7%) calculated by the TTM and decreased greatly from 76.1% to 21.4% (=76.1%−54.7%) by

the SBM. It shows that the new framework can better separate the impact of three factors on runoff.

Based on the above analysis, we found that multiyear drought changed the rainfall-runoff relationships of the control catchment

(Kileys Run) and the treated catchment (Red Hill). And differences among the three methods at the Red Hill experimental site

were still existed although a much longer observation period was used. The reason for the big difference is that the non-stationary

rainfall-runoff relationship of the treated catchment caused by multiyear drought that was neglected in the TTM and the SBM.

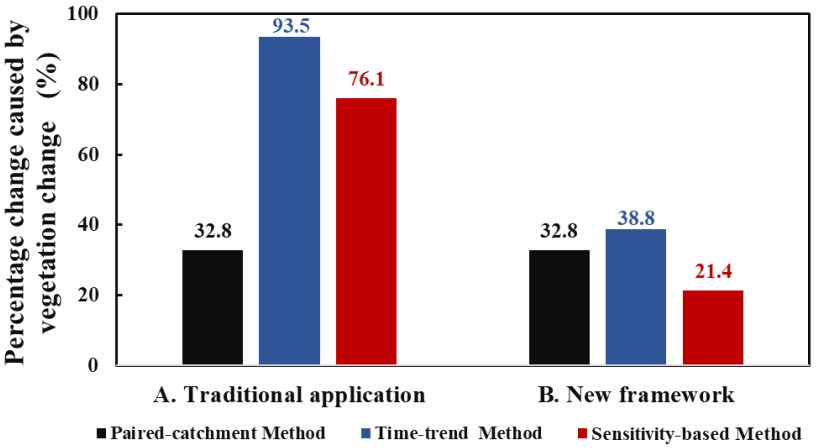

**Figure 6: The contribution of vegetation change to the total runoff changes of the Red Hill catchment, New South Wales, Australia, estimated using this three methods under the traditional application and the new framework.**

**5 Discussion**

**5.1 Differences in estimated impacts of vegetation change on runoff using three traditional methods**

The treated catchment (afforestation) experienced four different periods: (I) 1990–1996 pre-drought and untreated, (II) 1997–2001

pre-drought and treated, (III) 2002–2009 in-drought and treated, and (IV) 2010–2015 post-drought and treated. During the period

(I), runoff of the threated catchment has not been significantly affected, it can be considered as the calibration period for evaluating

the impact of vegetation change on runoff. During the period (II), the treated catchment was affected by both vegetation change

and climate variability. During the periods (III) and (IV), the treated catchment was affected by multiyear drought, vegetation

change and climate variability, because the rainfall-runoff relationship after multiyear drought may not recover to that before

multiyear drought (Fig. 4) yet and may persist such state for a long time (Peterson et al., 2021). When separating impacts of

vegetation change and multiyear drought on runoff, the data of the control and treated catchments need to be used at the same

period, that is to say, the same period needed to be applied to these two catchments. The (II), (III) and (IV) periods of the treated

catchment were combined into one period as the prediction period. Thus, Table 2 and Fig. 6 essentially compared runoff between

the untreated (1990–1996) and treated (1997–2015) periods. Runoff difference between the untreated and treated periods in the

treated catchment was caused by vegetation change, climate variability and multiyear drought, and runoff difference in the control catchment was caused by climate variability and multiyear drought. Comparing the results of the traditional application with the result of the new framework, the time-trend method (TTM) and the sensitivity-based method (SBM) significantly overestimate runoff reduction caused by vegetation change. The main reason for this difference is that runoff changes estimated by the TTM and the SBM are caused by the total non-stationary changes, and the non-stationary changes are caused by both vegetation change and multiyear drought in the Red Hill catchment. That is to say, both TTM and SBM significant overestimate the impact of vegetation change on mean annual runoff as both afforestation and multiyear drought induce runoff decrease.

Estimated changes between the period (I) (i.e., 1990–1996, pre-drought and untreated) and the period (II) (i.e., 1997–2001, pre-drought and treated) as well as between the period (I) (i.e., 1990–1996, pre-drought and untreated) and the period (III) & (IV) (i.e., 2002–2015, in- & post-drought and treated) can be seen in the supplementary Fig. S1. Impacts of afforestation on runoff were 34.3%, 65.9% and 41.5% of the total runoff changes during the period of 1997–2001 by the PCM, TTM and SBM, respectively. Impacts of afforestation on runoff were 32.4%, 100.8% and 68.4% of the total runoff changes during the period of 2002–2015. It can be seen that results of the TTM and SBM during the period of 2002–2015 were significantly higher than those during the period of 1997–2001, while the results of PCM were close. Because multiyear drought happened in 2002–2009 and caused persistent effects in 2010–2015 had a great impact on the rainfall-runoff relationship of the Red Hill catchment, which made the TTM and SBM overestimated the impact of vegetation change on runoff remarkedly. That is to say, errors of the impact of vegetation change on runoff estimated by the TTM and SBM will be larger as effects caused by multiyear drought are imposed on the paired catchments.

The response of runoff to vegetation change is estimated as −45.3 mm by the paired-catchment method (PCM) in this study and is relatively accurate. However, it is difficult for the PCM to further separate the effects of climate variability and multiyear drought on runoff because the total runoff changes minus runoff changes directly obtained by the PCM is the sum of runoff changes caused by climate variability and multiyear drought. At Red Hill experiment site, non-stationary changes of the treated catchment are caused by both vegetation change and multiyear drought, and stationary changes are caused by climate variability. Non-stationary changes of the control catchment are only caused by multiyear drought, and stationary changes are only caused by climate variability. According to the paradigm of PCM, shift in the rainfall-runoff relationship separated from the runoff correlation between the treated and control catchments should be caused only by the treatment of the treated catchment and effects of any other drivers can induce either stationary or non-stationary changes should be eliminated by making use of the control catchment. Therefore, the PCM is still the most reliable method compared with other methods and separated effect by the PCM is only caused by vegetation change (i.e., afforestation).

The TTM eliminates the influence of the stationary components by making use the rainfall-runoff relationship of the treated catchment during the calibration period. The result of the TTM, which is about 2.8 times greater than that of the PCM (see Table 2), are essentially runoff changes caused by the non-stationary changes in the rainfall-runoff relationship induced by both vegetation change and multiyear drought. Significant overestimation by the TTM is actually the effect of multiyear drought on runoff.

The SBM is sourced from the Budyko framework (Budyko, 1974). It assumes that steady state of catchment water balance is fundamentally determined by water input (represented by precipitation) and energy demand (represented by potential evapotranspiration) and transition from one steady state to another without any change in catchment properties should moving on

the Budyko curve (Roderick and Farquhar, 2011; Sun et al., 2014; Wang et al., 2021). Therefore, stationary changes driven by climate variability during post-treatment period can be separated by sensitivity of runoff to $P$ and PET established during the pre-treatment period. Figure 7 shows the change of annual $P$ and PET. Over the entire study period from 1990 to 2015, $P$ showed an insignificant decreasing trend of 3.4 mm year$^{-1}$ ($p$>0.1) and PET showed an insignificant increasing trend of 3.5 mm year$^{-1}$ ($p$>0.1). Both $P$ and PET decreased before 1996 and then increased after 1996. The rates of increase for annual $P$ and PET during 1997–2015 were 12.0 mm year$^{-1}$ and 2.6 mm year$^{-1}$, respectively, and the contributions of $P$ and PET to runoff changes caused by climate variability were −22.0 mm and −11.0 mm, respectively. The mean annual PET during the period of 1990–1996, 1997–2009, 2010–2015 and 1997–2015 were 1168 mm, 1262 mm, 1186 mm and 1238 mm, respectively. Compared with the period of 1990-1996 and 2010-2015, the mean annual PET during the period of 1997–2009 (the multiyear drought occurred) increased by 94 mm and 76 mm, respectively. Compared with the period of 1990–1996, the mean annual PET during the period of 1997–2015 increased by 70 mm. It was consistent with the cognition that afforestation and drought can make PET increase. The result estimated by the SBM is the impact of climate variability (without changing the catchment characters/non-stationary changes in the rainfall-runoff relationship) on runoff, that is, $\overline{\Delta Q_t^{clim}}$. It ignored the impact of multiyear drought on runoff, which has been demonstrated to cause non-stationary changes. Recent studies have also reported that multiyear drought can cause catchment properties changes and hydrological functionings (Kinal and Stoneman, 2012; Peterson et al., 2021; Saft et al., 2016; van Dijk et al., 2013), which may violate the assumptions of the SBM. Estimated change of the SBM, which is close to the result of the TTM (see Table 2) and is about 2.3 times greater than the result of the PCM, includes the non-stationary changes not only caused by vegetation change but also caused by multiyear drought.

There are missing values in both rainfall and runoff data. Both runoff and rainfall observations are missing from November 1999 to November 2000 and from October 2006 to October 2007. In order to minimize the influences of missing values on the annual total values, annual total is regarded as missing value if more than one month is missing. Thus, there are four missing data points in annual time series of rainfall (Fig.7). Two periods with missing data are just at the beginning and end of multiyear drought. Missing rainfall values should not differ significantly from the annual rainfall values during multiyear drought period. The overall trend or segmented trend during drought will not change much due to the lack of rainfall data. It is also true for annual runoff. In addition, the change point of annual runoff calculated with data including missing values was consistent with that by Zhao et al. (2010), both appeared in 1996. Based on data including missing data, estimated afforestation impacts were 31.4%, 84.7% and 64.9% of the total runoff changes during the period of 1990–2005 by the PCM, TTM and SBM, respectively. Results of Zhao et al. (2010) were 27.0%, 71.0% and 57% by the PCM, TTM and SBM, respectively. They were very close. Furthermore, same analysis was conducted based on the gridded rainfall data from SILO. Estimated afforestation impacts were 32.8%, 93.5% and 73.0% of the total runoff changes during the period of 1990–2015 by the PCM, TTM and SBM, respectively. The results were very close to results using in situ observed rainfall as presented in this study. Therefore, we believe processing of missing data has little influences on the estimated changes.

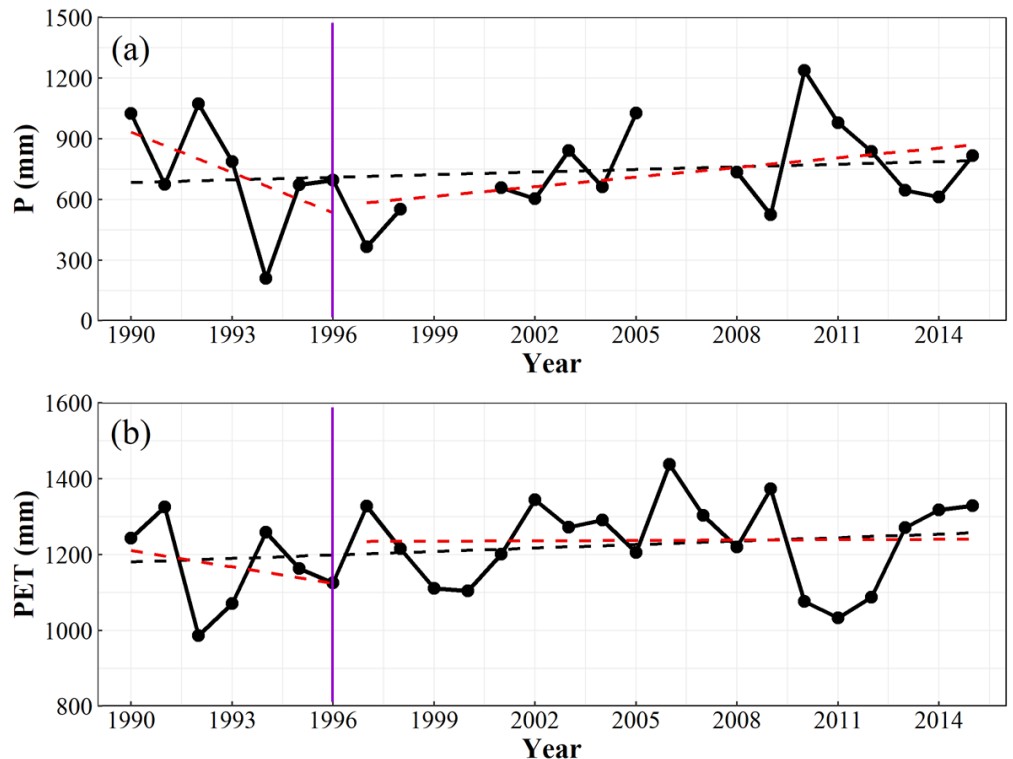

**Figure 7: Changes in (a) annual rainfall (*P*) and (b) annual potential evapotranspiration (PET) of the Kileys Run catchment (control catchment) during the period of 1990–2015.**

### 5.2 Multiyear drought induced changes in the rainfall-runoff relationship

According to the results in session 4.2, multiyear drought has led to shift in the rainfall-runoff relationship of paired catchments, which is similar to the significant downward shift of rainfall-runoff regression lines in basins in southeast Australia, the United States and China (Avanzi et al., 2020; Saft et al., 2015; Tian et al., 2018), and the increases in zero flow days with low flows being more affected than high flows of daily flow duration curves (FDCs) in 10 catchments from southeastern Australia, New Zealand and South Africa (Lane et al., 2005). In this study, the runoff coefficient decreased by 87.8% and 63.3% during the drought period

in the Red Hill and Kileys Run catchments, respectively. The latter was close to the decrease of runoff coefficient of 65.8% in Texas caused by extreme drought (Allen et al., 2011). Runoff coefficient decrease of the Red Hill catchment was higher than that of the Kileys Run catchment because runoff of the Red Hill catchment was also affected by afforestation, which can increased annual evaporation and decreased streamflow (Bruijnzeel, 1989; Cheng et al., 2017; Hoek Van Dijke et al., 2022). Multiyear drought can lead to more runoff reduction than predicted based on the rainfall-runoff relationship established during pre-drought

period as ignoring the impact of non-stationary changes may cause large errors in the results (Zhao et al., 2010). Compared with the line during drought period, the rainfall-runoff regression line moved up after multiyear drought due to heavy rainfall of 2010, but it did not return to the state before multiyear drought. Peterson et al. (2021) suggested that these changes may be due to severe water loss from transpiration during drought period.

    Inter-annual rainfall variability decreased and high rainfall years were missing during the drought period (see Fig. 2). Similar

changes were also reported in 124 watersheds in Australia during the drought period (Saft et al., 2015). The reduction of rainfall reduced runoff. In the Kileys Run and Red Hill catchments, rainfall primarily occurred in autumn and winter, less rainfall in autumn may resulted in lower antecedent soil moisture, which means more precipitation were used to replenish the soil water deficit in winter (Fig. 8). As a result, runoff in winter during drought period was less than that during pre-drought period and the decrease

of rainfall in next spring further aggravated runoff reduction. It was consistent with less runoff during the second period under the

same rainfall in Fig. 4. The decrease of GRACE satellite-observed average monthly terrestrial water storage and estimated groundwater storage in Murray–Darling Basin may support the above speculation about runoff reduction (van Dijk et al., 2013). The decline in groundwater levels may also be the reason for runoff reduction. Decline in precipitation usually resulted in a decline in groundwater levels (Peters et al., 2003), and may cause disconnection between groundwater and surface water (Kinal and Stoneman, 2012). Brutsaert (2008) demonstrated that annual lowest seven-day flow can be used indirectly to indicate the change

of ground water storage. The annual lowest seven-day flow in the Kileys Run catchment generally declined from 1990 to 1999, and was reduced further to zero from 2001 to 2010, suggesting ground water storage have dried up for a long time during multiyear drought.

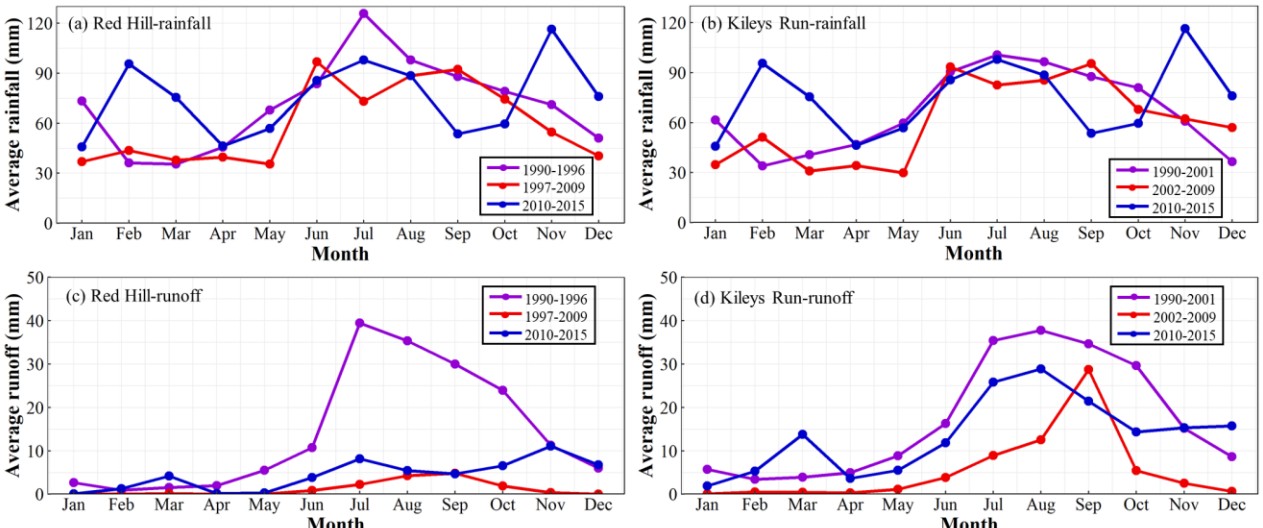

**Figure 8: Seasonal changes in (a) monthly rainfall and (c) monthly runoff of the Red Hill catchment (treated catchment),**
**(b) monthly rainfall and (d) monthly runoff of the Kileys Run catchment (control catchment) during the three different periods (1990-1996, 1997–2009, and 2010–2015 in Red Hill; 1990-2001, 2002–2009, and 2010–2015 in Kileys Run).**

**5.3 Application and suitability of the new framework under changing environments**

The traditional application of the three methods in catchments experienced multiyear drought may lead to large error because only two factors including non-stationary changes (or vegetation change) and stationary changes (or climate variability) are considered

to affect runoff, which is the essence of the limitations of traditional application (Dey and Mishra, 2017; Li et al., 2012; Zhang et al., 2019). In this study, a new framework was proposed by applying the TTM to the control catchment to quantify runoff changes caused by changes in the rainfall-runoff relationship induced by multiyear drought. Compared with the traditional application, the new framework further divided nonstationary changes into two parts driven by vegetation change and multiyear drought separately. Thus runoff changes caused by vegetation change, multiyear drought and climate variability can be partitioned and quantified (Fig.

3 and Table 2). This new framework also confirmed the fact that multiyear drought altered the rainfall-runoff relationship in the Red Hill catchment, and multiyear drought weakened the impact of vegetation change on runoff (see Table 2), which was important for us to design ecological engineering projects for sustainable water resources management (Brodribb et al., 2020; Newman et al., 2006; Xiao et al., 2020).

Climate variability and multiyear drought are supposed to have essentially different influences on the rainfall-runoff relationship

in this study. Climate variability is not supposed to result in non-stationary changes in the rainfall-runoff relationship, that is,

rainfall and runoff change at the same rate. While multiyear drought is assumed to result in non-stationary changes in the rainfall-runoff relationship, that is, it can be demonstrated by the significant abrupt change point on the double mass curves (DMCs) and the significant downward shift of rainfall-runoff linear regression line (Avanzi et al., 2020; Li et al., 2018). The multiyear drought in this study refers to drought with long duration and severe intensity, which can cause non-stationary changes in the rainfall-runoff relationship of catchments as shown in Fig. 4 and discussed in section 5.2. It is quite different from the wet/dry periods fluctuating near the average line (i.e., climate variability) (Han et al., 2019). For the two small studied catchments, the impact of climate fluctuations is very intense, and persistent fluctuations below the average are easy to cause non-stationary changes in the rainfall-runoff relationship because the long-term rainfall reduction may lead to changes of catchment characteristics, that is, lose connection between surface and groundwater. However, for large-scale watersheds, it is difficult to detect and to separate non-stationarity from variability because of the complexity and regional differences of positive and negative fluctuations or feedback of climate (Clark et al., 2016; Murakami et al., 2020). Negligence of non-stationarity induced by multiyear drought can result in significant differences in estimated effects of vegetation change as shown in Fig. 6, which has also been reported by Zhao et al. (2010) using about 16 years data at the same site. In the new framework, effect of multiyear drought is estimated between pre- and post-change periods as that of vegetation change, although two years of rainfall is above the average after 2000. Because the slope of DMCs is still very close to that during the period after 2009 (post-drought) (see Fig. 4).

Interactions between the impact of prolonged drought and that of land-use change may be existed. Several studies have reported that not only land use types but also soil and catchment properties may lead to different effects of drought on runoff (Saft et al., 2015; van Dijk et al., 2013). Here, one of the assumptions of the new framework is that the effects of three factors (vegetation change, hydroclimatic non-stationarity and climate variability) are independent of each other. We have to make this assumption to enable us to separate three effects with the help of paired catchments. The sum of the contribution of three factors to runoff changes is 111.4% in the Red Hill catchment, which is close to 100% and shows that the assumption is basically reasonable and valid. Considering these complex and secondary interactions amongst different factors, the new framework cannot separate them under the current experimental design and available data. How to estimate the interactions amongst different factors need to be carefully observed and investigated in the future.

In the new framework, the control catchment plays an irreplaceable role in estimating the impact of vegetation change and multiyear drought on runoff. Because the control catchment can eliminate the impact of climate variability and multiyear drought on runoff when the PCM is used to quantify runoff change caused by vegetation change, and the impact of multiyear drought on the treated catchment is transferred from the control catchment. The former must use the runoff data of the control catchment, and the latter needs both the rainfall and runoff data of the control catchment. One of the hypothesizes of the new framework is that the percentage of runoff reduction caused by multiyear drought of the control catchment ($r_c^n$) and the treated catchment ($r_t^n$) is same, and it might need further investigation in the future. The different response mechanism of runoff to multiyear drought in catchments with different properties is complex, it is closely related to climatic conditions, soil moisture, soil condition, groundwater levels, vegetation structure, etc. (Descroix et al., 2009; Stuart-Haëntjens et al., 2018; Yang et al., 2016). Most of the studies are qualitative description of the differences of runoff response in different catchments without quantitative analysis, such as, Jiao et al. (2020) and Liu et al. (2016) found cultivated lands and grasslands showed higher sensitivity to drought than natural biomes and forests exhibited the lowest sensitivity. However, it is difficult to quantify the difference in response to multiyear drought between the control catchment and the treated catchment, especially when afforestation and multiyear drought occur at the same time. Compared with runoff changes caused by multiyear drought in a single catchment, bias caused by different responses of vegetation cover of paired catchments to multiyear drought should be much smaller than non-stationary changes caused by

multiyear drought. Saft et al. (2016) re-evaluated a wide range of factors may be responsible for the additional runoff reductions and suggested that the shifts were mostly influenced by catchment characteristics related to pre-drought climate and soil and groundwater storage dynamics but less affected by the percentage of woody cover.

**5.4 The importance of the paired catchment experiments for estimating the effects of vegetation change on runoff under non-stationary conditions**

According to the results of this study, the non-stationary change of rainfall-runoff relationships in this two paired catchments caused by multiyear drought does not invalidate the paired-catchment method. The similar hydrological behavior of the control and treated catchments in terms of geomorphological, soil properties and climatic conditions determines that these two catchments have a relatively similar response process to multiyear drought and climate variability, which can be seen from the close occurrence time of the second abrupt change point in Fig. 4 (a) and (b). Therefore, the most significant difference between the control and

treated catchments between pre- and post-change periods is the change of vegetation cover type (the control catchment was kept as grassland unchanged and the treated catchment was covered by *P. radiata*). And the control catchment can eliminate the impact of multiyear drought and climate variability on the treated catchment by establishing the runoff-runoff relationship between this two catchments, so that the PCM can get true runoff changes caused by vegetation change. Therefore, the PCM is still a valid and fundamental method estimating the impact of vegetation change on runoff.

The length of data used in this study is extended from 16 years (used in Zhao et al. (2010) study) to 26 years. The difference between the contribution of vegetation change to the changes in total runoff estimated by this study and Zhao et al. (2010) is only 5.8%, which is also far less than the difference of the TTM and SBM. It shows that the increase of data length has little effect on the estimation of runoff change caused by vegetation change after runoff of catchment experiencing vegetation change has reached a new stable equilibrium state. The time required for runoff in different catchments to reach a new equilibrium state is different.

For example, the Red Hill catchment takes seven years (Zhao et al., 2010), Australia and New Zealand have suggested that three to 10 years, or even more (18 years for an afforested catchment in Biesievlei, South Africa (Brown et al., 2005)), majority of the time is between five and 10 years (Lane et al., 2005), are required for the treated catchment to reach a reasonably stable rainfall-runoff relationship after vegetation change. Han et al. (2020) provided a global assessment of the steady-state assumption in catchment water balance calculations for 1,057 global unimpaired catchments and shown that ~70% of the catchments attained

steady state within 10 years. For small catchment, it may need a shorter time to reach steady state. Thus the length of data used in this study (26 years) is enough to reach a steady state.

For Red Hill experiment site, the calibration period was from one year after treatment to the abrupt change point of annual runoff (1990–1996, seven years), because rainfall and runoff data before treatment were not measured. Zhao et al. (2010) compared the influences of two different methods for determining the calibration period on the estimated vegetation impacts at four paired

catchment sites. One is determined by the time of treatment. The other is determined by the abrupt change point of annual runoff. It was found that runoff changes caused by vegetation change were not sensitive to different calibration periods. Considering that runoff may not change significantly during the first few years after plantation of seedlings of *P. radiata*, we re-estimated the impact of vegetation change on runoff based on a calibration period with the data of the previous three years (1990–1992). Impacts of vegetation change were 34.2%, 74.2% and 61.0% of total runoff changes by the PCM, TTM and SBM, respectively. The

contributions of vegetation change, multiyear drought and climate variability to total runoff changes using the new framework were 34.2%, 37.4% and 39.0%, respectively. Comparing to those in Table 2, the difference of the contribution of vegetation change to total runoff changes was only 1.4%. It indicates that selection of the length of the calibration period may have little impact on

the estimation of runoff changes caused by vegetation change before the treated catchment reaches a new equilibrium state. This issue has also been discussed in Bren and Lane (2014) and they found that runoff of paired catchments had good calibrations (Nash-Sutcliffe efficiency (N-S) = 0.8) with 100 days of data and very little improvement after three years. For Red Hill experiment site, the change of N-S is close to that reported by Bren and Lane (2014). Good calibration (N–S > 0.85) is achieved with about 150 days. Similar results are obtained with monthly flows, good calibration (N–S > 0.35) is achieved with about 24 months. It suggests that runoff of the Red Hill and Kileys Run catchments will be well calibrated with calibration period exceeds 150 days (daily data) or 24 months (monthly data). Considering longer calibration period has lower mean error, calibration period is set from beginning of available data to the time of the abrupt change of annual runoff in this study.

## 6 Conclusions

Through the study of the typical paired-catchment experimental site – Red Hill, we found that multiyear drought during 2000–2009 had altered the stationary rainfall-runoff relationship of both the treated and control catchments. The runoff coefficient decreased by 87.8% and 63.3% during the drought period in the Red Hill and Kileys Run catchments, respectively. The paired-catchment method (PCM) is not invalidated by the non-stationarity induced by multiyear drought because of the role of the control catchment. However, the essence of the time-trend method (TTM) and the sensitivity-based method (SBM) is to separate runoff changes caused by non-stationary (vegetation change or/and multiyear drought) and stationary (climate variability) changes in the rainfall-runoff relationship, which makes the TTM and SBM significantly overestimate the impact of vegetation change on runoff. Estimated afforestation impacts were 32.8%, 93.5%, and 76.1% of total runoff changes by the PCM, TTM and SBM, respectively. On this basis, we propose a new framework by applying the TTM to the control catchment to quantify runoff changes caused by changes in the rainfall-runoff relationship induced by multiyear drought. Impacts of afforestation, multiyear drought and climate variability on runoff of the treated catchment (Red Hill) were 32.8%, 54.7% and 23.9%, respectively. The contribution of vegetation change to runoff reduction using the three methods under the new framework become consistent (32.8%, 38.8% and 21.4%). We demonstrated that the PCM was still a valid and fundamental method estimating the impact of vegetation change on runoff even the control catchment experienced hydroclimatic non-stationarity in the rainfall-runoff relationship under changing environments. This study provides a new way to more accurately quantify the impacts of vegetation change, climate variability and factors causing non-stationarity except vegetation change on runoff. The findings in this study not only gives insight in the change in hydrological processes caused by the combination of land use and climate changes, but also can help in developing strategies and management practices to ecological engineering under changing climate with frequent extremes in future.

## Code and data availability

The daily rainfall and runoff data are provided by Forests NSW (https://www.forestrycorporation.com.au/) and CSIRO (https://www.csiro.au/) in Australia. The monthly potential evapotranspiration data can be obtained from the SILO Data (www.longpaddock.qld.gov.au/silo/point-data/). All analyses were carried out with the open-source software R (https://www.r-project.org/). Code used in this manuscript are available from the corresponding author upon a reasonable request.

## Author contributions

YZ conceived the study, performed the analyses and prepared the manuscript. LC contributed to the study design and interpretation of the results. LZ provided data of rainfall and runoff. All the authors reviewed and edited the manuscript.

**Competing interests**

The authors declare that they have no conflict of interest.

**Acknowledgments**

This study was supported by the National Natural Science Foundation of China (51961145104, 51879193, 41890822). We thank all people and institutions who provide data used in this study.

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
