# Peer review of "Does non-stationarity induced by multiyear drought invalidate the paired-catchment method?"

_Hydrology and Earth System Sciences, 2022_

## Author Comment (AC2)

**Response to Reviewer #2**

**General comments**

**R1.** In general it's clear that the authors improved the original manuscript (https://doi.org/10.5194/hess-2021-5). They did improved the flow of the storyline and the analysis of the control catchment. The manuscript has appropriate objectives and gives insight in the change in hydrological processes caused by the combination of land use and climate changes.

Response: Thanks very much for your great efforts to assess our manuscript. We have studied your and reviewers' comments carefully and will make corrections/revisions as suggested. In the following, we have detailed how these comments (in black) are raised and our responses (in deep sky blue).

**R2.** Although the assumptions and fallowing conclusions are constructively fine and according to the applied methods, the arguments (especially in the abstract and conclusion) are not very clear. Although some comparative values per method are presented, for a reader it is not clear where the conclusions are based on.

Response: Thanks for bringing this to our attention. More specific descriptions about where the conclusions are based on will be mentioned in the mew manuscript.

Abstract:

Multiyear drought has been proved to cause non-stationary rainfall-runoff relationship. But whether this change can occur in catchments that have also experienced vegetation change and whether it invalidates the most widely used methods (the paired-catchment method (PCM), the time-trend method (TTM), and the sensitivity-based method (SBM)) for estimating impacts of vegetation change on runoff is still unknown and rarely discussed. In the Red Hill experimental catchment in Australia, which has experienced a 10-year multiyear drought (2000 – 2009) and afforestation, inconsistent results of the three methods were obtained. Estimated afforestation impacts were 32.8%, 93.5%, and 76.1% of total runoff changes by the PCM, TTM and SBM, respectively when the longer available observed record (1990 – 2015) was used in this study. In addition to afforestation, it is further found that multiyear drought which cannot be ignored has led to the non-stationary rainfall-runoff relationship. The runoff coefficient decreased by 87.8% and 63.3%, respectively, during the drought period in Red Hill and Kileys Run catchment. For the TTM and SBM, the paradigm and the calculations show that traditional application did not further differentiate different drivers (i.e. multiyear drought and vegetation change) of non-stationary rainfall-runoff relationship, which led to significant overestimation of afforestation effects (93.5% and 76.1%). However, for the PCM, the result (32.8%) is relatively correct because the runoff data of the control catchment offsets the common impacts on the two catchments, that is, multiyear drought and climate variability. Further, a new framework was proposed here to separate the effects of three factors on runoff changes including vegetation change, climate variability and hydroclimatic non-stationarity (i.e. multiyear drought). Based on the new framework, the percentage runoff reduction of the control catchment induced by multiyear drought is − 45%. Impacts of afforestation, multiyear drought and climate variability on runoff of the treated catchment (Red Hill) were 32.8%, 54.7% and 23.9%, respectively, and impacts of multiyear drought and climate variability on runoff of the control catchment (Kileys Run) were 87.2%% and 12.8%, respectively. With the new

framework, impacts of afforestation on runoff were 38.8% (93.5%−54.7%) and 21.4% (76.1%−54.7%) using the TTM and SBM, respectively, agreeing well with that by the PCM (32.8%). This study not only proved that multiyear drought can induce non-stationary rainfall-runoff relationship using experimental observations, but also proposed a new framework to better separate the impact of vegetation change on runoff. More importantly, it is proved that non-stationarity induced by multiyear drought does not invalidate the PCM, and PCM is still the most reliable method when paired catchments both experienced climate-induced shift in rainfall-runoff relationship.

**Specific comments**

**R3.** The abstract ends with "paired-catchment method is proven to be still the most reliable method even the control catchment experienced climate-induced shift in rainfall-runoff relationship", but within the abstract no information on the control catchments are presented. I would suggest to give actual numbers so that the reader is able to draw this conclusion himself.

Response: Thanks for your suggestions. "The percentage runoff reduction of the control catchment induced by multiyear drought is −45%.", "Impacts of multiyear drought and climate variability on runoff of the control catchment (Kileys Run) were 87.2% and 12.8%, respectively." will be added in abstract section.

**R4.** In my opinion it's a missed opportunity that the majority of input by the previous reviewers and especially the replies by the authors are not processed within this manuscript.

e. Response to reviewer 1 (https://doi.org/10.5194/hess-2021-5-AC1): "During 1988 the uppermost 50 ha of Red Hill was planted top. radiata, with the remaining 145 ha planted to p. radiata in April 1989. During 2003 the plantations in Red Hill catchment were thinned to remove pulpwood and to promote the growth of sawlogs in the remaining stands. During the prolonged drought period, no trees died in the treated catchment. The change of annual PET can be seen in Figure 8 (b) (Page 42). PET showed an insignificant (p-value>0.1) increasing trend of 3.5 mm year-1. PET initially decreased before 1996 and then increased after 1996. The mean annual PET during the period of 1990-1996, 1997-2009, 2010-2015 and 1997-2015 are 1168 mm, 1262 mm, 1186 mm and 1238 mm, respectively. Compared with the period of 1990-1996 and 2010-2015, the mean annual PET during the period of 1997-2009 (the prolonged drought occurred) increased by 94 mm and 76 mm, respectively. Compared with the period of 1990-1996, the mean annual PET during the period of 1997-2015 increased by 70 mm. It is consistent with the cognition that afforestation and drought can make PET increase."

e. reviewer 2 (https://doi.org/10.5194/hess-2021-5-AC2), which you may use in the introduction and/or discussion: "Considering the influence of prolonged drought on the rainfall-runoff relationship in the treated catchment, the result of the paired catchment method is closest to the real runoff change caused by vegetation change. Because the control catchment indirect eliminate the influence of prolonged drought and climate variability on the treated catchment under the assumption that the response of the two catchments to prolonged drought is similar. Interannual changes in watershed storage occur primarily in soil water and shallow groundwater, pools that are often hydrologically active at time scales shorter than 1 year (Sayama et al., 2011). Rice and Emanuel (2019) indicates that down-regulation of transpiration and inhibition of hydrologic connectivity by forest vegetation represent two

important negative feedback processes that can avert large losses in soil water or plant-accessible groundwater during dry periods. In doing so, this feedback mechanism has the potential to reinforce steady-state (or near-steady-state) conditions in dry conditions. So, the sensitivity-based method may be less affected by the prolonged drought because it is used in the forest and the time scale of PET and P data used in this method is annual scale. Runoff changes calculated by the sensitivity-based method are induced by climate variability."

Response: We apologize for these oversights, and thanks very much for your reminder.

(1) More detailed information about the characteristics of paired catchments, data and the history of vegetation change treatment will be mentioned in section 2 in the revised manuscript.
(2) The analysis of changes in potential evapotranspiration and rainfall will be mentioned in section 5.1 in the revised manuscript.
(3) More specific analysis of the application of traditional methods will be added in section 5.1 in the revised manuscript.

**R5.** For some statements I suggest to add some additional more recent papers.

e. at line 33.

e. at lines 52-53 "in many catchments around the world" two papers are cited one from 1980 and one from 2005, may be implying that these methods are not used that often anymore?

Response: Thanks for your suggestions. Changes will be made as suggested. (see references below).

line 33:

Cavalcante, R. B. L., Pontes, P. R. M., Souza Filho, P. W. M., and Souza, E. B.: Opposite Effects of Climate and Land Use Changes on the Annual Water Balance in the Amazon Arc of Deforestation, Water Resour. Res., 55, 3092-3106, https//doi.org/10.1029/2019WR025083, 2019.

Kalisa, W., Igbawua, T., Henchiri, M., Ali, S., Zhang, S., Bai, Y., and Zhang, J.: Assessment of climate impact on vegetation dynamics over East Africa from 1982 to 2015, Sci. Rep.-UK, 9, 16865, https//doi.org/10.1038/s41598-019-53150-0, 2019.

Zhang, J., Zhang, Y., Sun, G., Song, C., Dannenberg, M. P., Li, J., Liu, N., Zhang, K., Zhang, Q., and Hao, L.: Vegetation greening weakened the capacity of water supply to China's South-to-North Water Diversion Project, Hydrol. Earth Syst. Sci., 25, 5623-5640, https//doi.org/10.5194/hess-25-5623-2021, 2021.

lines 52-53:

Stoof, C. R., Vervoort, R. W., Iwema, J., van den Elsen, E., Ferreira, A. J. D., and Ritsema, C. J.: Hydrological response of a small catchment burned by experimental fire, Hydrol. Earth Syst. Sci., 16, 267-285, https//doi.org/10.5194/hess-16-267-2012, 2012.

Graeber, D., Goyenola, G., Meerhoff, M., Zwirnmann, E., Ovesen, N. B., Glendell, M., Gelbrecht, J., Teixeira De Mello, F., González-Bergonzoni, I., Jeppesen, E., and Kronvang, B.: Interacting effects of climate and agriculture on fluvial DOM in temperate and subtropical catchments, Hydrol. Earth Syst. Sci., 19, 2377-2394, https//doi.org/10.5194/hess-19-2377-2015, 2015.

Van Loon, A. F., Rangecroft, S., Coxon, G., Breña Naranjo, J. A., Van Ogtrop, F., and Van Lanen, H. A. J.:

Using paired catchments to quantify the human influence on hydrological droughts, Hydrol. Earth Syst. Sci., 23, 1725-1739, https//doi.org/10.5194/hess-23-1725-2019, 2019.

**R6.** Lines 79-93: The data and location description are very brief. I suggest to give some more extended information. i.e. It can be seen that Kileys Run experienced a multiyear drought that lasted … with the period of the Millennium Drought", explain why? What is lowest amount of rainfall measured?

Response: Thanks for your advice. Changes will be made as suggested in the revised manuscript.

(1) Red Hill catchment (1.95 km$^2$) and Kileys Run catchment (1.35 km$^2$) were paired catchments located 23 km north-east of Tumut and 100 km west of Canberra in New South Wales, Australia (35.322$^o$S, 149.137$^o$E) (Figure 1). The catchments are adjacent, and the soil texture, the topographic characteristics, and climatic conditions are similar. The altitude of the two catchments ranges from 590 m to 835 m above sea level. The slope in the lower part of catchments is mostly gentle, and gradually increases towards the ridge in a convex form. Geology of Red Hill catchment is predominately Young granodiorite, while Kileys Run catchment is dominated by Alkali diorite. Valley floor, midslope yellow duplex, shallow red soils and upslope red duplex are four main soil types in the two catchments. Upslope red duplex soils has the highest saturated hydraulic conductivities and valley floor soils has the lowest saturated hydraulic conductivities (Major et al., 1998). The climate of the two catchments is temperate with highly variable and winter-dominated rainfall. In 1988, *P. radiata* was planted in the eastern quarter of Red Hill catchment (0.5 km$^2$), and the remainder (1.45 km$^2$) was planted in April 1989. By 1997, pine occupied 78% of Red Hill catchment. During the multiyear drought period, no trees died in the treated catchment. The neighbouring catchment (Kileys Run) was the control catchment, which has been maintained as a grazed pasture control (Webb and Kathuria, 2012).

(2) Daily rainfall and runoff from the two catchments were collected during the period of 1990–2015. The daily rainfall was measured by tipping bucket rain gauges had been located at catchment outlet and the daily runoff was measured by a flat-v style crump weir at a gauging station at the outlet of each catchment (Major et al., 1998). The measured daily rainfall and runoff were made by government agency and were collected by Dr. Lu Zhang from CSIRO Land and Water, Black Mountain, Canberra, Australia. Mean annual rainfall and mean annual runoff of Red Hill catchment were 817 mm and 75 mm, respectively, during the study period. Mean annual rainfall and runoff of Kileys Run catchment were 817 mm and 161 mm, respectively, over the same period. Monthly potential evapotranspiration records were obtained from the SILO Data (www.longpaddock.qld.gov.au/silo/point-data/). The daily data were only used for the analysis of flow duration curve. The monthly data were used for the PCM, TTM, the new framework and double mass curve. The annual data are used in the SBM. Figure 2 shows the change of rainfall anomaly (%) in Kileys Run and Red Hill catchment. Rainfall anomaly (%) is defined as the percentage deviation of annual rainfall to mean manual rainfall. It can be seen that the three-year moving average of the rainfall anomaly (the black line) is less than zero from 2000 to 2009. According to the method of determining the multiyear drought period proposed by Saft et al. (2015), two catchments experienced a multiyear drought that lasted 10 years from 2000 to 2009 and this coincided with the period of the Millennium Drought. The minimum and maximum measured annual rainfall from 1990 to 2015 are 388.6 mm and 1334.4 mm, respectively.

**R7.** Lines 346-357, the paragraph about "Multiyear drought induced changes in the rainfall-runoff relationship" is short and doesn't discuss the subject. I suggest to compare your results with other locations where multiyear drought led to changes in the rainfall-runoff relationship.

Response: Thanks for your suggestions. Changes will be made as suggested in section 5.2.

It can be seen that multiyear drought has led to significant changes in rainfall-runoff relationship in Figure 4 and 5, which is similar to the significant downward shift of rainfall-runoff regression lines in basins in southeast Australia, the United States and China (Avanzi et al., 2020; Saft et al., 2015; Tian et al., 2018). In this study, the runoff coefficient decreased by 87.8% and 63.3% during the drought period in Red Hill and Kileys Run catchment, respectively. The latter is close to the decrease of runoff coefficient in Texas (Allen et al., 2011). The reason why the decrease of Red Hill is higher than that of Kileys Run may be that Red Hill is also affected by vegetation change (Saft et al., 2015). This will lead to more runoff reduction than predicted based on the rainfall-runoff relationship established in pre-drought period, and ignoring the impact of non-stationary change may cause large errors in the results (Zhao et al., 2010). Compared with the line during drought period, the rainfall-runoff regression line moved up after the drought due to heavy rainfall in 2010, but it did not completely return to the state before the drought. Peterson et al. (2021) suggested that these changes may be due to water loss from increased transpiration. Watersheds may thus have multiple states and a finite resilience to transient disturbances, and hydrological droughts can persist long after meteorological droughts.

**R8.** In this paragraph you do mention a change in physical processes (runoff, soil moisture and evapotranspiration?), but not really compared with studies elsewhere. Do you have any specific evidence available about changes processes?

Response: Thanks for your constructive comments. Changes will be made as suggested. More comparisons with studies elsewhere and descriptions of changes in annual lowest seven-day flow (it reflects the storage state of groundwater to a certain extent) will be added in section 5.2. Because only rainfall, runoff and potential evapotranspiration data can be obtained, the conjecture about the reasons is based on the analogy of similar catchments also affected by multiyear drought in Australia.

For precipitation input, inter-annual rainfall variability decreased and high rainfall years were missing during the drought period in Figure 2. Similar changes also occurred in 124 watersheds in Australia during the drought period (Saft et al., 2015). The reduction of rainfall input reduces runoff at the source. For runoff process, in Kileys Run and Red Hill catchment, rainfall primarily occurs in autumn and winter, less rainfall in autumn may result in lower antecedent soil moisture, which means more precipitation were used to replenish the soil water deficit in winter (Figure 7). The change of the monthly averages of rainfall and runoff is very similar to that of Murray Darling Basin, where rainfall-runoff relationship has changed caused by multiyear drought (Potter et al., 2010). As a result, the decrease of runoff began to increase in winter and the decrease of rainfall in spring further affected the runoff generation during the drought period, finally resulting in more runoff reduction than rainfall reduction. The decrease of GRACE satellite-observed average monthly terrestrial water storage and estimated groundwater storage in Murray–Darling Basin may support the above speculation about runoff reduction (van Dijk et al., 2013). The decline in groundwater levels may also be the reason for runoff reduction. Decline in precipitation usually results in a decline in groundwater levels (Peters et al., 2003), and may cause the connection between groundwater and surface water to be disrupted (Kinal and Stoneman, 2012). Brutsaert (2008)

demonstrated that annual lowest seven-day flow can be used to indicate the change of ground water storage in the absence of observations of groundwater level. The annual lowest seven-day flow in Kileys Run catchment generally declined from 1990 to 1999, and was reduced to 0 from 2001 to 2010.

**R9.** In addition to that, add some references about global and local knowledge about land use changes and their effects, such as the infiltration trade-off hypothesis (i.e. Bruijnzeel, 1989) and regional water availability caused by tree restoration such as (i.e. Hoek van Dijke, et al. 2022).

Response: Thanks for your advice. Changes will be made as suggested. (see references below).

Bruijnzeel, L. A.: Forestaion and dry season flow in the tropics: a closer look, J. Trop. For. Sci., 1, 229-243, 1989.

Wang-Erlandsson, L., Fetzer, I., Keys, P. W., van der Ent, R. J., Savenije, H. H. G., and Gordon, L. J.: Remote land use impacts on river flows through atmospheric teleconnections, Hydrol. Earth Syst. Sci., 22, 4311-4328, https//doi.org/10.5194/hess-22-4311-2018, 2018.

Hoek Van Dijke, A. J., Herold, M., Mallick, K., Benedict, I., Machwitz, M., Schlerf, M., Pranindita, A., Theeuwen, J. J. E., Bastin, J., and Teuling, A. J.: Shifts in regional water availability due to global tree restoration, Nat. Geosci., 15, 363-368, https//doi.org/10.1038/s41561-022-00935-0, 2022.

**R10.** Lines 412-426, I suggest to re-introduce abbreviations.

Response: Thanks for your advice. Changes will be made as suggested. The words will be modified to "paired-catchment method (PCM)", "time-trend method (TTM)" and "ssensitivity-based method (SBM)"

**R11.** Lines 417-418, conclusion. Add numbers so the reader is able to "agree" with your conclusion.

Response: Thanks for your suggestions. "The runoff coefficient decreased by 87.8% and 63.3%, respectively, during the drought period in Red Hill and Kileys Run catchment.", "Estimated afforestation impacts were 32.8%, 93.5%, and 76.1% of total runoff changes by the PCM, TTM and SBM, respectively.", "Impacts of afforestation, multiyear drought, and climate variability on runoff of the treated catchment (Red Hill) were 32.8%, 54.7% and 23.9%, respectively. Impacts of multiyear drought and climate variability on runoff of the control catchment (Kileys Run) were 87.2%% and 12.8%, respectively." will be added in the revised manuscript.

Conclusion:

Through the study of the typical paired-catchment experimental site – Red Hill, we found that multiyear drought during 2000 – 2009 had altered the stationary rainfall-runoff relationship of both the treated and control catchment. The runoff coefficient decreased by 87.8% and 63.3%, respectively, during the drought period in Red Hill and Kileys Run catchment. The paired-catchment method (PCM) is not invalidated by the non-stationarity induced by multiyear drought because of the role of the control catchment. However, the essence of the time-trend method (TTM) and the sensitivity-based method (SBM) is to separate runoff changes caused by non-stationary (vegetation change or/and multiyear drought) and stationary (climate variability) changes in rainfall-runoff relationship, which makes the TTM and SBM significantly overestimate the impact of vegetation change on runoff. Estimated

afforestation impacts were 32.8%, 93.5%, and 76.1% of total runoff changes by the PCM, TTM and SBM, respectively. On this basis, we propose a new framework by applying the TTM to the control catchment to quantify runoff changes caused by changes in rainfall-runoff relationship induced by multiyear drought. Impacts of afforestation, multiyear drought and climate variability on runoff of the treated catchment (Red Hill) were 32.8%, 54.7% and 23.9%, respectively. Impacts of multiyear drought and climate variability on runoff of the control catchment (Kileys Run) were 87.2%% and 12.8%, respectively. The contribution of vegetation change to runoff reduction using the three methods under the new framework become consistent (32.8%, 38.8%=93.5%−54.7% and 21.4%=76.1%−54.7%). We proved that the PCM is still a valid and fundamental method estimating the impact of vegetation change on runoff even paired catchments both experienced hydroclimatic non-stationarity in rainfall-runoff relationship under changing environments. This study provides a new way to quantify the impacts of vegetation change, climate variability and factors causing non-stationarity except vegetation change on runoff. The findings in this study not only gives insight in the change in hydrological processes caused by the combination of land use and climate changes, but also can help in developing strategies and management practices to ecological engineering under changing climate with frequent extremes in future.

**R12.** Idem line 420.

Response: Thanks for your suggestions. The lines will be modified to "The contribution of vegetation change to runoff reduction using the three methods under the new framework become consistent (32.8%, 38.8%=93.5%−54.7% and 21.4%=76.1%−54.7%)."

**Technical corrections**

**R13.** Line 82, what about Red Hill?

Response: Thanks for bringing this to our attention and more information about Red Hill catchment will be mentioned in the new manuscript.

Geology of Red Hill catchment is predominately Young granodiorite, while Kileys Run catchment also has Young granodiorite but is dominated by Alkali diorite. The ridges and upper slopes of each catchment have predominantly given rise to red duplex soils that comprise dark brown organic loam and silty loam overlying light to medium reddish-brown clay. In Red Hill catchment lower slopes comprise sandy soils consisting of dark grey-brown organic loam/silty loam overlying grey-brown bleached and heavily cemented loamy silt grading to yellow-brown or grey-brown mottled medium-heavy clay at depths > 45 cm. This soil type is absent from Kileys Run catchment and mean saturated hydraulic conductivities were 42.84 cm/day, 11.12 cm/day and 6.62 cm/day at depths of 20–50 cm, 50–80 cm and 80–100 cm, respectively (Major et al., 1998).

**R14.** Line 87, it would be nice to apply your analysis with data up to 2020?

Response: Thanks for bringing this to our attention. Unfortunately, we do not have measured runoff and rainfall data up to 2020. We can obtain the annual grid rainfall from 1990 to 2020, the average value is 940.0 mm. Based on the analysis of the grid rainfall anomaly from 1980 to 2020 and from 1990 to

2020, the two series both experienced multiyear drought from 2001 to 2009 (Figure R1), which is consistent with the period of multiyear drought in the manuscript (the drought period is from 2000 to 2009). There is an alternate change of wet and dry after 2015, especially, a large negative rainfall anomaly (%) is existed after 2017. Therefore, the extended data may not greatly improve the results.

[Figure]

Figure R1 Rainfall anomaly (%) as a percentage of the mean annual rainfall of Kileys Run and Red Hill catchment. (a) Grid rainfall data (1980-2020); (b) Grid rainfall data (1990-2020). Red bars represent dry years and blue bars represent wet years. The black line represents the three-year moving average of the rainfall anomaly.

**R15.** Line 87, which method did you used to measure the runoff? Of where did you collected the data?

Response: Thanks for bringing this to our attention and the source of the data will be mentioned in the revised manuscript.

Runoff data were measured on site by government agency and were collected by Dr. Lu Zhang from CSIRO Land and Water, Black Mountain, Canberra, Australia. Observations were made at the outlet of each catchment, a gauging station was installed and began operation on 26 May, 1989. Daily runoff data are therefore available from June 1989 onwards. The gauging stations comprise a flat-v style crump weir at Red Hill and Kileys Run catchments. The rating curves (stage-discharge relationships) for Red Hill and Kileys Run catchment had been developed and tested by a series of velocity-area gauging and, as a result, the runoff data from those stations was considered reliable (Webb and Kathuria, 2012).

**R16.** Lines 88-89, add more information/background. You use daily rainfall and runoff, but monthly PET?

Response: Thanks for your advice. Changes will be made as suggested in section 2.

The daily rainfall and runoff data were measured on site by government agency and were collected by Dr. Lu Zhang from CSIRO Land and Water. The potential evapotranspiration (PET) data were monthly data from the SILO Data ([www.longpaddock.qld.gov.au/silo/point-data/](www.longpaddock.qld.gov.au/silo/point-data/)). The daily data were only used for the analysis of flow duration curve. The monthly data were used for the PCM, TTM, the new framework and double mass curve. The annual data were used in the SBM.

**R17.** Line 90, "figure 2 shows the rainfall anomaly that was calculated by the method proposed by", I suggest to describe the method (define anomaly). Which in this case is the annual percentage of rainfall being larger or less than the average rainfall.

Response: Thanks for your advice. Changes will be made as suggested in section 2.

Rainfall anomaly (%) is defined as the percentage deviation of annual rainfall to mean manual rainfall. The period of drought is determined by rainfall anomaly (%) smoothed with a three-year moving window. The method for determining the multiyear drought period is as follows (Saft et al., 2015):

The first year of the drought remains the start of the first three-year negative anomaly period;

The end year is set as the last year of this three year negative anomaly period (unless: if the last two years have slightly positive anomalies (but each <15% of the mean), in which case the end year is set to the first year of positive anomaly);

The length of dry periods must be not less than seven years;

The mean dry period anomaly must be less than $-5\%$;

**R18.** Line 91-92, I suggest add to add the reference again.

Response: Thanks for your advice. Changes will be been made as suggested. (see references below).

van Dijk, A. I. J. M., Beck, H. E., Crosbie, R. S., de Jeu, R. A. M., Liu, Y. Y., Podger, G. M., Timbal, B., and Viney, N. R.: The Millennium Drought in southeast Australia (2001-2009): Natural and human causes and implications for water resources, ecosystems, economy, and society, Water Resour. Res., 49, 1040-1057, https//doi.org/10.1002/wrcr.20123, 2013.

Peterson, T. J., Saft, M., Peel, M. C., and John, A.: Watersheds may not recover from drought, Science, 372, 745-749, https//doi.org/10.1126/science.abd5085, 2021.

**R19.** Figures 3 – 5, and 7: used colours (combination of red and green) are not very suitable for colour blind readers (HESS - Submission (hydrology-and-earth-systemsciences.net))

Response: We apologize for these oversights. Figures 3 – 5, and 7 will be modified in the new manuscript.

**R20.** Lines 363-365, add reference?

Response: Thanks for your advice. Changes will be made as suggested. (see references below).

Li, H., Zhang, Y., Vaze, J., and Wang, B.: Separating effects of vegetation change and climate variability

using hydrological modelling and sensitivity-based approaches, J. Hydrol., 420-421, 403-418, https//doi.org/10.1016/j.jhydrol.2011.12.033, 2012.

Dey, P., and Mishra, A.: Separating the impacts of climate change and human activities on streamflow: A review of methodologies and critical assumptions, J. Hydrol., 548, https//doi.org/10.1016/j.jhydrol.2017.03.014, 2017.

Zhang, L., Nan, Z., Wang, W., Ren, D., Zhao, Y., and Wu, X.: Separating climate change and human contributions to variations in streamflow and its components using eight time‐trend methods, Hydrol. Process., 33, 383-394, https//doi.org/10.1002/hyp.13331, 2019.

**R21.** Line 372, add reference?

Response: Thanks for your advice. Changes will be made as suggested (see references below).

Xiao, Y., Xiao, Q., and Sun, X.: Ecological Risks Arising from the Impact of Large-scale Afforestation on the Regional Water Supply Balance in Southwest China, Sci. Rep.-UK, 10, 4150, https//doi.org/10.1038/s41598-020-61108-w, 2020.

Newman, B. D., Wilcox, B. P., Archer, S. R., Breshears, D. D., Dahm, C. N., Duffy, C. J., McDowell, N. G., Phillips, F. M., Scanlon, B. R., and Vivoni, E. R.: Ecohydrology of water-limited environments: A scientific vision, Water Resour. Res., 42, https//doi.org/https://doi.org/10.1029/2005WR004141, 2006.

Brodribb, T. J., Powers, J., Cochard, H., and Choat, B.: Hanging by a thread? Forests and drought, Science, 368, 261-266, https//doi.org/10.1126/science.aat7631, 2020.

**R22.** Figure 4, perhaps add cumulative rainfall and the separated periods.

Response: Thanks for your advice. Changes will be made as suggested in Figure 4.

The rainfall data used in the two catchments are the same. The measured cumulative rainfall shown in Figure R2 has two significant abrupt change points in October 1999 and March 2012. They are consistent with the beginning and end of multiyear drought (2000 – 2009).

[Figure]

Figure R2 Double mass curve of monthly rainfall and N (cumulative sum of 1 to 312, 321=12 months/year × 26 years.

**References:**

Allen, P. M., Harmel, R. D., Dunbar, J. A., and Arnold, J. G.: Upland contribution of sediment and runoff during extreme drought: A study of the 1947–1956 drought in the Blackland Prairie, Texas, J. Hydrol., 407, 1-11, https//doi.org/https://doi.org/10.1016/j.jhydrol.2011.04.039, 2011.

Avanzi, F., Rungee, J., Maurer, T., Bales, R., Ma, Q., Glaser, S., and Conklin, M.: Climate elasticity of evapotranspiration shifts the water balance of Mediterranean climates during multi-year droughts, Hydrol. Earth Syst. Sci., 24, 4317-4337, https//doi.org/10.5194/hess-24-4317-2020, 2020.

Brutsaert, W.: Long-term groundwater storage trends estimated from streamflow records: Climatic perspective, Water Resour. Res., 44, https//doi.org/10.1029/2007WR006518, 2008.

Kinal, J., and Stoneman, G. L.: Disconnection of groundwater from surface water causes a fundamental change in hydrology in a forested catchment in south-western Australia, J. Hydrol., 472-473, 14-24, https//doi.org/10.1016/j.jhydrol.2012.09.013, 2012.

Major, E. J., Cornish, P. M., and Whiting, J. K.: Red Hill hydrology project establishment report including a preliminary water yield analysis, Forest Research and Development Division, State Forests of New South Wales, Sydney, 24 pp., 1998.

Peters, E., Torfs, P. J. J. F., van Lanen, H. A. J., and Bier, G.: Propagation of drought through groundwater- A new approach using linear reservoir theory, Hydrol. Process., 17, 3023-3040, https//doi.org/10.1002/hyp.1274, 2003.

Peterson, T. J., Saft, M., Peel, M. C., and John, A.: Watersheds may not recover from drought, Science, 372, 745-749, https//doi.org/10.1126/science.abd5085, 2021.

Potter, N. J., Chiew, F. H. S., and Frost, A. J.: An assessment of the severity of recent reductions in rainfall and runoff in the Murray–Darling Basin, J. Hydrol., 381, 52-64, https//doi.org/10.1016/j.jhydrol.2009.11.025, 2010.

Saft, M., Western, A. W., Zhang, L., Peel, M. C., and Potter, N. J.: The influence of multiyear drought on the annual rainfall-runoff relationship: An Australian perspective, Water Resour. Res., 51, 2444-2463, https//doi.org/10.1002/2014WR015348, 2015.

Tian, W., Liu, X., Liu, C., and Bai, P.: Investigation and simulations of changes in the relationship of precipitation-runoff in drought years, J. Hydrol., 565, 95-105, https//doi.org/https://doi.org/10.1016/j.jhydrol.2018.08.015, 2018.

van Dijk, A. I. J. M., Beck, H. E., Crosbie, R. S., de Jeu, R. A. M., Liu, Y. Y., Podger, G. M., Timbal, B., and Viney, N. R.: The Millennium Drought in southeast Australia (2001-2009): Natural and human causes and implications for water resources, ecosystems, economy, and society, Water Resour. Res., 49, 1040-1057, https//doi.org/10.1002/wrcr.20123, 2013.

Webb, A., and Kathuria, A.: Response of streamflow to afforestation and thinning at Red Hill, Murray Darling Basin, Australia, J. Hydrol., 412-413, 133-140, https//doi.org/10.1016/j.jhydrol.2011.05.033, 2012.

Webb, A. A., and Kathuria, A.: Response of streamflow to afforestation and thinning at Red Hill, Murray Darling Basin, Australia, J. Hydrol., 412-413, 133-140, https//doi.org/10.1016/j.jhydrol.2011.05.033, 2012.

Zhao, F., Zhang, L., Xu, Z., and Scott, D. F.: Evaluation of methods for estimating the effects of vegetation change and climate variability on streamflow, Water Resour. Res., 46, https//doi.org/10.1029/2009WR007702, 2010.

---

## Author Comment (AC3)

**Response to Reviewer #3**

**General Comments:**

**R1**. This is an interesting paper on an important topic. In fact it could be argued the issues discussed in this paper are absolutely fundamental in hydrology. This is particularly true in light of the way "data" is used uncritically in many studies.

Response: Thanks very much for your great efforts to assess our manuscript. We have studied your and reviewers' comments carefully and will make corrections/revisions as suggested. In the following, we have detailed how these comments (in black) are raised and our responses (in deep sky blue).

**R2.** The paper is generally well written and structured, although there is a need for a more careful read-through as there is some awkward syntax and grammar (particularly in the Discussion) where the quality of the writing seems to wander a little. My main questions relate to the data set used and the essential paradigm of the PCM. The Redhill site did not have a calibration period. Although the authors suggest that the first period of treatment may be thought of as non-treated in that the trees were very small and not high water using, I feel the implications of this may be important. This then connects to the PCM paradigm; that is, that the length of calibration or the approach developing the calibrations in theory should account for the type of non-stationarity that is discussed (ie. drought). Putting this another way, how can we decide what is non-stationarity and what is variability? This is particularly germane to Australian hydrology where we experience significant variability. I am not suggesting climates are stationary, but disentangling non-stationarity from variability with a relatively short data period (in climate terms) is a question.

Response: Thanks for your constructive comments. We will add more contents about the calibration period, non-stationarity and variability in discussion section in the revised manuscript.

(1) We agree with the question of the calibration period. However, for Red Hill catchment, it is unfortunate that rainfall and runoff data before treatment were not measured. In Zhao et al. (2010), the impacts of the calibration period determined by treatment period and period before abrupt change point of annual runoff on the results at four paired catchments were compared, it was found that runoff changes caused by vegetation change are not sensitive to the different calibration periods. Moreover, considering that the runoff may not change significantly in the first few years after plantation of seedlings of *P. radiata*, we re-estimated the impact of vegetation change on runoff based on a calibration period with the data of the previous three years. The contribution of vegetation change to total runoff changes is 34.2%, and the difference with the result in the manuscript is only 1.4%. Therefore, the calibration period set in this manuscript is reasonable.

(2) In this study, we distinguish climate variability and non-stationarity by considering the influence on the rainfall-runoff relationship. For climate variability, we think that it will not lead to non-stationary changes in rainfall-runoff relationship, that is, rainfall and runoff changes at the same rate. For non-stationarity, it will lead to non-stationary changes in rainfall-runoff relationship, that is, it can be reflected by the significant abrupt change point of double mass curve and the

significant up/down movement of rainfall-runoff linear regression line (Avanzi et al., 2020; Li et al., 2018). For large-scale watersheds, it is really difficult to detect and confirm non-stationarity from variability because of the complexity and regional differences of positive and negative fluctuations or feedback of climate. However, for the two small studied catchments, the impact of climate fluctuations is very intense, and persistent fluctuations below the average are easy to cause non-stationary changes in rainfall-runoff relationship.

**R3.** The calibration period issue is a vexed one as there is no longer an appetite by funding bodies to set up a paired-catchment experiment and then wait for a lengthy period before anything happens. Bren and Lane (2014, JH 519) explored this issue and proposed a method using daily flows that rather obviously increases the number of data points. Somewhat surprisingly the analysis showed that good calibrations (Nash-Sutcliffe E = 0.8) with 100 days of data, and very little improvement after 3 years. Apologies for the treatise, but I wonder if this approach might be useful in thinking about the PCM. That is, if such an analysis was performed and compared with the other analyses it might be very useful. Data could be randomly pulled out of the Kileys Run data. At the very least it should be discussed.

Response: Thanks for your suggestion. We will add discussions regarding to your concerns in the revised manuscript.

We use the data of seven years before the abrupt change point of annual runoff and three years after the beginning of the data series to explore the change of Nash–Sutcliffe coefficient (N-S) with the increasing length of the observation set. For seven years (1990-1996), the calibration set is from 1990 to 1992 and the verification set is from 1993 to 1996 (Figure R1 (b) and (d)). For three years (1990-1992), the calibration set is from 1990 to 1992 and the verification set is from 1993 to 1996 (Figure R1 (a) and (c)). According to Figure R1, it can be found that the change of N-S is similar to Bren and Lane (2014). It shows that good calibration (N–S > 0.85) is achieved in about 150 days, and then maintains at the high-level value. Similar results are obtained with monthly of flows, calibration (N–S > 0.35) is achieved in about 24 months.

We use the data of the previous three years (1990-1992) as pre-treatment period, and re-estimate the effects of vegetation change on runoff by the PCM, TTM, SBM and new framework. Impacts of vegetation change were 34.2%, 74.2% and 61.0% of total runoff changes by the PCM, TTM and SBM, respectively. The contributions of vegetation change, multiyear drought and climate variability to total runoff changes are 34.2%, 37.4% and 39.0%, respectively, using the new framework. The sum of the three terms is 110.6%, with a difference of 10.6% from 100%. When the pre-treatment period is as pre-change period of runoff (1990-1996) used in the manuscript, impacts of vegetation change were 32.8%, 93.5% and 76.1% of total runoff changes by the PCM, TTM and SBM, respectively. The contributions of vegetation change, multiyear drought and climate variability to total runoff changes are 32.8%, 54.7% and 23.9%, respectively, using the new framework. The sum of the three terms is 111.4%, with a difference of 11.4% from 100% and a difference of 0.8% from 110.6%. It also can be found that the difference in the contribution of vegetation change to total runoff changes estimated by the two different ways to determine the calibration period is only 1.4%. It shows that taking period before the abrupt change point of annual runoff as the calibration period is also reasonable considering a lower mean error.

[Figure]

Figure R1. Mean of 10 values of the N–S coefficient of the Coefficient of Determination as a function of the number of days and months in the calibration data set for the Kileys Run and Red Hill data using the 10-fold cross-validation approach to monitor the development of the calibration. ''Rand'' refers to the data being randomized.

**Specific Comments:**

**R4.** Line 40 - there are more updated references for the research in Australian catchment behaviour that are relevant (eg Petersen et al, 2021, Science 372)

Response: Thanks for your advice. Changes will be made as suggested. (see references below).

King, A. D., Pitman, A. J., Henley, B. J., Ukkola, A. M., and Brown, J. R.: The role of climate variability in Australian drought, Nat. Clim. Change, 10, 177-179, https//doi.org/10.1038/s41558-020-0718-z, 2020.

Peterson, T. J., Saft, M., Peel, M. C., and John, A.: Watersheds may not recover from drought, Science, 372, 745-749, https//doi.org/10.1126/science.abd5085, 2021.

**R5.** L46. I think this statement about PCM and non-stationarity requires more justification. How does it not deal with the issue given that is the paradigm of PCM?

Response: Thanks for your constructive comments. We will add more justification about PCM and non-stationarity in the revised manuscript.

At Red Hill site, the treated catchment suffered from non-stationary changes caused by vegetation change and multiyear drought, and stationary changes caused by climate variability. The control catchment suffered from non-stationary changes only caused by multiyear drought, and stationary changes caused by climate variability. The PCM can offset the effects on both paired catchments induced by multiyear drought and climate variability based on data of the control catchment, and the effects of

vegetation change on the treated catchment can then be separated. That is, for the paradigm of PCM, when the treated catchment is affected by N+1 kinds of non-stationary changes and the control catchment is also affected by the same N kinds of non-stationary changes, this method still can separate the effect of the N+1 non-stationary change that does not occur in the control catchment on the treated catchment. The PMC is still the most reliable method compared with other methods.

**R6.** L 65+ Both TTM and SBM require lengthy records; is this an issue with these analyses at Redhill? There may be an argument to see the drought as a plus in terms of a record with wet/dry periods.

Response: Thanks for your constructive comments. The discussion about this question will be mentioned in the revised manuscript.

(1) The length of data is essentially a problem of reaching an equilibrium state for catchments. Han et al. (2020) provided a global assessment of the steady-state assumption in catchment water balance calculations for 1,057 global unimpaired catchments. Results show that ~70% of the catchments attain steady state within 10 years. The time needed for a catchment to reach steady state shows a close relationship with climatic aridity and vegetation coverage, with arid/semiarid and sparsely vegetated catchments generally having a longer time. For small catchment, it may need a shorter time to reach steady state. It can be seen in Figure 4 (a), the double mass curve becomes a straight line in a very short time. It demonstrates that the length of data used in this study (26 years) is acceptable.

(2) The multiyear drought mentioned in the manuscript is a drought with longer duration and greater intensity, which will cause non-stationary changes in rainfall-runoff relationship of catchments. It is quite different from the wet/dry periods floating near the average line. The effects of multiyear drought on runoff are very significant and cannot be ignored. It is necessary to consider impacts of multiyear drought on runoff in the new framework.

**R7.** Lane et al. 2005 (JH) used FDCs that included Redhill – might be worth including these results as a comparison from a different method. This paper also has some estimates of time to equilibrium.

Response: Thanks for your advice. Changes will be made as suggested in section 5.2 and 5.4 in the new manuscript.

**R8.** L 69 – syntax not great "issues about this hypothesis" could be improved

Response: We apologize for these oversights. The lines will be modified to "However, this hypothesis has not been explored and verified, and it is important to examine whether non-stationarity induced by multiyear drought invalidate the PCM and the applicability of the three widely used methods under changing climate with frequent extremes in the future."

**R9.** L 107 – should be double mass curves, FDCs etc. There are quite a few examples of this, need a careful read.

Response: We apologize for these oversights. We will modify them in the revised manuscript.

**R10.** L 168- See earlier general discussion. It does trouble me that a site with no calibration is used for this study. In addition, the vegetation effect is dynamic; growing from seedlings to (presumably, given there are no growth data) a closed canopy. I do wonder if this really is the best data set for such a study, or what might be gained from using more data sets.

Response: Thanks for your constructive comments. Except for Red Hill site, there are no paired catchments that have experienced vegetation change and multiyear drought at the same time. The use of data and the calibration period are also further discussed in the responses to comments R2 and R3. It is found that the difference in the contribution of vegetation change to total runoff changes estimated by the two different ways to determine the calibration period (the previous three years after treatment and seven years before the abrupt change point of annual runoff in the manuscript) is only 1.4%. In this study, we not only find the reasons why the results of the three traditional methods are inconsistent, but also think more deeply about the paradigm of PCM. For PCM, when the treated catchment is affected by N+1 kinds of non-stationary changes and the control catchment is affected by the same N kinds of non-stationary changes, this method still can separate the effect of the N+1 non-stationary change that does not occur in the control catchment on the treated catchment. The PMC is still the most reliable method compared with other methods.

**R11.** Figure 3 is a great figure!

Response: Thanks for your comments.

**R12.** Table 2 – total flow changes would be useful. They appear later in the text but I think having totals in the table make it easier to evaluate the methods.

Response: Thanks for your advice. The total flow changes have been mentioned in Table 2.

**R13.** This also brings up another point that I don't think has been discussed properly. $Q_{clim}$ is conceptualised as the climate effect, encompassing wet and dry and mean climate inputs. I am not sure there is adequate discussion of how this does not deal with the climate issue as formulated.

Response: Thanks for your constructive comments. The discussion about this question will be mentioned in the revised manuscript.

$Q_{clim}$ in this study refers to the impact of climate variability (wet and dry spells) on runoff that does not lead to non-stationary changes in rainfall-runoff relationship. It represents the runoff changes caused by climate stability changes and it is estimated by the SBM. The SBM is derived from the Budyko framework. In the formula, the impact of climate change on runoff is estimated by rainfall and potential evapotranspiration changes. In the Budyko curve, it is reflected in moving from one point to another point on the same curve. There is a necessary assumption in the SBM, that is, a transition from one steady state to another with no change in catchment properties (the same curve) (Roderick and Farquhar, 2011; Sun et al., 2014). When the SBM is applied to Red Hill catchment that has experienced multiyear drought, the result is actual the impact of climate variability (without changing the catchment characters/non-stationary changes in rainfall-runoff relationship) on runoff, that is, $Q_{clim.}$ It may ignore

the impact of multiyear drought on runoff and indirectly overestimate the impact of vegetation change on runoff. Because catchment properties of Red Hill catchment may have changed due to multiyear drought (Kinal and Stoneman, 2012; Peterson et al., 2021; Saft et al., 2016; van Dijk et al., 2013), which violates the assumptions of the SBM.

**R14.** 5.2.1 – this paragraph brings up the interesting point (that is the subject of the Saft/Peterson/Fowler etc studies); is it the climate that is non-stationary or is the processes (obviously driven by the climate).

Response: Thanks for your constructive comments. We think that climate change is a combination of non-stationary changes and process. The non-stationary change may be due to the sudden and drastic changes of human activities (explosive increase of industrial activities (Bauska et al., 2015)) and/or the amount of solar energy that gets to earth (Karl and Trenberth, 2003) and other factors. Most of these non-stationary changes occur suddenly and change from one equilibrium state to another through a relatively short period, such as non-stationary change in rainfall-runoff relationship caused by multiyear drought (Fowler et al., 2018; Peterson et al., 2021; Saft et al., 2015). The process is long-term, it may keep the equilibrium state and change continuously, or show a trend change (increase or decrease) over time. For example, the air temperature shows a gradual upward trend in a longer period (years and decades), and in a shorter period (days, months and seasons), the temperature constantly fluctuates around the average temperature, which means that there will still be those days which are cool and those days which are warm (Hoegh-Guldberg et al., 2019).

**R15.** The Paragraph around Line 375 needs some rewriting, the syntax is jarring. For example "the" control..

Response: We apologize for these oversights. Changes will be made as suggested in the revised manuscript.

**R16.** L 388 "Because Saft.." this is a poor sentence

Response: We apologize for these oversights. The lines will be modified to "Saft et al. (2016) re-evaluate a large range of factors suggested to be responsible for the additional runoff reductions. Results suggest that the shifts were mostly influenced by catchment characteristics related to pre-drought climate and soil and groundwater storage dynamics, but less affected by the percentage of woody cover."

**R17.** L 399 "pines" should not be italicized. P.radiata would be

Response: We apologize for these oversights. The word "*pines*" will be modified to "pines".

**References:**

Avanzi, F., Rungee, J., Maurer, T., Bales, R., Ma, Q., Glaser, S., and Conklin, M.: Climate elasticity of evapotranspiration shifts the water balance    of Mediterranean climates during multi-year droughts,

Hydrol. Earth Syst. Sci., 24, 4317-4337, https//doi.org/10.5194/hess-24-4317-2020, 2020.

Bauska, T. K., Joos, F., Mix, A. C., Roth, R., Ahn, J., and Brook, E. J.: Links between atmospheric carbon dioxide, the land carbon reservoir and climate over the past millennium, Nat. Geosci., 8, 383-387, https//doi.org/10.1038/ngeo2422, 2015.

Bren, L. J., and Lane, P. N. J.: Optimal development of calibration equations for paired catchment projects, J. Hydrol., 519, 720-731, https//doi.org/https://doi.org/10.1016/j.jhydrol.2014.07.059, 2014.

Fowler, K., Coxon, G., Freer, J., Peel, M., Wagener, T., Western, A., Woods, R., and Zhang, L.: Simulating Runoff Under Changing Climatic Conditions: A Framework for Model Improvement, Water Resour. Res., 54, 9812-9832, https//doi.org/10.1029/2018WR023989, 2018.

Han, J., Yang, Y., Roderick, M. L., McVicar, T. R., Yang, D., Zhang, S., and Beck, H. E.: Assessing the Steady-State Assumption in Water Balance Calculation Across Global Catchments, Water Resour. Res., 56, https//doi.org/10.1029/2020WR027392, 2020.

Hoegh-Guldberg, O., Jacob, D., Taylor, M., Guillén, B. T., Bindi, M., Brown, S., Camilloni, I. A., Diedhiou, A., Djalante, R., Ebi, K., Engelbrecht, F., Guiot, J., Hijioka, Y., Mehrotra, S., Hope, C. W., Payne, A. J., Pörtner, H. O., Seneviratne, S. I., Thomas, A., Warren, R., and Zhou, G.: The human imperative of stabilizing global climate change at 1.5°C, Science, 365, eaaw6974, https//doi.org/10.1126/science.aaw6974, 2019.

Karl, T. R., and Trenberth, K. E.: Modern Global Climate Change, Science, 302, 1719-1723, https//doi.org/10.1126/science.1090228, 2003.

Kinal, J., and Stoneman, G. L.: Disconnection of groundwater from surface water causes a fundamental change in hydrology in a forested catchment in south-western Australia, J. Hydrol., 472-473, 14-24, https//doi.org/10.1016/j.jhydrol.2012.09.013, 2012.

Li, Q., Wei, X., Zhang, M., Liu, W., Giles-Hansen, K., and Wang, Y.: The cumulative effects of forest disturbance and climate variability on streamflow components in a large forest-dominated watershed, J. Hydrol., 557, 448-459, https//doi.org/10.1016/j.jhydrol.2017.12.056, 2018.

Peterson, T. J., Saft, M., Peel, M. C., and John, A.: Watersheds may not recover from drought, Science, 372, 745-749, https//doi.org/10.1126/science.abd5085, 2021.

Roderick, M. L., and Farquhar, G. D.: A simple framework for relating variations in runoff to variations in climatic conditions and catchment properties, Water Resour. Res., 47, https//doi.org/10.1029/2010WR009826, 2011.

Saft, M., Peel, M. C., Western, A. W., and Zhang, L.: Predicting shifts in rainfall-runoff partitioning during multiyear drought: Roles of dry period and catchment characteristics, Water Resour. Res., 52, 9290-9305, https//doi.org/10.1002/2016WR019525, 2016.

Saft, M., Western, A., Zhang, L., Peel, M., and Potter, N.: The influence of multiyear drought on the annual rainfall-runoff relationship: An Australian perspective, Water Resour. Res., 51, https//doi.org/10.1002/2014WR015348, 2015.

Sun, Y., Tian, F., Yang, L., and Hu, H.: Exploring the spatial variability of contributions from climate variation and change in catchment properties to streamflow decrease in a mesoscale basin by three different methods, J. Hydrol., 508, 170-180, https//doi.org/https://doi.org/10.1016/j.jhydrol.2013.11.004, 2014.

van Dijk, A. I. J. M., Beck, H. E., Crosbie, R. S., de Jeu, R. A. M., Liu, Y. Y., Podger, G. M., Timbal, B., and Viney, N. R.: The Millennium Drought in southeast Australia (2001-2009): Natural and human causes and implications for water resources, ecosystems, economy, and society, Water Resour. Res., 49, 1040-1057, https//doi.org/10.1002/wrcr.20123, 2013.

Zhao, F., Zhang, L., Xu, Z., and Scott, D. F.: Evaluation of methods for estimating the effects of vegetation change and climate variability on streamflow, Water Resour. Res., 46, https//doi.org/10.1029/2009WR007702, 2010.

---

## Author Response (AR1)

**Response to Reviewer #1**

**R1.** This article looks at methods for partitioning changes in the rainfall-runoff relationship between vegetation changes, climatic variation, and non-stationarity in runoff generation.

**Response:** Thanks very much for your great efforts to assess our manuscript. We have studied your and reviewers' comments carefully and have made corrections/revisions as suggested. The point-to-point responses to the comments and revision are detailed below. In the following, we have detailed how these comments (in black) are raised and our responses (in deep sky blue).

**R2.** The three acronyms PCM, TTM and SBM are introduced and explained in parentheses by the second sentence of the Abstract, but not until the third paragraph of the Introduction and then after the acronyms are already used. Please use and explain them as soon as they are mentioned in the main text of the paper.

**Response:** We apologize for these oversights. Following your suggestion, we have added the introduction of three acronyms PCM, TTM and SBM in the main text of the paper (Line 52-53; Line 102; Line 334; Line 549-550).

**Relevant text reads (Line 52-53):** "Three commonly used methods for separating the impacts of vegetation change on catchment water yield are the paired-catchment method (PCM), the time-trend method (TTM), and the sensitivity-based method (SBM)."

**(Line 102):** "The monthly data were used for the paired-catchment method (PCM), time-trend method (TTM), …"

**(Line 334):** "By using the paired-catchment method (PCM), time-trend method (TTM), and sensitivity-based method (SBM), …"

**(Line 549-550):** "The paired-catchment method (PCM) is not invalidated by the non-stationarity induced by multiyear drought because of the role of the control catchment. However, the essence of the time-trend method (TTM) and the sensitivity-based method (SBM) is to separate runoff changes caused by non-stationary…"

**R3.** Is the Pettitt (1975) method used in this work? It is mentioned once (p4 L101) where the authors state that the Mann-Kendall test is used for ranking tests of non-stationarity then never again.

**Response:** Thanks for bringing this to our attention. In the revised manuscript, what kinds of time series had been analysed using these two methods and the purposes of the applications have been mentioned in section 3.1, section 4.1 and Table 1 (Line 119-126; Line 256; Line 321).

The Pettitt (1979) method is mainly used to solve two problems in this study. One is to analyse the abrupt change points of annual rainfall, runoff and potential evapotranspiration (refer to Table 1). The abrupt change point of annual runoff is used to divide the calibration period and the prediction period. And the other one is to analyse the abrupt change points of the slope of the rainfall-runoff cumulative curve (refer to Fig. 4). The existence of the abrupt change points of the slope of the rainfall-runoff cumulative curve means that the relationship between rainfall and runoff become non-stationary.

The Mann-Kendall test (Kendall, 1975; Mann, 1945) is used to analyse the change trend (increase or decrease) of annual rainfall, runoff, and potential evapotranspiration (refer to Table 1).

**Relevant text reads (Line 119-126):** "The Mann-Kendall test (Kendall, 1975; Mann, 1945) was used to detect the long-term trend of annual rainfall, runoff, and potential evapotranspiration and the Sen's slope estimator (Sen, 1968) was used to obtain the degrees of above changes. If the $Z$ value estimated by the Mann-Kendall test method is less than zero, it indicates a downward trend; on the contrary, if the $Z$ value is greater than zero, it indicates an upward trend. $\beta$ estimated by the Sen's method represents the slope of the change trend. The Pettitt method (Pettitt, 1979) is a rank-based nonparametric statistical test method and is used to detect abrupt change points of annual rainfall, runoff, potential evapotranspiration records and the rainfall and runoff cumulative curves. The abrupt change point of annual runoff is used to divide the calibration period and the prediction period. The Mann-Kendall and Pettitt methods are the most frequently used statistical methods for identification of changes in hydro-meteorological data (Peng et al., 2020)."

**(Line 256):** "Two change points estimated by the Pettitt method occurred in December 1996 and January 2010 in the Red Hill catchment,…"

**(Line 321):** "[a]the change point year estimated by the Pettitt method."

**R4.** Question of equilibration between rainfall-runoff process within catchment and between paired catchments? The catchments must be small enough and the changes of a suitable scale that either both equilibrate quickly, or at the same rate so that cumulative fluxes still appear as a straight line.

**Response:** Thanks for your constructive comment. In the revised manuscript, the principle and the purposes of the applications of double mass curve have been described in more detail in section 3.1 (Line 128-133).

The double mass curve is the simplest, but is most intuitive and most widely used method to analyse the stationarity or multiyear evolution trend of hydrometeorological variables. It is true that cumulative fluxes can still appear as a straight line when both hydrometeorological variables equilibrate quickly, or at the same rate under the condition of stationary changes. It can be seen from the straight line of the period before October 2001 in Fig. 4 (b). However, when catchments are affected by non-stationary changes (such as multiyear drought, vegetation change, etc.), the amount and rate of changes to reach new equilibrium of both hydrometeorological variables (rainfall-runoff process within catchment or runoff-runoff process between paired catchments) will be different. For example, the same rainfall will lead to less runoff after multiyear drought and afforestation (refer to Fig. 4 (a)). These different changes will lead to the abrupt change points of the double mass curve, and there are different equilibrium states before and after the abrupt change points (the slope of curve during different periods is changed). The double mass curve in this study is used to explore the impact of multiyear drought and vegetation change on the non-stationary changes of rainfall-runoff relationship of paired catchments.

**Relevant text reads (Line 128-133):** "The DMCs plot the accumulated values of one variable against the accumulated values of another related variable for a concurrent period (Searcy and Hardison, 1960; Wang et al., 2013). It can still appear as a straight line when both hydrometeorological variables (rainfall and runoff) equilibrate quickly, or at the same rate under the condition of stationary changes. A break in the slope of the DMCs detected by the Pettitt method means that a change in the constant of proportionality between rainfall and runoff has occurred. The difference in the slope of the lines

indicates the shift in the rainfall-runoff relationship and the degree of change in the relation."

**R5.** Do the authors think that having a single effect occur with reasonable gaps is necessary to analyse the data? In the case of Red Hill/Kileys Run the catchments were paired (hydrologically) well then had the afforestation and years of data, then the drought with years of data, then the post-drought conditions and again with years of data. Is there a risk if changing climate conditions inducing a non-stationary response would interfere with the land-use response if they occurred closely chronologically? Could a method determine the changes and separate them?

**Response:** Thanks for your constructive comments. We have added discussions regarding to your concerns in the revised manuscript (Line 470-483).

We agree that one of investigated effects (i.e. multiyear drought) was occurred part of the study period (2000 – 2009). We think it is still necessary to analysis the impacts of drought as we have demonstrated that multiyear drought has induced rainfall-runoff relationship changes in the control catchment. Negligence of such effect can result in significant differences in estimated effects of vegetation change as shown in Fig. 6, which is also reported by Zhao et al. (2010) using about 16 years data.

Only two years of rainfall was above the average after 2000 and drought lasted from 2000 to 2009 in the Red Hill paired catchments. After 2009 (post drought), the slope of double mass curve is still very close to that during the multiyear drought period. Thus the single effects of drought are considered to have similar effects as vegetation treatment and are evaluated between two periods (i.e. pre- and post-change periods).

We agree the risks of interferences between drought and land-use response are existed. Several studies have reported that not only land use types but also soil and catchment properties may lead to different effects of drought on runoff (Saft et al., 2015; van Dijk et al., 2013). Here, one of the assumptions of the new framework is that the effects of three factors (vegetation change, hydroclimatic non-stationarity and climate variability) are independent of each other. We have to make this assumption to enable us to separate three effects with the help of paired catchments. The sum of the contribution of three factors to runoff change is 111.4% in the Red Hill catchment and it is close to 100%, which proves that the assumption is basically reasonable and valid. However, considering the complexity of the interaction amongst different factors and the values that are difficult to quantify, the new framework cannot separate the interaction of three factors under the current experimental conditions and data. The research about how to estimate the interactions amongst different factors need to be carefully observed and investigated in the future.

**Relevant text reads (Line 470-483):** "Negligence of non-stationarity induced by multiyear drought can result in significant differences in estimated effects of vegetation change as shown in Fig. 6, which has also been reported by Zhao et al. (2010) using about 16 years data at the same site. In the new framework, effect of multiyear drought is estimated between pre- and post-change periods as that of vegetation change, although two years of rainfall is above the average after 2000. Because the slope of DMCs is still very close to that during the period after 2009 (post-drought) (see Fig. 4). Interactions between the impact of prolonged drought and that of land-use change may be existed. Several studies have reported that not only land use types but also soil and catchment properties may lead to different effects of drought on runoff (Saft et al., 2015; van Dijk et al., 2013). Here, one of the assumptions of the new framework is that the effects of three factors (vegetation change, hydroclimatic non-stationarity and climate variability) are independent of each other. We have to make this assumption to enable us

to separate three effects with the help of paired catchments. The sum of the contribution of three factors to runoff changes is 111.4% in the Red Hill catchment, which is close to 100% and proves that the assumption is basically reasonable and valid. Considering these complex and secondary interactions amongst different factors, the new framework cannot separate them under the current experimental design and available data. How to estimate the interactions amongst different factors need to be carefully observed and investigated in the future."

**References:**

Kendall, M. G.: Rank-Correlation Measures, Charles Griffin, London, 202 pp., 1975.

Mann, H. B.: Nonparametric tests against trend, Econometrica, 13, 245-259, https//doi.org/10.2307/1907187, 1945.

Peng, T., Tian, H., Singh, V. P., Chen, M., Liu, J., Ma, H., and Wang, J.: Quantitative assessment of drivers of sediment load reduction in the Yangtze River basin, China, J. Hydrol., 580, 124242, https//doi.org/https://doi.org/10.1016/j.jhydrol.2019.124242, 2020.

Pettitt, A. N.: A non-parametric approach to the change-point problem, Journal of the Royal Statistical Society: Series C (Applied Statistics), 28, 126-135, https//doi.org/10.2307/2346729, 1979.

Saft, M., Western, A. W., Zhang, L., Peel, M. C., and Potter, N. J.: The influence of multiyear drought on the annual rainfall-runoff relationship: An Australian perspective, Water Resour. Res., 51, 2444-2463, https//doi.org/10.1002/2014WR015348, 2015.

Searcy, J. K., and Hardison, C. H.: Double-mass Curves, United states government printing office, Washington, 65 pp., 1960.

Sen, P. K.: Estimates of the Regression Coefficient Based on Kendall's Tau, J. Am. Stat. Assoc., 63, 1379-1389, https//doi.org/10.1080/01621459.1968.10480934, 1968.

van Dijk, A. I. J. M., Beck, H. E., Crosbie, R. S., de Jeu, R. A. M., Liu, Y. Y., Podger, G. M., Timbal, B., and Viney, N. R.: The Millennium Drought in southeast Australia (2001-2009): Natural and human causes and implications for water resources, ecosystems, economy, and society, Water Resour. Res., 49, 1040-1057, https//doi.org/10.1002/wrcr.20123, 2013.

Wang, W., Shao, Q., Yang, T., Peng, S., Xing, W., Sun, F., and Luo, Y.: Quantitative assessment of the impact of climate variability and human activities on runoff changes: a case study in four catchments of the Haihe River basin, China, Hydrol. Process., 27, 1158-1174, https//doi.org/https://doi.org/10.1002/hyp.9299, 2013.

Zhao, F., Zhang, L., Xu, Z., and Scott, D. F.: Evaluation of methods for estimating the effects of vegetation change and climate variability on streamflow, Water Resour. Res., 46, https//doi.org/10.1029/2009WR007702, 2010.

**Response to Reviewer #2**

**General comments**

**R1.** In general it's clear that the authors improved the original manuscript (https://doi.org/10.5194/hess-2021-5). They did improved the flow of the storyline and the analysis of the control catchment. The manuscript has appropriate objectives and gives insight in the change in hydrological processes caused by the combination of land use and climate changes.

**Response:** Thanks very much for your great efforts to assess our manuscript. We have studied your and reviewers' comments carefully and have made corrections/revisions as suggested. The point-to-point responses to the comments and our plans for revision are listed below. In the following, we have detailed how these comments (in black) are raised and our responses (in deep sky blue).

**R2.** Although the assumptions and fallowing conclusions are constructively fine and according to the applied methods, the arguments (especially in the abstract and conclusion) are not very clear. Although some comparative values per method are presented, for a reader it is not clear where the conclusions are based on.

**Response:** Thanks for bringing this to our attention. More specific descriptions about where the conclusions are based on have been mentioned in the mew manuscript (Line 22-24; Line 289-290; Line 347-351; Line 355-357; Line 547-548; Line 553; Line 555-556; Line 556-557).

**Relevant text reads (Line 22-24):** "Based on the new framework, impacts of multiyear drought and climate variability on runoff of the control catchment (Kileys Run) were 87.2% and 12.8%, respectively. Impacts of afforestation, multiyear drought and climate variability on runoff of the treated catchment (Red Hill) were 32.8%, 54.7% and 23.9%, respectively."

**(Line 289-290):** "The runoff coefficient decreased by 87.8% and 63.3% during the drought period in the Red Hill and Kileys Run catchments, respectively."

**(Line 347-351):** "For the Kileys Run catchment, runoff change induced by multiyear drought ($\overline{\Delta Q_c^n}$) is $-110.2$ mm, and runoff change caused by climate variability ($\overline{\Delta Q_c^{clim}}$) by subtracting runoff change caused by multiyear drought from the total runoff changes ($\overline{\Delta Q_c^{total}} = -126.4$ mm) is $-16.2$ mm. Impacts of afforestation, multiyear drought and climate variability on runoff of the Red Hill catchment are 32.8%, 54.7% and 23.9%, respectively, and impacts of multiyear drought and climate variability on runoff of the Kileys Run catchment are 87.2% and 12.8%, respectively."

**(Line 355-357):** "Estimated impacts of afforestation on runoff decreased greatly from 93.5% to 38.8% (=93.5%-54.7%) calculated by the TTM and decreased greatly from 76.1% to 21.4% (=76.1%-54.7%) by the SBM."

**(Line 547-548):** "The runoff coefficient decreased by 87.8% and 63.3% during the drought period in the Red Hill and Kileys Run catchments, respectively."

**(Line 553):** "Estimated afforestation impacts were 32.8%, 93.5%, and 76.1% of total runoff changes by the PCM, TTM and SBM, respectively."

**(Line 555-556):** "Impacts of afforestation, multiyear drought, and climate variability on runoff of the

treated catchment (Red Hill) were 32.8%, 54.7% and 23.9%, respectively."

**(Line 556-557):** "The contribution of vegetation change to runoff reduction using the three methods under the new framework become consistent (32.8%, 38.8% and 21.4%)."

**Specific comments**

**R3.** The abstract ends with "paired-catchment method is proven to be still the most reliable method even the control catchment experienced climate-induced shift in rainfall-runoff relationship", but within the abstract no information on the control catchments are presented. I would suggest to give actual numbers so that the reader is able to draw this conclusion himself.

**Response:** Thanks for your suggestions. We have added the results about the control catchment in abstract section (Line 22-23).

**Relevant text reads (Line 22-23):** "Impacts of multiyear drought and climate variability on runoff of the control catchment (Kileys Run) were 87.2% and 12.8%, respectively."

**R4.** In my opinion it's a missed opportunity that the majority of input by the previous reviewers and especially the replies by the authors are not processed within this manuscript.

e. Response to reviewer 1 (https://doi.org/10.5194/hess-2021-5-AC1): "During 1988 the uppermost 50 ha of Red Hill was planted top. radiata, with the remaining 145 ha planted to p. radiata in April 1989. During 2003 the plantations in Red Hill catchment were thinned to remove pulpwood and to promote the growth of sawlogs in the remaining stands. During the prolonged drought period, no trees died in the treated catchment. The change of annual PET can be seen in Figure 8 (b) (Page 42). PET showed an insignificant (p-value>0.1) increasing trend of 3.5 mm year-1. PET initially decreased before 1996 and then increased after 1996. The mean annual PET during the period of 1990-1996, 1997-2009, 2010-2015 and 1997-2015 are 1168 mm, 1262 mm, 1186 mm and 1238 mm, respectively. Compared with the period of 1990-1996 and 2010-2015, the mean annual PET during the period of 1997-2009 (the prolonged drought occurred) increased by 94 mm and 76 mm, respectively. Compared with the period of 1990-1996, the mean annual PET during the period of 1997-2015 increased by 70 mm. It is consistent with the cognition that afforestation and drought can make PET increase."

e. reviewer 2 (https://doi.org/10.5194/hess-2021-5-AC2), which you may use in the introduction and/or discussion: "Considering the influence of prolonged drought on the rainfall-runoff relationship in the treated catchment, the result of the paired catchment method is closest to the real runoff change caused by vegetation change. Because the control catchment indirect eliminate the influence of prolonged drought and climate variability on the treated catchment under the assumption that the response of the two catchments to prolonged drought is similar. Interannual changes in watershed storage occur primarily in soil water and shallow groundwater, pools that are often hydrologically active at time scales shorter than 1 year (Sayama et al., 2011). Rice and Emanuel (2019) indicates that down-regulation of transpiration and inhibition of hydrologic connectivity by forest vegetation represent two important negative feedback processes that can avert large losses in soil water or plant-accessible groundwater during dry periods. In doing so, this feedback mechanism has the potential to reinforce steady-state (or near-steady-state) conditions in dry conditions. So, the sensitivity-based method may

be less affected by the prolonged drought because it is used in the forest and the time scale of PET and P data used in this method is annual scale. Runoff changes calculated by the sensitivity-based method are induced by climate variability."

**Response:** We apologize for these oversights, and thanks very much for your reminder. We have added more detailed information about the characteristics of paired catchments, data and the history of vegetation change treatment in section 2 (Line 84-109). The analysis of changes in potential evapotranspiration and rainfall have been mentioned in section 5.1 (Fig. 7; Line 394-402). More specific analysis of the application of traditional methods have been added in section 5.1 (Line 376-383; Line 389-409).

[revised manuscript text omitted]

**R5.** For some statements I suggest to add some additional more recent papers.

e. at line 33.

e. at lines 52-53 "in many catchments around the world" two papers are cited one from 1980 and one from 2005, may be implying that these methods are not used that often anymore?

**Response:** Thanks for your suggestions. Changes have been made as suggested. (Line 36-38; Line 56-58, see references below).

**Relevant text reads (Line 36-38):** "However, separating the effects of vegetation change and climate variability on runoff remains a great challenge due to the complex interactions between climate variability and vegetation change (Cavalcante et al., 2019; Jones et al., 2006; Zhang et al., 2021)."

**(Line 56-58):** "This method has been applied in many paired catchments around the world to provide fundamental understanding and knowledge for water resource management under vegetation change (Brown et al., 2005; Stoof et al., 2012; Van Loon et al., 2019)."

Cavalcante, R. B. L., Pontes, P. R. M., Souza Filho, P. W. M., and Souza, E. B.: Opposite Effects of Climate and Land Use Changes on the Annual Water Balance in the Amazon Arc of Deforestation, Water Resour. Res., 55, 3092-3106, https//doi.org/10.1029/2019WR025083, 2019.

Zhang, J., Zhang, Y., Sun, G., Song, C., Dannenberg, M. P., Li, J., Liu, N., Zhang, K., Zhang, Q., and Hao, L.: Vegetation greening weakened the capacity of water supply to China's South-to-North Water Diversion Project, Hydrol. Earth Syst. Sci., 25, 5623-5640, https//doi.org/10.5194/hess-25-5623-2021, 2021.

Stoof, C. R., Vervoort, R. W., Iwema, J., van den Elsen, E., Ferreira, A. J. D., and Ritsema, C. J.: Hydrological response of a small catchment burned by experimental fire, Hydrol. Earth Syst. Sci., 16, 267-285, https//doi.org/10.5194/hess-16-267-2012, 2012.

Van Loon, A. F., Rangecroft, S., Coxon, G., Breña Naranjo, J. A., Van Ogtrop, F., and Van Lanen, H. A. J.: Using paired catchments to quantify the human influence on hydrological droughts, Hydrol. Earth Syst. Sci., 23, 1725-1739, https//doi.org/10.5194/hess-23-1725-2019, 2019.

**R6.** Lines 79-93: The data and location description are very brief. I suggest to give some more extended information. i.e. It can be seen that Kileys Run experienced a multiyear drought that lasted … with the period of the Millennium Drought", explain why? What is lowest amount of rainfall measured?

**Response:** Thanks for your advice. We have added more extended information about paired catchments and the determination of multiyear drought in the revised manuscript (Line 84-95; Line 96-109).

**Relevant text reads (Line 84-95):** "The Red Hill catchment (1.95 km$^2$) and the Kileys Run catchment (1.35 km$^2$) were paired catchments located 23 km northeast of Tumut and 100 km west of Canberra in New South Wales, Australia (35.322°S, 149.137°E) (Fig. 1). The catchments are adjacent, and the soil texture, topographic characteristics, and climatic conditions are similar. The altitude of the two catchments ranges from 590 m to 835 m above sea level. The slope in the lower part of catchments is mostly gentle, and gradually increases towards the ridge in a convex form. Geology of the Red Hill catchment is predominately Young granodiorite, while the Kileys Run catchment is dominated by Alkali diorite. Valley floor, midslope yellow duplex, shallow red soils and upslope red duplex are four main soil types in these two catchments. Upslope red duplex soils has the highest saturated hydraulic conductivities and valley

floor soils has the lowest saturated hydraulic conductivities (Major et al., 1998). The climate of these two catchments is temperate with highly variable and winter-dominated rainfall. In 1988, P. radiata was planted in the Red Hill catchment (0.5 km$^2$), and the remainder (1.45 km$^2$) was planted in April 1989. By 1997, pine occupied 78% of the Red Hill catchment. During multiyear drought period, no trees died in the treated catchment (Bren et al., 2006). The neighboring catchment (Kileys Run) was the control catchment, which has been maintained as a grazed pasture control over the entire observation period (Webb and Kathuria, 2012)."

**(Line 96-109):** "Daily rainfall and runoff from these two catchments were collected during the period of 1990–2015. The daily rainfall was measured by tipping bucket rain gauges had been located at catchment outlet and the daily runoff was measured by a flat-v style crump weir at a gauging station at the outlet of each catchment (Major et al., 1998). Mean annual rainfall and mean annual runoff of the Red Hill catchment were 817 mm and 75 mm, respectively, during the study period. Mean annual rainfall and runoff were 817 mm and 161 mm, respectively, in the Kileys Run catchment over the same period. Monthly potential evapotranspiration (PET) records were obtained from the SILO Data (www.longpaddock.qld.gov.au/silo/point-data/). The daily data were only used for the analysis of flow duration curves (FDCs). The monthly data were used for the paired-catchment method (PCM), time-trend method (TTM), the new framework and the analysis of double mass curves (DMCs). The annual data are used in the sensitivity-based method (SBM). Figure 2 shows the change of rainfall anomaly (%) in the Kileys Run and Red Hill catchments. Rainfall anomaly (%) is defined as the percentage deviation of annual rainfall to mean manual rainfall. It can be seen that three-year moving average of the rainfall anomaly (the black line) is lower than zero from 2000 to 2009. According to the method of determining multiyear drought period proposed by Saft et al. (2015), two catchments experienced prolonged drought lasted 10 years from 2000 to 2009 and this coincided with the period of the Millennium Drought of Australia (van Dijk et al., 2013). The minimum measured annual rainfall from 1990 to 2015 were 388.6 mm."

**R7.** Lines 346-357, the paragraph about "Multiyear drought induced changes in the rainfall-runoff relationship" is short and doesn't discuss the subject. I suggest to compare your results with other locations where multiyear drought led to changes in the rainfall-runoff relationship.

**Response:** Thanks for your suggestions. We have added a few sentences to discussion the subject that multiyear drought induced changes in the rainfall-runoff relationship in section 5.2 (Line 414-427).

It can be seen that multiyear drought has led to significant changes in the rainfall-runoff relationship in Fig. 4 and 5, which is similar to the significant downward shift of rainfall-runoff regression lines in basins in southeast Australia, the United States and China (Avanzi et al., 2020; Saft et al., 2015; Tian et al., 2018). In this study, the runoff coefficient decreased by 87.8% and 63.3% during drought period in the Red Hill and Kileys Run catchment, respectively. The latter was close to the decrease of runoff coefficient of 65.8% in Texas caused by extreme drought (Allen et al., 2011). T Runoff coefficient decrease of the Red Hill catchment was higher than that of the Kileys Run catchment because runoff of the Red Hill catchment was also affected by afforestation, which can increased annual evaporation and decreased streamflow (Cheng et al., 2017; Hoek Van Dijke et al., 2022; Wang-Erlandsson et al., 2018). This will lead to more runoff reduction than predicted based on the rainfall-runoff relationship established in pre-drought period, and ignoring the impact of non-stationary change may cause large errors in the results (Zhao et al., 2010). Compared with the line during drought period, the rainfall-runoff regression line moved up after the drought due to heavy rainfall in 2010, but it did not completely return to the state before the

drought. Peterson et al. (2021) suggested that these changes may be due to water loss from increased transpiration during drought period.

**Relevant text reads (Line 414-427):** "According to the results in session 4.2, multiyear drought has led to shift in the rainfall-runoff relationship of paired catchments, which is similar to the significant downward shift of rainfall-runoff regression lines in basins in southeast Australia, the United States and China (Avanzi et al., 2020; Saft et al., 2015; Tian et al., 2018), and the increases in zero flow days with low flows being more affected than high flows of FDCs in 10 catchments from southeastern Australia, New Zealand and South Africa (Lane et al., 2005). In this study, the runoff coefficient decreased by 87.8% and 63.3% during the drought period in the Red Hill and Kileys Run catchments, respectively. The latter was close to the decrease of runoff coefficient of 65.8% in Texas caused by extreme drought (Allen et al., 2011). Runoff coefficient decrease of the Red Hill catchment was higher than that of the Kileys Run catchment because runoff of the Red Hill catchment was also affected by afforestation, which can increased annual evaporation and decreased streamflow (Cheng et al., 2017; Hoek Van Dijke et al., 2022; Wang-Erlandsson et al., 2018). Multiyear drought can lead to more runoff reduction than predicted based on the rainfall-runoff relationship established during pre-drought period as ignoring the impact of non-stationary changes may cause large errors in the results (Zhao et al., 2010). Compared with the line during drought period, the rainfall-runoff regression line moved up after multiyear drought due to heavy rainfall of 2010, but it did not return to the state before multiyear drought. Peterson et al. (2021) suggested that these changes may be due to severe water loss from transpiration during drought period."

**R8.** In this paragraph you do mention a change in physical processes (runoff, soil moisture and evapotranspiration?), but not really compared with studies elsewhere. Do you have any specific evidence available about changes processes?

**Response:** Thanks for your constructive comments. Changes have been made as suggested. More comparisons with studies elsewhere and descriptions of changes in annual lowest seven-day flow (it reflects the storage state of groundwater to a certain extent) have been added in section 5.2 (Line 428-441). Because only rainfall, runoff and potential evapotranspiration data can be obtained, the conjecture about the reasons is based on the analogy of similar catchments also affected by multiyear drought in Australia.

For precipitation input, inter-annual rainfall variability decreased and high rainfall years were missing during the drought period in Fig. 2. Similar changes also occurred in 124 watersheds in Australia during drought period (Saft et al., 2015). The reduction of rainfall input reduces runoff at the source. For runoff process, in the Kileys Run and Red Hill catchments, rainfall primarily occurs in autumn and winter, less rainfall in autumn may result in lower antecedent soil moisture, which means more precipitation were used to replenish the soil water deficit in winter (Fig. 8). The changes of the monthly averages of rainfall and runoff were very similar to that of Murray Darling Basin, where rainfall-runoff relationship has changed caused by multiyear drought (Potter et al., 2010). As a result, runoff in winter during drought period was less than that during pre-drought period and the decrease of rainfall in next spring further aggravated runoff reduction. The decrease of GRACE satellite-observed average monthly terrestrial water storage and estimated groundwater storage in Murray–Darling Basin may support the above speculation about runoff reduction (van Dijk et al., 2013). The decline in groundwater levels may also be the reason for runoff reduction. Decline in precipitation usually resulted in a decline in groundwater levels (Peters et al., 2003), and may cause the connection between groundwater and surface water to be disrupted (Kinal and Stoneman, 2012). Brutsaert (2008) demonstrated that annual lowest seven-day

flow can be used to indicate the change of ground water storage in the absence of observations of groundwater level. The annual lowest seven-day flow in the Kileys Run catchment generally declined from 1990 to 1999, and was reduced to zero from 2001 to 2010.

**Relevant text reads (Line 428-441):** "Inter-annual rainfall variability decreased and high rainfall years were missing during the drought period (see Fig. 2). Similar changes were also reported in 124 watersheds in Australia during the drought period (Saft et al., 2015). The reduction of rainfall reduced runoff. In the Kileys Run and Red Hill catchments, rainfall primarily occurred in autumn and winter, less rainfall in autumn may resulted in lower antecedent soil moisture, which means more precipitation were used to replenish the soil water deficit in winter (Fig. 8). As a result, runoff in winter during drought period was less than that during pre-drought period and the decrease of rainfall in next spring further aggravated runoff reduction. It was consistent with less runoff during the second period under the same rainfall in Fig. 4. The decrease of GRACE satellite-observed average monthly terrestrial water storage and estimated groundwater storage in Murray–Darling Basin may support the above speculation about runoff reduction (van Dijk et al., 2013). The decline in groundwater levels may also be the reason for runoff reduction. Decline in precipitation usually resulted in a decline in groundwater levels (Peters et al., 2003), and may cause disconnection between groundwater and surface water (Kinal and Stoneman, 2012). Brutsaert (2008) demonstrated that annual lowest seven-day flow can be used indirectly to indicate the change of ground water storage. The annual lowest seven-day flow in the Kileys Run catchment generally declined from 1990 to 1999, and was reduced further to zero from 2001 to 2010, suggesting ground water storage have dried up for a long time during multiyear drought."

**R9.** In addition to that, add some references about global and local knowledge about land use changes and their effects, such as the infiltration trade-off hypothesis (i.e. Bruijnzeel, 1989) and regional water availability caused by tree restoration such as (i.e. Hoek van Dijke, et al. 2022).

**Response:** Thanks for your advice. Changes have been made as suggested. (Line 420-422, see references below).

**Relevant text reads (Line 420-422):** "Runoff coefficient decrease of the Red Hill catchment was higher than that of the Kileys Run catchment because runoff of the Red Hill catchment was also affected by afforestation, which can increase annual evaporation and decrease streamflow (Bruijnzeel, 1989; Cheng et al., 2017; Hoek Van Dijke et al., 2022)."

Bruijnzeel, L. A.: Forestaion and dry season flow in the tropics: a closer look, J. Trop. For. Sci., 1, 229-243, 1989.

Hoek Van Dijke, A. J., Herold, M., Mallick, K., Benedict, I., Machwitz, M., Schlerf, M., Pranindita, A., Theeuwen, J. J. E., Bastin, J., and Teuling, A. J.: Shifts in regional water availability due to global tree restoration, Nat. Geosci., 15, 363-368, https//doi.org/10.1038/s41561-022-00935-0, 2022.

Cheng, L., Zhang, L., Chiew, F. H. S., Canadell, J. G., Zhao, F., Wang, Y., Hu, X., and Lin, K.: Quantifying the impacts of vegetation changes on catchment storage-discharge dynamics using paired-catchment data, Water Resour. Res., 53, 5963-5979, https//doi.org/10.1002/2017WR020600, 2017.

**R10.** Lines 412-426, I suggest to re-introduce abbreviations.

**Response:** Thanks for your advice. The words have been modified to "paired-catchment method (PCM)",

"time-trend method (TTM)" and "sensitivity-based method (SBM)" (Line 548; Line 549).

**R11.** Lines 417-418, conclusion. Add numbers so the reader is able to "agree" with your conclusion.

**Response:** Thanks for your suggestions. We have added a few sentences to support our conclusion in the new manuscript (Line 547-548; Line 553; Line 555-556; Line 556-557).

**Relevant text reads (Line 547-548):** "The runoff coefficient decreased by 87.8% and 63.3% during the drought period in the Red Hill and Kileys Run catchments, respectively."

**(Line 553):** "Estimated afforestation impacts were 32.8%, 93.5%, and 76.1% of total runoff changes by the PCM, TTM and SBM, respectively."

**(Line 555-556):** "Impacts of afforestation, multiyear drought, and climate variability on runoff of the treated catchment (Red Hill) were 32.8%, 54.7% and 23.9%, respectively."

**(Line 556-557):** "The contribution of vegetation change to runoff reduction using the three methods under the new framework become consistent (32.8%, 38.8% and 21.4%)."

**R12.** Idem line 420.

**Response:** Thanks for your suggestions. The lines have been modified to "The contribution of vegetation change to runoff reduction using the three methods under the new framework become consistent (32.8%, 38.8% and 21.4%)." (Line 556-557).

**Technical corrections**

**R13.** Line 82, what about Red Hill?

**Response:** Thanks for bringing this to our attention and more information about the Red Hill catchment have been mentioned in the new manuscript (Line 88-91).

Geology of the Red Hill catchment is predominately Young granodiorite, while the Kileys Run catchment also has Young granodiorite but is dominated by Alkali diorite. The ridges and upper slopes of each catchment have predominantly given rise to red duplex soils that comprise dark brown organic loam and silty loam overlying light to medium reddish-brown clay. In Red Hill catchment lower slopes comprise sandy soils consisting of dark grey-brown organic loam/silty loam overlying grey-brown bleached and heavily cemented loamy silt grading to yellow-brown or grey-brown mottled medium-heavy clay at depths > 45 cm. This soil type is absent from the Kileys Run catchment and mean saturated hydraulic conductivities were 42.84 cm/day, 11.12 cm/day and 6.62 cm/day at depths of 20–50 cm, 50–80 cm and 80–100 cm, respectively.

**Relevant text reads (Line 88-91):** "Geology of the Red Hill catchment is predominately Young granodiorite, while the Kileys Run catchment is dominated by Alkali diorite. Valley floor, midslope yellow duplex, shallow red soils and upslope red duplex are four main soil types in these two catchments. Upslope red duplex soils has the highest saturated hydraulic conductivities and valley floor soils has the lowest saturated hydraulic conductivities (Major et al., 1998)."

**R14.** Line 87, it would be nice to apply your analysis with data up to 2020?

**Response:** Thanks for bringing this to our attention. Unfortunately, we do not have measured runoff and rainfall data up to 2020. We can obtain the annual grid rainfall from 1990 to 2020, the average value is 940.0 mm. Based on the analysis of the grid rainfall anomaly from 1980 to 2020 and from 1990 to 2020, the two series both experienced multiyear drought from 2001 to 2009 (Fig. R1), which is consistent with the period of multiyear drought in the manuscript (the drought period is from 2000 to 2009). There is an alternate change of wet and dry after 2015, especially, a large negative rainfall anomaly (%) is existed after 2017. Therefore, the extended data may not greatly improve the results.

[Figure]

Figure R1 Rainfall anomaly (%) as a percentage of the mean annual rainfall of Kileys Run and Red Hill catchment. (a) Grid rainfall data (1980-2020); (b) Grid rainfall data (1990-2020). Red bars represent dry years and blue bars represent wet years. The black line represents the three-year moving average of the rainfall anomaly.

**R15.** Line 87, which method did you used to measure the runoff? Of where did you collected the data?

**Response:** Thanks for bringing this to our attention and the source of the data have been mentioned in the revised manuscript (Line 96-98).

Runoff data were measured on site by government agency and were collected by Dr. Lu Zhang from CSIRO Land and Water, Black Mountain, Canberra, Australia. Observations were made at the outlet of each catchment, a gauging station was installed and began operation on 26 May, 1989. Daily runoff data are therefore available from June 1989 onwards. The gauging stations comprise a flat-v style crump weir at the Red Hill and Kileys Run catchments. The rating curves (stage-discharge relationships) for the Red

Hill and Kileys Run catchments had been developed and tested by a series of velocity-area gauging and, as a result, the runoff data from those stations was considered reliable (Webb and Kathuria, 2012).

**Relevant text reads (Line 96-98):** "The daily rainfall was measured by tipping bucket rain gauges had been located at catchment outlet and the daily runoff was measured by a flat-v style crump weir at a gauging station at the outlet of each catchment (Major et al., 1998)."

**R16.** Lines 88-89, add more information/background. You use daily rainfall and runoff, but monthly PET?

**Response:** Thanks for your advice. More information about data have been added in section 2 (Line 101-104).

The daily rainfall and runoff data were measured on site by government agency and were collected by Dr. Lu Zhang from CSIRO Land and Water. The potential evapotranspiration (PET) data were monthly data from the SILO Data (www.longpaddock.qld.gov.au/silo/point-data/). The daily data were only used for the analysis of flow duration curve. The monthly data were used for the PCM, TTM, the new framework and double mass curve. The annual data were used in the SBM.

**Relevant text reads (Line 101-104):** "The daily data were only used for the analysis of flow duration curves (FDCs). The monthly data were used for the PCM, TTM, the new framework and double mass curves (DMCs). The annual data are used in the SBM."

**R17.** Line 90, "figure 2 shows the rainfall anomaly that was calculated by the method proposed by", I suggest to describe the method (define anomaly). Which in this case is the annual percentage of rainfall being larger or less than the average rainfall.

**Response:** Thanks for your advice. Changes have been made as suggested in section 2 (Line 104-108).

Rainfall anomaly (%) is defined as the percentage deviation of annual rainfall to mean manual rainfall. The period of drought is determined by rainfall anomaly (%) smoothed with a three-year moving window. The method for determining the multiyear drought period is as follows (Saft et al., 2015):

The first year of the drought remains the start of the first three-year negative anomaly period;

The end year is set as the last year of this three-year negative anomaly period (unless: if the last two years have slightly positive anomalies (but each <15% of the mean), in which case the end year is set to the first year of positive anomaly);

The length of dry periods must be not less than seven years;

The mean dry period anomaly must be less than −5%;

**Relevant text reads (Line 104-108):** "Rainfall anomaly (%) is defined as the percentage deviation of annual rainfall to mean manual rainfall. It can be seen that three-year moving average of the rainfall anomaly (the black line) is lower than zero from 2000 to 2009. According to the method of determining multiyear drought period proposed by Saft et al. (2015), two catchments experienced prolonged drought lasted 10 years from 2000 to 2009 and this coincided with the period of the Millennium Drought of Australia (van Dijk et al., 2013)."

**R18.** Line 91-92, I suggest add to add the reference again.

**Response:** Thanks for your advice. Changes have been made as suggested. (Line 108-109, see references below).

**Relevant text reads (Line 108-109):** "this coincided with the period of the Millennium Drought of Australia (Peterson et al., 2021; van Dijk et al., 2013)."

van Dijk, A. I. J. M., Beck, H. E., Crosbie, R. S., de Jeu, R. A. M., Liu, Y. Y., Podger, G. M., Timbal, B., and Viney, N. R.: The Millennium Drought in southeast Australia (2001-2009): Natural and human causes and implications for water resources, ecosystems, economy, and society, Water Resour. Res., 49, 1040-1057, https//doi.org/10.1002/wrcr.20123, 2013.

Peterson, T. J., Saft, M., Peel, M. C., and John, A.: Watersheds may not recover from drought, Science, 372, 745-749, https//doi.org/10.1126/science.abd5085, 2021.

**R19.** Figures 3 – 5, and 7: used colours (combination of red and green) are not very suitable for colour blind readers (HESS - Submission (hydrology-and-earth-systemsciences.net))

**Response:** We apologize for these oversights. Fig. 3 – 5, and 7 have been modified in the new manuscript (Line 220; Line 275; Line 307; Line 442, refer to Fig. 3 – 5 and 8).

**R20.** Lines 363-365, add reference?

**Response:** Thanks for your advice. Changes have been made as suggested (Line 449-450, see references below).

**Relevant text reads (Line 449-450):** "which is the essence of the limitations of traditional application (Dey and Mishra, 2017; Li et al., 2012; Zhang et al., 2019)."

Li, H., Zhang, Y., Vaze, J., and Wang, B.: Separating effects of vegetation change and climate variability using hydrological modelling and sensitivity-based approaches, J. Hydrol., 420-421, 403-418, https//doi.org/10.1016/j.jhydrol.2011.12.033, 2012.

Dey, P., and Mishra, A.: Separating the impacts of climate change and human activities on streamflow: A review of methodologies and critical assumptions, J. Hydrol., 548, https//doi.org/10.1016/j.jhydrol.2017.03.014, 2017.

Zhang, L., Nan, Z., Wang, W., Ren, D., Zhao, Y., and Wu, X.: Separating climate change and human contributions to variations in streamflow and its components using eight time‐trend methods, Hydrol. Process., 33, 383-394, https//doi.org/10.1002/hyp.13331, 2019.

**R21.** Line 372, add reference?

**Response:** Thanks for your advice. Changes have been made as suggested (Line 455-457, see references below).

**Relevant text reads (Line 455-457):** "and multiyear drought weakened the impact of vegetation change on runoff (see Table 2), which was important for us to design ecological engineering projects for

sustainable water resources management (Brodribb et al., 2020; Newman et al., 2006; Xiao et al., 2020)."

Xiao, Y., Xiao, Q., and Sun, X.: Ecological Risks Arising from the Impact of Large-scale Afforestation on the Regional Water Supply Balance in Southwest China, Sci. Rep.-UK, 10, 4150, https//doi.org/10.1038/s41598-020-61108-w, 2020.

Newman, B. D., Wilcox, B. P., Archer, S. R., Breshears, D. D., Dahm, C. N., Duffy, C. J., McDowell, N. G., Phillips, F. M., Scanlon, B. R., and Vivoni, E. R.: Ecohydrology of water-limited environments: A scientific vision, Water Resour. Res., 42, https//doi.org/https://doi.org/10.1029/2005WR004141, 2006.

Brodribb, T. J., Powers, J., Cochard, H., and Choat, B.: Hanging by a thread? Forests and drought, Science, 368, 261-266, https//doi.org/10.1126/science.aat7631, 2020.

**R22.** Figure 4, perhaps add cumulative rainfall and the separated periods.

**Response:** Thanks for your advice. Changes have been made as suggested in Fig. 4 and Line 261-263.

The rainfall data used in these two catchments are the same. The measured cumulative rainfall shown in Fig. 4 has 
[revised manuscript text omitted]

**Response to Reviewer #3**

**General Comments:**

**R1**. This is an interesting paper on an important topic. In fact it could be argued the issues discussed in this paper are absolutely fundamental in hydrology. This is particularly true in light of the way "data" is used uncritically in many studies.

**Response:** Thanks very much for your great efforts to assess our manuscript. We have studied your and reviewers' comments carefully and have made corrections/revisions as suggested. The point-to-point responses to the comments and our plans for revision are listed below. In the following, we have detailed how these comments (in black) are raised and our responses (in deep sky blue).

**R2.** The paper is generally well written and structured, although there is a need for a more careful read-through as there is some awkward syntax and grammar (particularly in the Discussion) where the quality of the writing seems to wander a little. My main questions relate to the data set used and the essential paradigm of the PCM. The Redhill site did not have a calibration period. Although the authors suggest that the first period of treatment may be thought of as non-treated in that the trees were very small and not high water using, I feel the implications of this may be important. This then connects to the PCM paradigm; that is, that the length of calibration or the approach developing the calibrations in theory should account for the type of non-stationarity that is discussed (ie. drought). Putting this another way, how can we decide what is non-stationarity and what is variability? This is particularly germane to Australian hydrology where we experience significant variability. I am not suggesting climates are stationary, but disentangling non-stationarity from variability with a relatively short data period (in climate terms) is a question.

**Response:** Thanks for your constructive comments. We have added more contents about the calibration period, non-stationarity and variability in discussion section in the revised manuscript (Line 458-474; Line 526-544).

(1) We agree with the question of the calibration period. However, for the Red Hill catchment, it was unfortunate that rainfall and runoff data before treatment were not measured. In Zhao et al. (2010), the impacts of the calibration period determined by treatment period and period before abrupt change point of annual runoff on the results at four paired catchments were compared, it was found that runoff changes caused by vegetation change were not sensitive to the different calibration periods. Moreover, considering that the runoff may not change significantly in the first few years after plantation of seedlings of *P. radiata*, we re-estimated the impact of vegetation change on runoff based on a calibration period with the data of the previous three years. The contribution of vegetation change to total runoff changes was 34.2%, and the difference with the result in the manuscript was only 1.4%. Therefore, the calibration period set in this manuscript was reasonable.

(2) In this study, we distinguish climate variability and non-stationarity by considering the influence on the rainfall-runoff relationship. For climate variability, we think that it will not lead to non-stationary changes in the rainfall-runoff relationship, that is, rainfall and runoff changes at the same rate. For non-stationarity, it will lead to non-stationary changes in the rainfall-runoff

relationship, that is, it can be reflected by the significant abrupt change point of double mass curve and the significant up/down movement of rainfall-runoff linear regression line (Avanzi et al., 2020; Li et al., 2018). For large-scale watersheds, it is difficult to detect and confirm non-stationarity from variability because of the complexity and regional differences of positive and negative fluctuations or feedback of climate. However, for the two small studied catchments, the impact of climate fluctuations is very intense, and persistent fluctuations below the average are easy to cause non-stationary changes in the rainfall-runoff relationship.

[revised manuscript text omitted]

**R3.** The calibration period issue is a vexed one as there is no longer an appetite by funding bodies to set up a paired-catchment experiment and then wait for a lengthy period before anything happens. Bren and Lane (2014, JH 519) explored this issue and proposed a method using daily flows that rather obviously increases the number of data points. Somewhat surprisingly the analysis showed that good calibrations (Nash-Sutcliffe E = 0.8) with 100 days of data, and very little improvement after 3 years. Apologies for the treatise, but I wonder if this approach might be useful in thinking about the PCM. That is, if such an analysis was performed and compared with the other analyses it might be very useful. Data could be randomly pulled out of the Kileys Run data. At the very least it should be discussed.

**Response:** Thanks for your suggestion. We have added discussions regarding to your concerns in the revised manuscript (Line 536-544).

We used the data of seven years before the abrupt change point of annual runoff and three years after the beginning of the data series to explore the change of Nash–Sutcliffe coefficient (N-S) with the increasing length of the observation set. For seven years (1990-1996), the calibration set was from 1990 to 1992 and the verification set was from 1993 to 1996 (Fig. R1 (b) and (d)). For three years (1990-1992), the calibration set was from 1990 to 1992 and the verification set was from 1993 to 1996 (Fig. R1 (a) and (c)). According to Fig. R1, it can be found that the change of N-S was close to Bren and Lane (2014). It showed that good calibration (N–S > 0.85) was achieved in about 150 days, and then maintained at the high-level value. Similar results were obtained with monthly of flows, calibration (N–S > 0.35) was achieved in about 24 months.

We used the data of the previous three years (1990-1992) as pre-treatment period, and re-estimated the effects of vegetation change on runoff by the PCM, TTM, SBM and the new framework. Impacts of vegetation change were 34.2%, 74.2% and 61.0% of total runoff changes by the PCM, TTM and SBM, respectively. The contributions of vegetation change, multiyear drought and climate variability to total runoff changes were 34.2%, 37.4% and 39.0%, respectively, using the new framework. The sum of the three terms was 110.6%, with a difference of 10.6% from 100%. When the pre-treatment period was as pre-change period of runoff (1990-1996) used in the manuscript, impacts of vegetation change were 32.8%, 93.5% and 76.1% of total runoff changes by the PCM, TTM and SBM, respectively. The contributions of vegetation change, multiyear drought and climate variability to total runoff changes were 32.8%, 54.7% and 23.9%, respectively, using the new framework. The sum of the three terms was 111.4%, with a difference of 11.4% from 100% and a difference of 0.8% from 110.6%. It also can be found that the difference in the contribution of vegetation change to total runoff changes estimated by the two different ways to determine the calibration period was only 1.4%. It showed that taking period before the abrupt change point of annual runoff as the calibration period was also reasonable considering a lower mean error.

**Relevant text reads (Line 536-544):** "It indicates that selection of the length of the calibration period may have little impact on the estimation of runoff changes caused by vegetation change before the treated catchment reaches a new equilibrium state. This issue has also been discussed in Bren and Lane (2014) and they found that runoff of paired catchments had good calibrations (Nash-Sutcliffe efficiency (N-S) = 0.8) with 100 days of data and very little improvement after three years. For Red Hill experiment site, the change of N-S is close to that reported by Bren and Lane (2014). Good calibration (N–S > 0.85) is achieved with about 150 days. Similar results are obtained with monthly flows, good calibration (N–S > 0.35) is achieved with about 24 months. It suggests that runoff of the Red Hill and Kileys Run catchments will be well calibrated with calibration period exceeds 150 days (daily data) or 24 months (monthly data). Considering longer calibration period has lower mean error, calibration period is set from beginning of available data to the time of the abrupt change of annual runoff in this study."

[Figure]

Figure R1. Mean of 10 values of the N–S coefficient of the Coefficient of Determination as a function of the number of days and months in the calibration data set for the Kileys Run and Red Hill data using the 10-fold cross-validation approach to monitor the development of the calibration. ''Rand'' refers to the data being randomized.

**Specific Comments:**

**R4.** Line 40 - there are more updated references for the research in Australian catchment behaviour that are relevant (eg Petersen et al, 2021, Science 372)

**Response:** Thanks for your advice. Changes will be made as suggested (Line 46-47, see references below).

**Relevant text reads (Line 46-47):** "It is widely known that Australia experienced multiyear drought (known as the Millennium Drought) between 1997 and 2009 (King et al., 2020; Peterson et al., 2021)."

King, A. D., Pitman, A. J., Henley, B. J., Ukkola, A. M., and Brown, J. R.: The role of climate variability in Australian drought, Nat. Clim. Change, 10, 177-179, https//doi.org/10.1038/s41558-020-0718-z, 2020.

Peterson, T. J., Saft, M., Peel, M. C., and John, A.: Watersheds may not recover from drought, Science,

372, 745-749, https//doi.org/10.1126/science.abd5085, 2021.

**R5.** L46. I think this statement about PCM and non-stationarity requires more justification. How does it not deal with the issue given that is the paradigm of PCM?

**Response:** Thanks for your constructive comments. We have added more justification about the paired-catchment method (PCM) and non-stationarity in the revised manuscript (Line 376-383).

At Red Hill site, the treated catchment suffered from non-stationary changes caused by vegetation change and multiyear drought, and stationary changes caused by climate variability. The control catchment suffered from non-stationary changes only caused by multiyear drought, and stationary changes caused by climate variability. The PCM can offset the effects on both paired catchments induced by multiyear drought and climate variability based on data of the control catchment, and the effects of vegetation change on the treated catchment can then be separated. That is, for the paradigm of PCM, shift in the rainfall-runoff relationship separated from the runoff correlation between the treated and control catchments should be caused only by the treatment of the treated catchment and effects of any other drivers can induce either stationary or non-stationary changes should be eliminated by making use of the control catchment. Therefore, the PCM is still the most reliable method compared with other methods.

**Relevant text reads (Line 376-383):** "At Red Hill experiment site, non-stationary changes of the treated catchment are caused by both vegetation change and multiyear drought, and stationary changes are caused by climate variability. Non-stationary changes of the control catchment are only caused by multiyear drought, and stationary changes are only caused by climate variability. According to the paradigm of PCM, shift in the rainfall-runoff relationship separated from the runoff correlation between the treated and control catchments should be caused only by the treatment of the treated catchment and effects of any other drivers can induce either stationary or non-stationary changes should be eliminated by making use of the control catchment. Therefore, the PCM is still the most reliable method compared with other methods and separated effect by the PCM is only caused by vegetation change (i.e., afforestation)."

**R6.** L 65+ Both TTM and SBM require lengthy records; is this an issue with these analyses at Redhill? There may be an argument to see the drought as a plus in terms of a record with wet/dry periods.

**Response:** Thanks for your constructive comments. The discussion about this question have been mentioned in the revised manuscript (Line 462-472; Line 516-525).

(1) The length of data is essentially a problem of reaching an equilibrium state for catchments. Han et al. (2020) provided a global assessment of the steady-state assumption in catchment water balance calculations for 1,057 global unimpaired catchments. Results showed that ~70% of the catchments attain steady state within 10 years. The time needed for a catchment to reach steady state showed a close relationship with climatic aridity and vegetation coverage, with arid/semiarid and sparsely vegetated catchments generally having a longer time. For small catchment, it may need a shorter time to reach steady state. It can be seen in Fig. 4 (a), the double mass curve becomes a straight line in a very short time. It demonstrates that the length of data used in this study (26 years) is acceptable.

(2) Multiyear drought mentioned in the manuscript is a drought with longer duration and greater intensity, which will cause non-stationary changes in the rainfall-runoff relationship of catchments. It is quite different from the wet/dry periods floating near the average line. The effects of multiyear drought on runoff are very significant and cannot be ignored. It is necessary to consider impacts of multiyear drought on runoff in the new framework.

**Relevant text reads (Line 462-472):** "The multiyear drought in this study refers to drought with long duration and severe intensity, which can cause non-stationary changes in the rainfall-runoff relationship of catchments as shown in Fig. 4 and discussed in section 5.2. It is quite different from the wet/dry periods fluctuating near the average line (i.e., climate variability) (Han et al., 2019). For the two small studied catchments, the impact of climate fluctuations is very intense, and persistent fluctuations below the average are easy to cause non-stationary changes in the rainfall-runoff relationship because the long-term rainfall reduction may lead to changes of catchment characteristics, that is, lose connection between surface and groundwater. However, for large-scale watersheds, it is difficult to detect and to separate non-stationarity from variability because of the complexity and regional differences of positive and negative fluctuations or feedback of climate (Clark et al., 2016; Murakami et al., 2020). Negligence of non-stationarity induced by multiyear drought can result in significant differences in estimated effects of vegetation change as shown in Fig. 6, which has also been reported by Zhao et al. (2010) using about 16 years data at the same site."

**(Line 516-525):** "It shows that the increase of data length has little effect on the estimation of runoff change caused by vegetation change after runoff of catchment experiencing vegetation change has reached a new stable equilibrium state. The time required for runoff in different catchments to reach a new equilibrium state is different. For example, the Red Hill catchment takes seven years, Australia and New Zealand have suggested that three to 10 years, or even more (18 years for an afforested catchment in Biesievlei, South Africa (Brown et al., 2005)), majority of the time is between five and 10 years (Lane et al., 2005), are required for the treated catchment to reach a reasonably stable rainfall-runoff relationship after vegetation change (Zhao et al., 2010). Han et al. (2020) provided a global assessment of the steady-state assumption in catchment water balance calculations for 1,057 global unimpaired catchments and shown that ~70% of the catchments attained steady state within 10 years. For small catchment, it may need a shorter time to reach steady state. Thus the length of data used in this study (26 years) is enough to reach a steady state."

**R7.** Lane et al. 2005 (JH) used FDCs that included Redhill – might be worth including these results as a comparison from a different method. This paper also has some estimates of time to equilibrium.

**Response:** Thanks for your advice. Changes have been made as suggested in section 5.2 and 5.4 in the new manuscript (Line 416-418; Line 520-522).

**Relevant text reads (Line 416-418):** "and the increases in zero flow days with low flows being more affected than high flows of daily flow duration curves (FDCs) in 10 catchments from southeastern Australia, New Zealand and South Africa (Lane et al., 2005)."

**(Line 520-522):** "majority of the time is between five and 10 years (Lane et al., 2005), are required for the treated catchment to reach a reasonably stable rainfall-runoff relationship after vegetation change."

**R8.** L 69 – syntax not great "issues about this hypothesis" could be improved

**Response:** We apologize for these oversights. We have modified these sentences in the new manuscript (Line 75-77).

**Relevant text reads (Line 75-77):** "However, this question has not been explored and verified, and clarifying whether multiyear drought will have an important impact on the application ability of the three widely used methods will provide a meaningful reference for ecological engineering under changing climate with frequent extremes in future."

**R9.** L 107 – should be double mass curves, FDCs etc. There are quite a few examples of this, need a careful read.

**Response:** We apologize for these oversights. We have modified them in the revised manuscript (Line 101-103; Line 127-128; Line 254; Line 461).

**Relevant text reads (Line 101-103):** "The daily data were only used for the analysis of flow duration curves (FDCs). The monthly data were used for the PCM, TTM, the new framework and double mass curves (DMCs)."

**(Line 127-128):** "Double mass curves (DMCs), flow duration curves (FDCs), and rainfall-runoff linear regression curves were employed to detect changes in the rainfall-runoff relationship caused by vegetation change and multiyear drought."

**(Line 254):** "The double mass curves (DMCs) of monthly rainfall and runoff of the two paired catchments are shown in Fig. 4 (a) and Fig. 4 (b)."

**(Line 461):** "it can be demonstrated by the significant abrupt change point on the double mass curves (DMCs)"

**R10.** L 168- See earlier general discussion. It does trouble me that a site with no calibration is used for this study. In addition, the vegetation effect is dynamic; growing from seedlings to (presumably, given there are no growth data) a closed canopy. I do wonder if this really is the best data set for such a study, or what might be gained from using more data sets.

**Response:** Thanks for your constructive comments. Except for Red Hill site, there are no paired catchments that have experienced vegetation change and multiyear drought at the same time. The use of data and the calibration period were also further discussed in section 5.4 in the new manuscript (Line 526-544). It was found that the difference in the contribution of vegetation change to total runoff changes estimated by the two different ways to determine the calibration period (the previous three years after treatment and seven years before the abrupt change point of annual runoff in the manuscript) was only 1.4%. In this study, we not only found the reasons why results of the three traditional methods were inconsistent, but also thought more deeply about the paradigm of PCM. For PCM, shift in the rainfall-runoff relationship separated from the runoff correlation between the treated and control catchments should be caused only by the treatment of the treated catchment and effects of any other drivers can induce either stationary or non-stationary changes should be eliminated by making use of the control catchment. Therefore, the PCM is still the most reliable method compared with other methods.

**Relevant text reads (Line 526-544):** "For Red Hill experiment site, the calibration period was from one year after treatment to the abrupt change point of annual runoff (1990–1996, seven years), because rainfall and runoff data before treatment were not measured. Zhao et al. (2010) compared the influences of two different methods for determining the calibration period on the estimated vegetation impacts at four paired catchment sites. One is determined by the time of treatment. The other is determined by the abrupt change point of annual runoff. It was found that runoff changes caused by vegetation change were not sensitive to different calibration periods. Considering that runoff may not change significantly during the first few years after plantation of seedlings of *P. radiata*, we re-estimated the impact of vegetation change on runoff based on a calibration period with the data of the previous three years (1990–1992). Impacts of vegetation change were 34.2%, 74.2% and 61.0% of total runoff changes by the PCM, TTM and SBM, respectively. The contributions of vegetation change, multiyear drought and climate variability to total runoff changes using the new framework were 34.2%, 37.4% and 39.0%, respectively. Comparing to those in Table 2, the difference of the contribution of vegetation change to total runoff changes was only 1.4%. It indicates that selection of the length of the calibration period may have little impact on the estimation of runoff changes caused by vegetation change before the treated catchment reaches a new equilibrium state. This issue has also been discussed in Bren and Lane (2014) and they found that runoff of paired catchments had good calibrations (Nash-Sutcliffe efficiency (N-S) = 0.8) with 100 days of data and very little improvement after three years. For Red Hill experiment site, the change of N-S is close to that reported by Bren and Lane (2014). Good calibration (N–S > 0.85) is achieved with about 150 days. Similar results are obtained with monthly flows, good calibration (N–S > 0.35) is achieved with about 24 months. It suggests that runoff of the Red Hill and Kileys Run catchments will be well calibrated with calibration period exceeds 150 days (daily data) or 24 months (monthly data). Considering longer calibration period has lower mean error, calibration period is set from beginning of available data to the time of the abrupt change of annual runoff in this study."

**R11**. Figure 3 is a great figure!

**Response:** Thanks for your comments.

**R12**. Table 2 – total flow changes would be useful. They appear later in the text but I think having totals in the table make it easier to evaluate the methods.

**Response:** Thanks for your advice. The total flow changes have been mentioned in Table 2 (Line 330).

**R13.** This also brings up another point that I don't think has been discussed properly. $Q_{clim}$ is conceptualised as the climate effect, encompassing wet and dry and mean climate inputs. I am not sure there is adequate discussion of how this does not deal with the climate issue as formulated.

**Response:** Thanks for your constructive comments. The discussion about this question have been mentioned in the revised manuscript (Line 389-394; Line 402-407).

$Q_{clim}$ in this study refers to the impact of climate variability (wet and dry spells) on runoff that does not lead to non-stationary changes in rainfall-runoff relationship. It represents the runoff changes caused by climate stability changes and it is estimated by the SBM. The SBM is derived from the Budyko framework. In the formula, the impact of climate change on runoff is estimated by rainfall and potential

evapotranspiration changes. In the Budyko curve, it is reflected in moving from one point to another point on the same curve. There is a necessary assumption in the SBM, that is, a transition from one steady state to another with no change in catchment properties (the same curve) (Roderick and Farquhar, 2011; Sun et al., 2014). When the SBM is applied to the Red Hill catchment that has experienced multiyear drought, the result is actual the impact of climate variability (without changing the catchment characters/non-stationary changes in rainfall-runoff relationship) on runoff, that is, $Q_{clim.}$ It may ignore the impact of multiyear drought on runoff and indirectly overestimate the impact of vegetation change on runoff. Because catchment properties of the Red Hill catchment may have changed due to multiyear drought (Kinal and Stoneman, 2012; Peterson et al., 2021; Saft et al., 2016; van Dijk et al., 2013), which violates the assumptions of the SBM.

**Relevant text reads (Line 389-394):** "The SBM is sourced from the Budyko framework (Budyko, 1974). It assumes that steady state of catchment water balance is fundamentally determined by water input (represented by precipitation) and energy demand (represented by potential evapotranspiration) and transition from one steady state to another without any change in catchment properties should moving on the Budyko curve (Roderick and Farquhar, 2011; Sun et al., 2014; Wang et al., 2021).Therefore, stationary changes driven by climate variability during post-treatment period can be separated by sensitivity of runoff to $P$ and PET established during the pre-treatment period."

**(Line 402-407):** "The result estimated by the SBM is the impact of climate variability (without changing the catchment characters/non-stationary changes in the rainfall-runoff relationship) on runoff, that is, $\overline{\Delta Q_t^{clim}}$. It ignored the impact of multiyear drought on runoff, which have proved to cause non-stationary changes. Recent studies have also reported that multiyear drought can cause catchment properties changes and hydrological functionings (Kinal and Stoneman, 2012; Peterson et al., 2021; Saft et al., 2016; van Dijk et al., 2013), which may violate the assumptions of the SBM."

**R14.** 5.2.1 – this paragraph brings up the interesting point (that is the subject of the Saft/Peterson/Fowler etc studies); is it the climate that is non-stationary or is the processes (obviously driven by the climate).

**Response:** Thanks for your constructive comments. We think that climate change is a combination of non-stationary changes and process. The non-stationary change may be due to the sudden and drastic changes of human activities (explosive increase of industrial activities (Bauska et al., 2015)) and/or the amount of solar energy that gets to earth (Karl and Trenberth, 2003) and other factors. Most of these non-stationary changes occur suddenly and change from one equilibrium state to another through a relatively short period, such as non-stationary change in rainfall-runoff relationship caused by multiyear drought (Fowler et al., 2018; Peterson et al., 2021; Saft et al., 2015). The process is long-term, it may keep the equilibrium state and change continuously, or show a trend change (increase or decrease) over time. For example, the air temperature shows a gradual upward trend in a longer period (years and decades), and in a shorter period (days, months and seasons), the temperature constantly fluctuates around the average temperature, which means that there will still be those days which are cool and those days which are warm (Hoegh-Guldberg et al., 2019).

**R15.** The Paragraph around Line 375 needs some rewriting, the syntax is jarring. For example "the" control..

**Response:** We apologize for these oversights. Changes have be made as suggested in the revised manuscript (Line 484-488).

**Relevant text reads (Line 484-488):** "In the new framework, the control catchment plays an irreplaceable role in estimating the impact of vegetation change and multiyear drought on runoff. Because the control catchment can eliminate the impact of climate variability and multiyear drought on runoff when the PCM is used to quantify runoff change caused by vegetation change, and the impact of multiyear drought on the treated catchment is transferred from the control catchment. The former must use the runoff data of the control catchment, and the latter needs both the rainfall and runoff data of the control catchment."

**R16.** L 388 "Because Saft.." this is a poor sentence

**Response:** We apologize for these oversights. We have modified these sentences in the manuscript (Line 499-501).

**Relevant text reads (Line 499-501):** "Saft et al. (2016) re-evaluated a wide range of factors may be responsible for the additional runoff reductions and suggested that the shifts were mostly influenced by catchment characteristics related to pre-drought climate and soil and groundwater storage dynamics but less affected by the percentage of woody cover."

**R17.** L 399 "pines" should not be italicized. P.radiata would be

**Response:** We apologize for these oversights. The word "P. radiata " have be modified to "*P. radiata*" (Line 92; Line 510; Line 531).

**Relevant text reads (Line 92):** "In 1988, *P. radiata* was planted in the Red Hill catchment (0.5 km$^2$)"

**(Line 510):** "the treated catchment was covered by *P. radiata*"

**(Line 531):** "the first few years after plantation of seedlings of *P. radiata*"

---

## Author Response (AR2)

**Response to Reviewer #1**

**R1.** The authors use three periods for each catchment, but only one of these is common, i.e. the first two periods overlap but are not coincident making their results hard to compare. There are four periods present: (i) 1990-96 pre-drought and untreated, (ii) 1997-2000 pre-drought and treated, (iii) 2001-09 in-drought and treated, and (iv) 2010-15 post-drought and treated. Table 2 and Figure 6 essentially compares pre-drought+untreated to post-drought+treated where all effects have been expressed. Do the traditional applications work consistently when comparing the pre- and in-drought conditions prior to significant changes in the post-drought rainfall-runoff relationship? It would be good to see the different methods applied to matching temporal periods of rainfall and flow data, and the gradual divergence of the attribution of vegetation and climatic effects as more effects are imposed on the paired catchments.

**Response:** Thanks very much for your great efforts to assess our manuscript. We have studied your comments carefully and have made corrections/revisions as suggested. We have detailed how these comments (in black) are raised and our responses (in deep sky blue).

(1) Thanks for your constructive comment. We agree that the treated catchment (afforestation) experienced four periods, (i) 1990–1996 pre-drought and untreated, (ii) 1997–2001 pre-drought and treated, (iii) 2002–2009 in-drought and treated, and (iv) 2010–2015 post-drought and treated. In the (i) period, runoff of the threated catchment has not been significantly affected, it can be considered as the calibration period for evaluating the impact of vegetation change on runoff. In the (ii) period, the treated catchment was affected by both vegetation change and climate variability. In the (iii) and (iv) periods, the treated catchment was affected by multiyear drought, vegetation change and climate variability, because the rainfall-runoff relationship after multiyear drought still cannot recover to that before multiyear drought (Fig. 4) and may persist such state for a long time (Peterson et al., 2021). When separating impacts of vegetation change and multiyear drought on runoff, the data of the control and treated catchments need to be used at the same period, that is to say, the same period needed to be applied to these two catchments. After comprehensively considering the principles of the three methods, we combined the (ii), (iii) and (iv) periods of the treated catchment into one period as the prediction period. That is to say, Table 2 and Fig. 6 essentially compared untreated (1990–1996) to treated (1997–2015). Runoff difference between the untreated and treated periods in the treated catchment was caused by vegetation change, climate variability and multiyear drought, and runoff difference in the control catchment was caused by climate variability and multiyear drought. The traditional applications of the PCM, TTM and SBM did not consider runoff changes caused by non-stationary rainfall-runoff relationship induced by multiyear drought, so the TTM and SBM overestimated the impact of vegetation change on runoff. In the new framework, the impact of multiyear drought on runoff can be independently quantified by combining the TTM and the data of the control catchment.

(2) We compared estimated changes between pre-drought+untreated (1990-1996) and pre-drought+treated (1997–2001) as well as between pre-drought+untreated (1990–1996) and in- and post-drought+treated (2002–2015). The contribution of vegetation change to the total runoff changes of the Red Hill catchment can be seen in Fig. R1. Impacts of afforestation on runoff were 34.3%, 65.9% and 41.5% of the total runoff changes during the period of 1997–2001 by the PCM, TTM and SBM, respectively. Impacts of afforestation on runoff were 32.4%, 100.8% and 68.4% of the total runoff changes during the period of 2002–2015. It can be seen that results of the TTM and

SBM during the period of 2002–2015 were significantly higher than those during the period of 1997–2001, while the results of PCM were close. Because multiyear drought happened in 2002–2009 and caused persistent effects in 2010–2015 had a great impact on the rainfall-runoff relationship of the Red Hill catchment, which made the TTM and SBM overestimated the impact of vegetation change on runoff more seriously. That is to say, errors of the gradual attribution of vegetation change to runoff total changes estimated by the TTM and SBM will become larger and larger as more effects are imposed on the paired catchments.

[Figure]

Figure R1: The contribution of vegetation changes to the total runoff changes of the Red Hill catchment estimated traditionally using all three methods during the pre-drought+treated period (1997–2001) and during the in- and post-drought+treated period (2002–2015).

**R2.** It is not clear what the x-axis units are in Figure 4e. The caption states "cumulative monthly rainfall" while the x-axis is labelled "N" and runs from zero to 50,000. It cannot be days passed as this implies over 100-years, or months, or even rainfall as this would mean there is more than double the net rainfall in the adjacent catchment. Please clarify the units and label.

**Response:** We apologize for this mistake. The x-axis of Fig. 4 (e) should be number of cumulative months. The total number of months of study period is 321 (12×26=321). Mistake in x-axis of Fig. 4 (e) is corrected and shown as follows.

[Figure]

Figure 4: (e) Cumulative monthly rainfall of paired catchments during the period of 1990–2015. The dashed lines represent the linear regression lines between cumulative rainfall and number of cumulative months during three different periods (January 1990 to October 2001 (purple), November 2001 to May 2010 (red), and June 2010 to December 2015 (blue)).

Cumulative monthly rainfall figure can identify the prolonged low or high rainfall periods visually. Essentially, Fig. 4 (e) shows same information as Fig. 2. Moreover, information about prolonged drought period in Fig. 2 is more obvious and clearer using the anomaly. Therefore, we would like to delete redundant Fig. 4 (e) in the revised manuscript (see Line 275).

**R3.** Both the panels in Figure 5 appear to be the same – the FDC for Kileys Run is very different in the previous version.

**Response:** We apologize for this mistake. The FDC of the Kileys Run catchment in previous version is right. Figure 5 has been modified in the revised manuscript (see Line 305).

[Figure]

Figure 5: Daily flow duration curves of (a) the Red Hill catchment (treated catchment) and (b) the Kileys Run catchment

**R4.** Why do we get to Page 16 Figure 7 before seeing that four individual years of rainfall data (1999, 2000, 2006, 2007) appear to be missing? Are these data interpolated from SILO? Are some other data used in the climate analysis, or runoff analysis? Does their presence or otherwise influence the detection of change points or the regressions and relative contributions to runoff variation?

**Response:** Thanks for your constructive comment. The rainfall data used in this study was observed on

site rather than interpolated gridded SILO data. There are missing values in both rainfall and runoff data. Both runoff and rainfall observations are missing from November 1999 to November 2000 and from October 2006 to October 2007. In order to minimize the influences of missing values on the annual total values, annual total is regarded as missing value if more than one month is missing. Thus, there are four missing data points in annual time series of rainfall.

We believe that the missing data has little impact on the final results. Two periods with missing data are just at the beginning and end of multiyear drought. Missing rainfall values should not differ significantly from the annual rainfall values during multiyear drought period. The overall trend or segmented trend during drought will not change much due to the lack of rainfall data. It is also true for annual runoff. In addition, the change point of annual runoff calculated with data including missing values was consistent with that by Zhao et al. (2010), both appeared in 1996. Based on data including missing data, estimated afforestation impacts were 31.4%, 84.7% and 64.9% of the total runoff changes during the period of 1990–2005 by the PCM, TTM and SBM, respectively. Results of Zhao et al. (2010) were 27.0%, 71.0% and 57% by the PCM, TTM and SBM, respectively. They were very close. Furthermore, same analysis was conducted based on the gridded rainfall data from SILO. Estimated afforestation impacts were 32.8%, 93.5% and 73.0% of the total runoff changes during the period of 1990–2015 by the PCM, TTM and SBM, respectively. The results were very close to results using in-situ observed rainfall as presented in the manuscript. Therefore, we believe processing of missing data has little influences on the estimated changes.

**R5.** The authors over-use the word "proved" in this paper, as they have "shown" by way of a single case study that rainfall-runoff is non-stationary but it does not invalidate PCM, for example. And in answer to the article's title, no I do not believe that the PCM is invalidated, and the method presented can account for such changes, assuming linearity of response between control and treated catchments.

**Response:** We appreciate this insightful comment. We agree with reviewer's arguments. We replaced the word "proved" with "demonstrated" or "shown" where they were suitable in the revised manuscript.

**Relevant text reads (Line 9):** "Multiyear drought has been demonstrated to cause non-stationary rainfall-runoff relationship."

**(Line 15):** "In addition to afforestation, the Red Hill paired experimental catchments have experienced a 10-year drought (2000–2009) and have been demonstrated to lead to non-stationary rainfall-runoff relationships of paired catchments."

**(Line 26):** "This study not only demonstrated that multiyear drought can induce non-stationary rainfall-runoff relationship using field observations,…"

**(Line 28):** "More importantly, it is shown that non-stationarity induced by multiyear drought does not invalidate the PCM, and PCM is still the most reliable method even the control catchment experienced climate-induced shift in the rainfall-runoff relationship."

**(Line 77):** "If this hypothesis mentioned above is demonstrated to be correct,…"

**(Line 302):** "In summary, the shape and percentage of the zero flows of FDCs in Fig. 5 further demonstrated that the relationship between rainfall and runoff of the two catchments changed significantly over the three periods,…"

**(Line 403):** "It ignored the impact of multiyear drought on runoff, which has been demonstrated to cause

non-stationary changes. "

**(Line 479):** "..., which is close to 100% and shows that the assumption is basically reasonable and valid."

**(Line 557):** "We demonstrated that the PCM was still a valid and fundamental method estimating the impact of vegetation change on runoff even the control catchment experienced hydroclimatic non-stationarity in the rainfall-runoff relationship under changing environments."

**References:**

Peterson, T. J., Saft, M., Peel, M. C., and John, A.: Watersheds may not recover from drought, Science, 372, 745-749, https//doi.org/10.1126/science.abd5085, 2021.

Zhao, F., Zhang, L., Xu, Z., and Scott, D. F.: Evaluation of methods for estimating the effects of vegetation change and climate variability on streamflow, Water Resour. Res., 46, https//doi.org/10.1029/2009WR007702, 2010.

**Response to Reviewer #2**

**Response:** Thanks very much for your great efforts to assess our manuscript. We have made technical corrections in the revised manuscript (see Table 2).

---

## Author Response (AR3)

**Response to Editor**

Thank your for submitting a revised version of your work. While I agree with the arguments presented in your rebuttal, I cannot see how the contents of the manuscript was changed to account for the main comment (R1). Since the reviewer's view likely reflects the view of the average reader, please account for the comment in your revised manuscript by including some of the argumentation presented in the rebuttal in the manuscript. Please also consider using Fig. R1 in the manuscript (can be supplied as a supplementary figure). I hope including these changes won't take much time, since the text is effectively already there. Since the other comments seem adequately addressed, I should be able to accept a revised version without much delay.

**Response:** Thanks very much for your great efforts to assess our manuscript. We have made corrections/revisions as suggested in the revised manuscript. The Fig. R1 has been supplied as a supplementary figure in the supplement file (Fig. S1).

[revised manuscript text omitted]